# Ablation of SNX6 leads to defects in synaptic function of CA1 pyramidal neurons and spatial memory

Yang Niu[1,2,3†], Zhonghua Dai[1,2†], Wenxue Liu[4,5,6,7†], Cheng Zhang[1,2], Yanrui Yang[1,2], Zhenzhen Guo[1,2,3], Xiaoyu Li[1,3], Chenchang Xu[1,2,3], Xiahe Huang[1], Yingchun Wang[1], Yun S Shi[5,6,7*], Jia-Jia Liu[1,2*]

[1]State Key Laboratory of Molecular Developmental Biology, Institute of Genetics and Developmental Biology, Chinese Academy of Sciences, Beijing, China; [2]CAS Center for Excellence in Brain Science and Intelligence Technology, Chinese Academy of Sciences, Shanghai, China; [3]Graduate School, University of Chinese Academy of Sciences, Beijing, China; [4]Department of Anesthesiology, Jinling Hospital, School of Medicine, Nanjing University, Nanjing, China; [5]State Key Laboratory of Pharmaceutical Biotechnology, Nanjing University, Nanjing, China; [6]MOE Key Laboratory of Model Animal for Disease Study, Nanjing University, Nanjing, China; [7]Model Animal Research Center, Nanjing University, Nanjing, China

*For correspondence: yunshi@nju. edu.cn (YSS); jjliu@genetics.ac.cn (J-JL)

[†]These authors contributed equally to this work

Competing interests: The authors declare that no competing interests exist.

**Abstract** SNX6 is a ubiquitously expressed PX-BAR protein that plays important roles in retromer-mediated retrograde vesicular transport from endosomes. Here we report that CNS-specific *Snx6* knockout mice exhibit deficits in spatial learning and memory, accompanied with loss of spines from distal dendrites of hippocampal CA1 pyramidal cells. SNX6 interacts with Homer1b/c, a postsynaptic scaffold protein crucial for the synaptic distribution of other postsynaptic density (PSD) proteins and structural integrity of dendritic spines. We show that SNX6 functions independently of retromer to regulate distribution of Homer1b/c in the dendritic shaft. We also find that Homer1b/c translocates from shaft to spines by protein diffusion, which does not require SNX6. Ablation of SNX6 causes reduced distribution of Homer1b/c in distal dendrites, decrease in surface levels of AMPAR and impaired AMPAR-mediated synaptic transmission. These findings reveal a physiological role of SNX6 in CNS excitatory neurons.

## Introduction

SNX6 is a member of the sorting nexin (SNX) family that plays important roles in retromer-mediated, dynein−dynactin-driven retrograde vesicular transport from endosomes to the *trans*-Golgi network (TGN). The retromer complex functions in endosomal protein sorting and trafficking. It is composed of the VPS26-VPS29-VPS35 core complex and a SNX subunit or subcomplex (*Gallon and Cullen, 2015*). In mammalian epithelial cells, SNX6 serves as dynein adaptor in retromer-mediated vesicular transport to regulate both cargo recognition and release via its interaction with the motor and the target membrane. SNX6 contains an amino-terminal Phox Homology (PX) domain that is evolutionarily conserved among SNXs and a carboxyl-terminal Bin/Amphiphysin/Rvs (BAR) domain that allows for dimerization with BAR domains of other proteins. It dimerizes with the SNX1 subunit of retromer through its BAR domain and binds to dynactin p150[Glued] through its PX domain, linking the dynein−dynactin motor complex to retromer-associated vesicular cargoes (*Hong et al., 2009*; *Wassmer et al., 2009*). Its PX domain also interacts with the TGN-enriched phospholipid PtdIns(4)P,

**eLife digest** Neurons are the building blocks of the nervous system. These cells generally consist of a round portion called the cell body and a long cable-like axon. The cell body bears numerous branches called dendrites, which are in turn covered in spines. Neurons communicate with one another at junctions – or synapses – that typically form between the end of the axon of one cell and a dendritic spine on another.

Specialized proteins stabilize the dendritic spines and enable the cells to exchange messages across the synapse. However, it is the cell body – rather than the dendrites – that produces most of these proteins. Structures called molecular motors transport proteins to their destinations within the cell along fixed tracks, similar to how a freight train carries cargo over the rail network. One of the key molecular motors within neurons is called dynein–dynactin. This in turn interacts with other proteins called adaptors, enabling it to transport specific types of cargo.

Niu, Dai, Liu et al. have now examined the role of SNX6, an adaptor protein for the dynein–dynactin motor. Mice that have been genetically modified to lack SNX6 in their brains have fewer spines on their dendrites compared with normal mice. This was particularly true for dendrites that contain AMPAR, a protein that receives signals sent across synapses. Niu, Dai, Liu et al. showed that SNX6 interacts with another protein called Homer1b/c and is responsible for distributing this protein in dendrites far from the cell body. The Homer1b/c protein helps to stabilize dendritic spines and to regulate the number of AMPAR proteins within them. Mice that lack SNX6 therefore have less Homer1b/c in the dendrites furthest from the cell body, and fewer spines on these dendrites too. These mice also have fewer AMPAR proteins at their synapses than control mice.

Mice that lack SNX6 show impaired learning and memory compared to control mice. This is consistent with the fact that changes in the strength of synapses that possess AMPAR proteins are thought to underlie learning and memory. Additional experiments are required to explore these relationships further, and to determine whether SNX6 helps to localize any other proteins that also contribute to changes in the strength of synapses.

which inhibits the interaction between SNX6 and p150$^{Glued}$ to facilitate dissociation of the retrograde motor from the retromer-associated cargo at the TGN (*Niu et al., 2013*). Although retromer is involved in endosomal sorting and trafficking of amyloid precursor protein (APP) (*Fjorback et al., 2012*; *Sullivan et al., 2011*), and transport, surface expression and endocytic recycling of AMPAR (*Choy et al., 2014*; *Munsie et al., 2015*; *Zhang et al., 2012*) in neurons, the biological function of SNX6 in the CNS, whether retromer-dependent or not, remains to be explored.

In most of the principal neurons in the central nervous system (CNS), dendritic spines, the micron-sized membrane protrusions covering dendritic shaft, provide major sites of excitatory inputs. They are highly specialized postsynaptic structures containing transmembrane neurotransmitter receptors and proteins with signaling and scaffolding functions. Among them, scaffold proteins of the postsynaptic density (PSD) play crucial roles in glutamatergic neurotransmission by organizing glutamate receptors and signaling molecules at the postsynaptic terminal. One group of PSD scaffold proteins is the PSD95 membrane-associated guanylyl kinase (MAGUK) family proteins that anchor glutamate receptors to the PSD (*Elias and Nicoll, 2007*). Another group is the Homer and Shank family proteins. They interact with each other and form a high-order complex with a mesh-like network structure, which is believed to serve as a structural platform of the PSD essential for the structural integrity of dendritic spines (*Hayashi et al., 2009*). The Homer family proteins also regulate trafficking and signaling of the group one metabotropic glutamate receptors (mGluR1/5) and synaptic plasticity (*Ango et al., 2001*, *2002*; *Gerstein et al., 2012*; *Mao et al., 2005*; *Roche et al., 1999*; *Ronesi and Huber, 2008*). Moreover, Homer1b and 1c, the long isoforms encoded by the Homer1 gene that are differentiated by an insertion of 12 amino acid (aa) residues at aa 177 in Homer1c (*Xiao et al., 1998*), regulate surface expression of α-amino-3-hydroxy-5-methyl-4-isoxazolepropionic acid receptors (AMPAR) at synaptic sites through endocytic recycling (*Lu et al., 2007*). Although mechanisms underlying glutamate receptor trafficking to dendrites as well as their local trafficking into and out of synaptic sites have been intensively studied (*Anggono and Huganir, 2012*;

*Hoerndli et al., 2013*; *Horak et al., 2014*; *Huganir and Nicoll, 2013*; *Ladépêche et al., 2014*; *Setou et al., 2000*, *2002*), the molecular basis for dendritic distribution and spine localization of most PSD scaffolding proteins including Homer remains largely unexplored.

In this study, we investigated the physiological function(s) of SNX6 in mouse CNS neurons using multiple approaches including mouse genetics, behavior assays and electrophysiology, biochemistry and fluorescence imaging. Ablation of SNX6 in the CNS causes deficits in spatial learning and memory, decrease in spine density of the distal dendrites of hippocampal CA1 neurons and impairment of their AMPAR-mediated synaptic transmission, suggesting a role for SNX6 in synaptic structure and function. SNX6 interacts with Homer1b/c and loss of SNX6 leads to a reduction in its distribution in distal dendrites. Intriguingly, although SNX6 is required for the motility of a subpopulation of Homer1c on vesicles in dendritic shaft, live imaging and FRAP analyses indicate that Homer1c enters dendritic spines via protein diffusion but not SNX6-dependent active transport. Overexpression of SNX6 or Homer1c restores the spine density and AMPAR surface levels of $Snx6^{-/-}$ neurons. These findings uncover a physiological function for SNX6 in hippocampal CA1 excitatory neurons.

## Results

### Ablation of SNX6 in the CNS causes deficits in hippocampal-dependent spatial learning and memory

Immunoblotting analysis of tissue lysates indicated that SNX6 was ubiquitously expressed in mouse (*Figure 1—figure supplement 1A*). In mouse brain, SNX6 was expressed in both the somatodendritic area and processes of neurons in the cortex and the hippocampal formation (*Figure 1—figure supplement 1B–E*). Co-immunostaining with antibodies to SNX6 and axonal or dendritic markers revealed its distribution in a punctate pattern in both axon and dendrites of hippocampal neurons (*Figure 1—figure supplement 1F*). Quantitative analysis of fluorescence signal intensity revealed that SNX6 was primarily located in dendrites (*Figure 1—figure supplement 1G*). Moreover, SNX6 partially colocalized with PSD95, a postsynaptic marker in dendritic spines but not synaptophysin, a presynaptic marker (*Figure 1—figure supplement 1H*).

To investigate physiological function(s) of SNX6 in the CNS, we generated a conditional allele by floxing exon 5 of the *Snx6* gene (*Figure 1A–C*), and obtained CNS-specific knockout mice (CNS-*Snx6* KO) by mating *Snx6* conditional KO mice ($Snx6^{fl/fl}$) to Nestin promoter-driven Cre recombinase transgenic mice (*Nestin-Cre*). Lack of SNX6 protein expression in the CNS of *Nestin-Cre; $Snx6^{fl/fl}$* mice was confirmed by immunoblotting analysis of mouse brain homogenates (*Figure 1D*). The CNS neurons of $Snx6^{fl/fl}$ and *Nestin-Cre; $Snx6^{fl/fl}$* mice were hence referred to as $Snx6^{+/+}$ and $Snx6^{-/-}$ neurons, respectively. CNS-*Snx6* KO mice were born with the expected Mendelian ratio and appeared indistinguishable from wild-type littermates. Their brain size was comparable with that of wild-type (*Figure 1E*), and no gross abnormalities in the structure of the cortex, hippocampus and cerebellum were observed by histological examination (*Figure 1F*).

Next we conducted behavioral analyses on *Nestin-Cre; $Snx6^{fl/fl}$* mice and their wild-type littermates. No change in locomotor activity was detected by rotarod and open field assays (*Figure 2A, B*), and the mood levels of CNS-*Snx6* KO were also similar to that of wild-type mice in elevated plus maze, tail suspension and forced swimming tests (*Figure 2C–E*). In the Three-Chamber test, the CNS-*Snx6* KO mice showed no abnormality in sociability and social novelty (*Figure 2F*), nor did they display repetitive behaviors (*Figure 2G*). We then focused on their performance in learning and memory. Although *Nestin-Cre; $Snx6^{fl/fl}$* mice performed as well as their littermates in Y maze and shuttle box (*Figure 2H,I*), in the Morris water maze test, they were significantly retarded in spatial learning using latency traveled to reach the hidden platform as measures (*Figure 2J*). A probe trial showed that they were also severely impaired in spatial memory (*Figure 2K*). Moreover, these mice exhibited deficits in memory recall (*Figure 2L,M*). As the hippocampal region participates in the processes of the encoding, storage, consolidation and retrieval of spatial memory (*Riedel et al., 1999*), the behavioral phenotypes suggest that ablation of SNX6 affects synaptic function of hippocampal neurons.

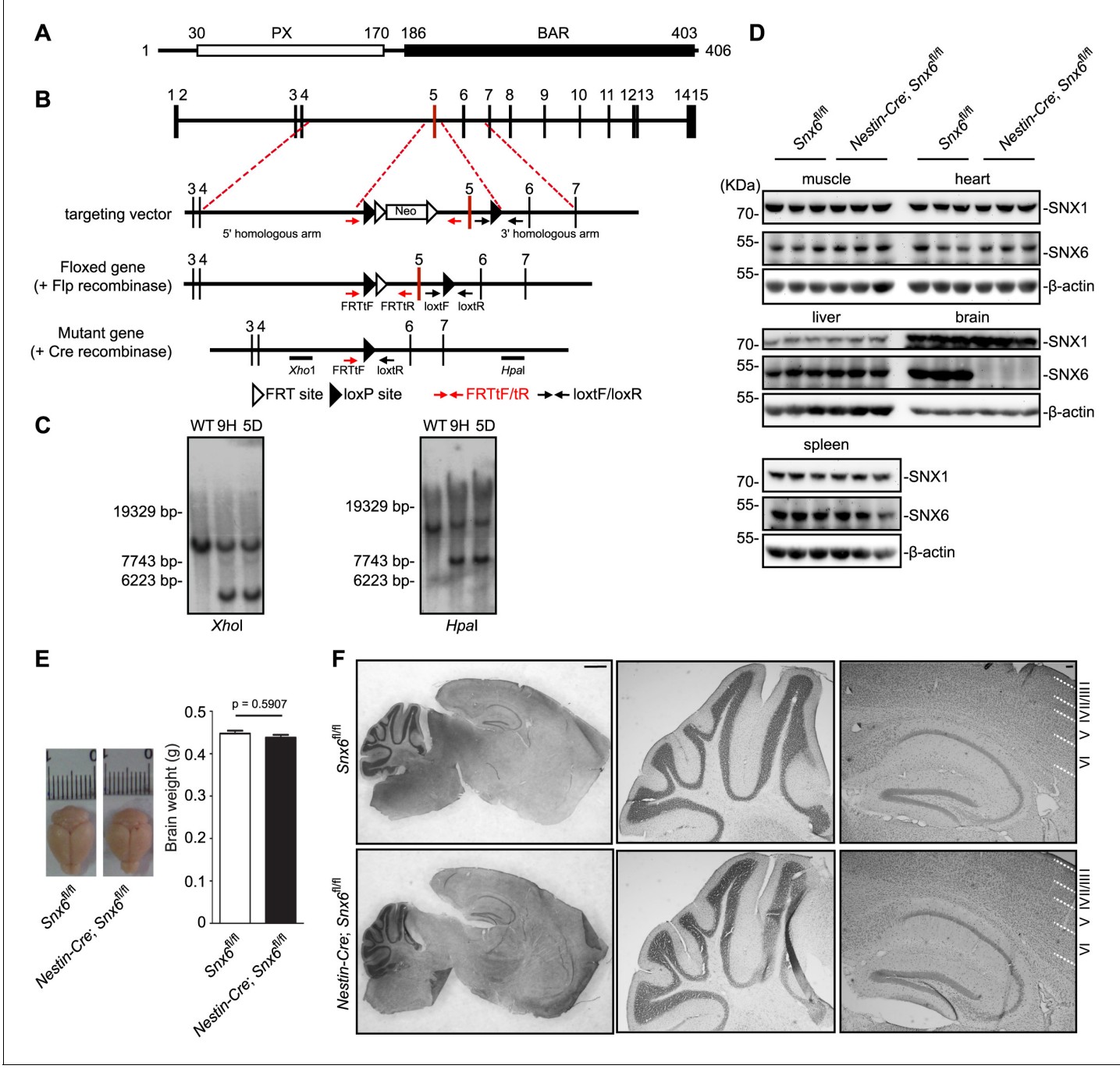

**Figure 1.** Generation and characterization of *Snx6* CNS-specific knockout mice. (**A**) Domain structure of SNX6. (**B**) Schematic diagram of the *Snx6* gene locus, the targeting vector, and the mutant alleles after homologous recombination. FRTtF/FRTtR and loxtF/loxtR: primer pairs used for genotyping. The *Xho*I and *Hpa*I probes used for Southern blotting analysis are shown. Neo: the neomycin resistance cassette. (**C**) Southern blotting analysis of wild-type (WT) and two independent clones of targeted ES cells (9 hr and 5D). (**D**) Immunoblots of tissue lysates from mouse littermates, probed with antibodies to SNX6. (**E**) Comparison of brain weight of *Snx6^fl/fl^* (15) and *Nestin-Cre; Snx6^fl/fl^* mice (12). Data represent mean ± SEM for each group. (**F**) Nissl staining of sagittal sections of whole brain from *Snx6^fl/fl^* and *Nestin-Cre; Snx6^fl/fl^* mice. Also shown are magnification of the cerebellum (middle panel) and the hippocampus/cortex area (right panel). Scale bar: 1 mm.

The following figure supplement is available for figure 1:

**Figure supplement 1.** Expression and subcellular distribution of SNX6 in the CNS.

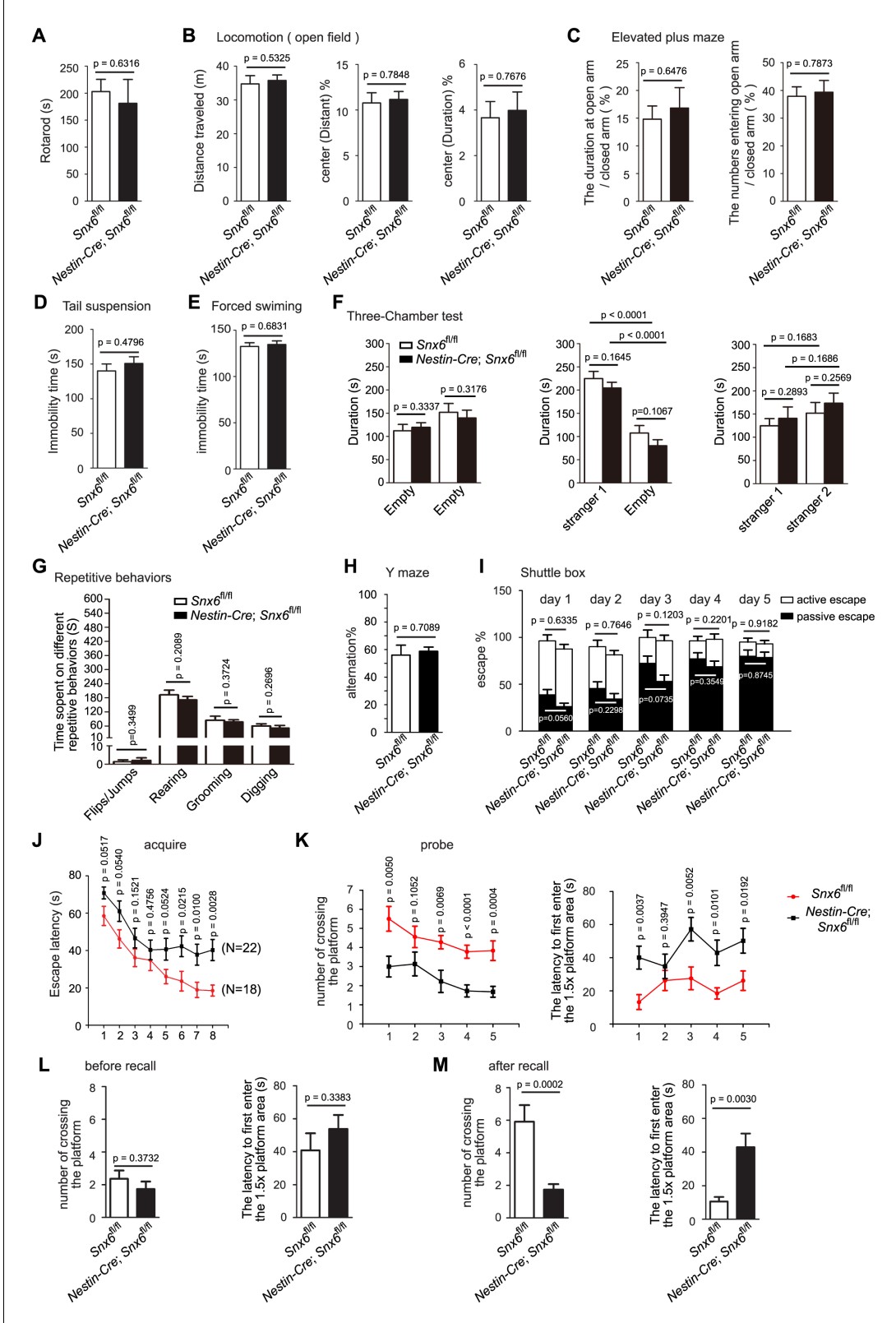

**Figure 2.** Impaired spatial learning and memory in *Nestin-Cre; Snx6^fl/fl* mice. (A–I) No effects of SNX6 ablation on the performance in assays of rotarod (A) (13 *Snx6^fl/fl* and 16 *Nestin-Cre; Snx6^fl/fl* mice), open field (B) (23 *Snx6^fl/fl* and 23 *Nestin-Cre; Snx6^fl/fl* mice), elevated plus maze (C) (14 *Snx6^fl/fl* and 13 *Nestin-Cre; Snx6^fl/fl* mice), tail suspension (D) (14 *Snx6^fl/fl* and 24 *Nestin-Cre; Snx6^fl/fl* mice), forced swimming (E) (15 *Snx6^fl/fl* and 25 *Nestin-Cre; Snx6^fl/fl* mice), Three-Chamber test (F) (10 *Snx6^fl/fl* and 9 *Nestin-Cre; Snx6^fl/fl* mice), repetitive behaviors (G) (12 *Snx6^fl/fl* and 10 *Nestin-Cre; Snx6^fl/fl* mice), Y maze

*Figure 2 continued on next page*

*Figure 2 continued*

(H) (11 *Snx6^{fl/fl}* and 15 *Nestin-Cre; Snx6^{fl/fl}* mice) and shuttle box (I) (20 *Snx6^{fl/fl}* and 13 *Nestin-Cre; Snx6^{fl/fl}* mice). The data represent mean ± SEM for each group. (J–K) Increased escape latency at acquisition learning (J) (data represent mean ± SEM of four trials per day), decreased number of crossing and increased latency to first enter the 1.5x area at probe test (K) (the data represent mean ± SEM for each group) in *Nestin-Cre; Snx6^{fl/fl}* mice in the Morris water maze. Subject numbers were 18 *Snx6^{fl/fl}* and 22 *Nestin-Cre; Snx6^{fl/fl}* mice. (L) After a 20-day rest, both *Snx6^{fl/fl}* and *Nestin-Cre; Snx6^{fl/fl}* mice exhibited memory extinguishment. (M) Decreased number of crossing and increased latency to first enter the 1.5x area at probe test in *Nestin-Cre; Snx6^{fl/fl}* mice after one recall training. The data represent mean ± SEM. N = 3 independent experiments.

## Ablation of SNX6 causes a decrease in spine density in distal apical dendrites of hippocampal CA1 pyramidal neurons

To investigate changes in synaptic function caused by SNX6 ablation at the cellular level, we examined neuronal morphology in the hippocampal region by crossing *Snx6^{fl/fl}* and *Nestin-Cre; Snx6^{fl/fl}* mice with *Thy1-EGFP* transgenic mice and analyzing brain sections by confocal microscopy (*Figure 3A*). We focused on the morphology of CA1 and CA3 pyramidal cells for two reasons: first, neurons in the CA1 and CA3 region were sparsely labeled by EGFP and hence easily distinguishable from neighboring ones for the purpose of morphological assessment; second, changes in the morphology and density of dendritic spines have been linked to synaptic function and plasticity. For quantification of spine number and morphology, we imaged segments of dendrites that are easily distinguishable from those of neighboring neurons, i.e., the oriens/distal branches of the basal and radiatum/thin branches of the apical dendrites of CA1 neurons, and secondary/tertiary branches of the basal and apical dendrites of CA3 neurons in stratum oriens and stratum radiatum, respectively (*Figure 3B*). Quantitative analysis showed that, although spine morphology did not change in either CA1 or CA3 pyramidal cells (*Figure 3C–F*), there was a decrease in the spine density of the distal portion of apical dendrites of *Snx6^{-/-}* CA1 neurons (*Figure 3C,D*). In contrast, no change in spine density was detected in *Snx6^{-/-}* CA3 neurons (*Figure 3E,F*). Consistently, ultrastructural analysis revealed a decrease in the number of asymmetric/excitatory synapses in the CA1, but not in the CA3 region of *Nestin-Cre; Snx6^{fl/fl}* mouse brain (*Figure 3G,H*). Together, these data indicate that SNX6 is required for spine morphogenesis and/or maintenance of distal apical dendrites of CA1 pyramidal neurons.

## SNX6 directly interacts with Homer1b/c

That ablation of SNX6 causes a decrease in spine density of distal dendrites suggests that it functions in the formation/stabilization of dendritic spines, probably via regulating dendritic distribution of postsynaptic proteins such as PSD components and/or neurotransmitter receptors. As the first step to investigate its molecular function, we determined the subcellular distribution of SNX6 in dendrites by co-immunostaining of SNX6 and vesicular markers in cultured mature hippocampal neurons. Confocal microscopy revealed that the majority of SNX6 signals colocalized with EEA1 and Rab5B, the early endosome markers (*Figure 4A,B*). SNX6 also partially colocalized with the late endosome marker Rab7 and Rab4, marker for the fast recycling pathway, though to a lesser extent, but not Rab11 recycling endosomes (*Figure 4A,B*). Intriguingly, although SNX6 signals did not colocalize with Golgin97, a TGN resident protein, they overlapped partially with TGN46 (*Figure 4A, B*), a protein involved in membrane traffic to and from the TGN (*Ponnambalam et al., 1996*), suggesting that SNX6 associates with endosomes and transport carriers in the dendrite.

Next we attempted to identify SNX6-interacting protein(s) in dendrite. We performed pull down experiment from mouse brain lysates using a His-tagged SNX6 N-terminus (aa 1–181, encompassing the PX domain) immobilized on Ni-NTA agarose. Mass spectrometry analysis revealed six matching peptides with 24% sequence coverage corresponding to Homer1b/c, a postsynaptic scaffold protein (*Figure 4C*). Immunoblotting analysis verified that Homer1b/c, not Homer1a, the shorter isoform encoded by the *Homer1* gene, was pulled down by the N-terminus of SNX6 but not SNX1 (*Figure 4D,E*). Moreover, in vitro binding assays showed that SNX6-N interacted directly with the coiled-coil domain of Homer1b/c (*Figure 4F*), which is not present in Homer1a. In contrast, neither Homer2b nor Homer3, the longer isoforms of other Homer family members, interacted with SNX6-N (*Figure 4G*). Further, mEmerald-Homer1c co-immunoprecipitated with Flag-tagged SNX6 in transiently transfected HEK293T cells (*Figure 4H*). Consistently, reciprocal co-immunoprecipitations of

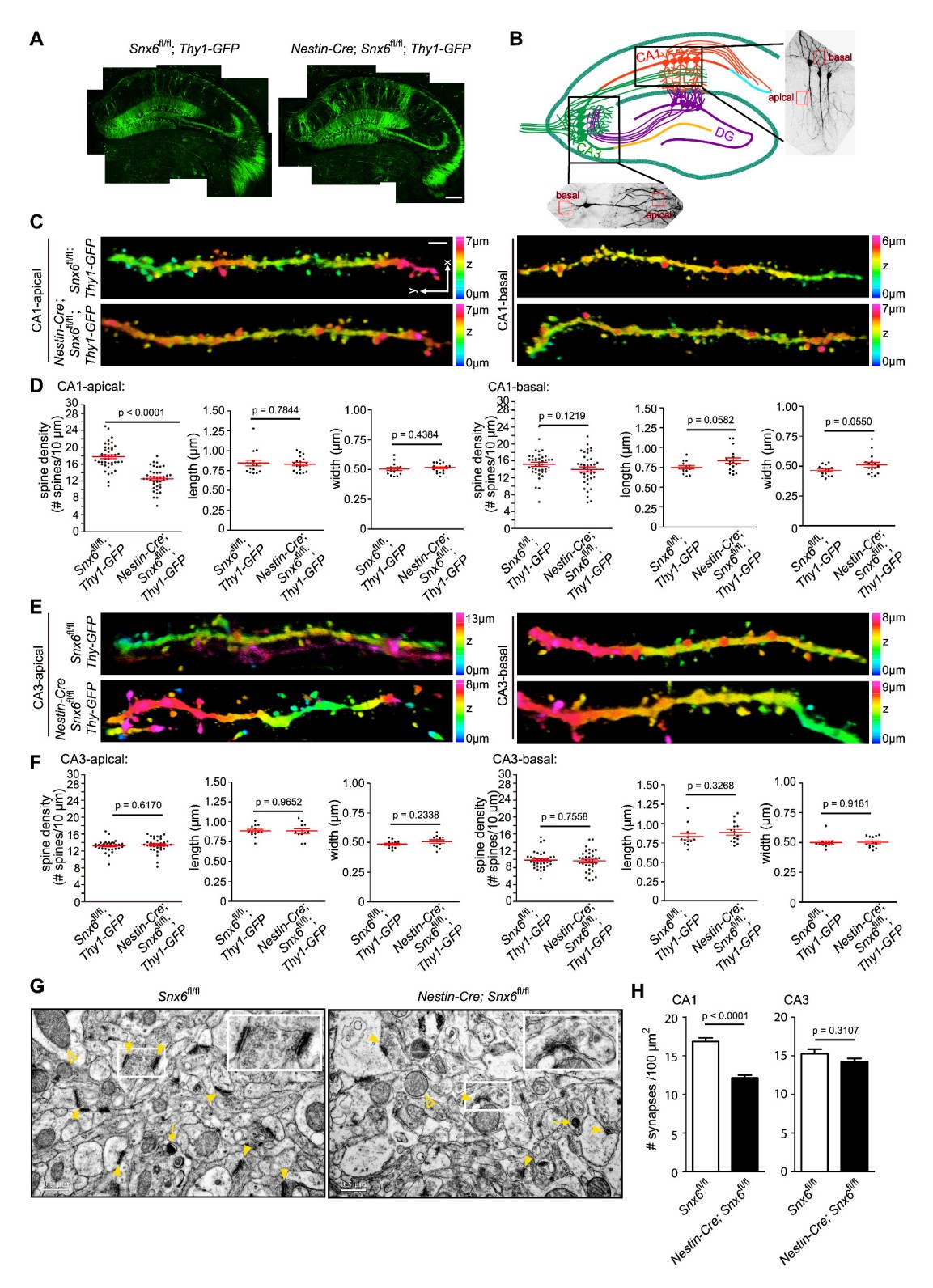

**Figure 3.** Decreases in spine density of hippocampal CA1 apical dendrites and number of excitatory synapses in the CA1 region in *Nestin-Cre; Snx6*[fl/fl] Mice. (A) Confocal images of coronal sections of hippocampi from *Snx6*[fl/fl]*; Thy1-GFP* and *Nestin-Cre; Snx6*[fl/fl]*; Thy1-GFP*. (B) Schematic of the location of the dendritic segments selected for morphological analysis. (C) Representative 3D-reconstructed confocal images of dendrites of CA1 pyramidal cells. The z-dimension position is color-coded according to the color scale bar. (D) Quantification of spine density (n = 5 pairs of mice, apical/basal: 43/

*Figure 3 continued on next page*

Figure 3 continued

40 cells, 95/82 dendritic segments and 4979/3721 spines for *Snx6fl/fl; Thy1-GFP*; 40/38 cells, 96/78 dendritic segments and 3662/3341 spines for *Nestin-Cre; Snx6fl/fl; Thy1-GFP*) and morphology (n = 2 pairs, apical/basal: 16/14 cells, 1053/764 spines for *Snx6fl/fl; Thy1-GFP*; 18/17 cells, 1099/894 spines for *Nestin-Cre; Snx6fl/fl; Thy1-GFP*) of CA1 dendrites. (E) Representative 3D-reconstructed confocal images of CA3 dendrites. (F) Quantification of spine density (n = 5 pairs, apical/basal: 34/34 cells, 80/75 dendritic segments and 3155/2261 spines for *Snx6fl/fl; Thy1-GFP*; 34/34 cells, 77/73 dendritic segments and 3053/2216 spines for *Nestin-Cre; Snx6fl/fl; Thy1-GFP*) and morphology (n = 2 pairs, apical/basal: 13/12 cells, 822/595 spines for *Snx6fl/fl; Thy1-GFP*; 11/14 cells, 739/568 spines for *Nestin-Cre; Snx6fl/fl; Thy1-GFP*) of CA3 dendrites. (G) Representative TEM images of hippocampal CA1 regions of adult animals. Yellow solid arrowheads indicate asymmetric (excitatory) synapses. Insets are representative higher magnification images of synapses in the boxed areas. Yellow empty arrowheads indicate mitochondria. Yellow arrows indicate lysosomes. (H) Quantification of synapse density (n = 3 pairs, CA1: 1553 synapses for *Snx6fl/fl* and 1038 synapses for *Nestin-Cre; Snx6fl/fl*. CA3: 1102 synapses for *Snx6fl/fl* and 1069 synapses for *Nestin-Cre; Snx6fl/fl*). Data represent mean ± SEM. Bars: 200 µm in (A), 2 µm in (C) and 500 nm in (G).

endogenous proteins from mouse brain lysates verified that SNX6 and Homer1b/c interact with each other (*Figure 4I*, left and center panels). Moreover, immunoisolation of SNX6-positive vesicles from membrane fractions of mouse brain lysates detected Homer1b/c together with p150[Glued] and dynein intermediate chain (DIC), subunits of the dynein−dynactin complex (*Figure 4I*, right panel). In contrast, neither Homer1a nor subunits of the N-methyl-D-aspartate receptor (NMDAR) were detected on SNX6-positive vesicles (*Figure 4I*, right panel). In dendrites, Homer1b/c not only colocalized with SNX6 on vesicular structures (*Figure 4J,K*), but also colocalized with EEA1 (*Figure 4L,M*). Both wide-field microscopy with deconvolution and superresolution fluorescence microscopy revealed colocalization of EEA1, Homer1b/c and SNX6 in dendrites (*Figure 4N*, *Table 1*, *Figure 4—figure supplement 1* and *Video 1*), indicating that SNX6 associates with Homer1b/c on endosomes.

## Ablation of SNX6 causes decrease in distribution of Homer1b/c in distal dendrites

Dendritic distribution of Homer1b/c is essential for its scaffolding and signaling functions at the PSD. To determine whether SNX6 regulates Homer1b/c distribution in dendrites, we examined Homer1b/c expression and subcellular distribution in *Snx6-/-* hippocampal neurons by immunofluorescence staining. Indeed, although no change in the protein levels of Homer1b/c was detected in the hippocampi of *Nestin-Cre; Snx6fl/fl* mice (*Figure 5—figure supplement 1*), quantitative analysis revealed not only a decrease in the number of Homer1b/c puncta in dendritic segments (30–120 µm from the cell body) of *Snx6-/-* neurons as compared with that of *Snx6+/+*, but also a significant reduction in the fluorescence intensity of Homer1b/c puncta in spines (*Figure 5A,B*). Overexpression of EGFP-SNX6 rescued both puncta number and spine distribution of Homer1b/c in *Snx6-/-* neurons (*Figure 5A,B*). In contrast, neither the number nor the fluorescence intensity of PSD95 puncta was significantly affected in *Snx6-/-* neurons (*Figure 5C,D*). Further, quantification of the Homer1b/c signal intensity over distance from the cell body revealed a decrease in both shaft and spines of the distal dendrites and concurrent accumulation in the soma of *Snx6-/-* neurons, which was rescued by mCherry-SNX6 (*Figure 5E,F*). Together these data indicate that SNX6 is required for Homer1b/c distribution in distal dendrites.

Homer and Shank are among the most abundant postsynaptic scaffolding proteins that contribute to the structural and functional integrity of dendritic spines. Consistent with the in vivo data, the spine density of *Snx6-/-* neurons was lower than that of *Snx6+/+* in dissociated cultures (*Figure 5G,H*, and *Figure 5—figure supplement 2*). Overexpression of EGFP-SNX6 or mEmerald-Homer1c but not a Homer1c fragment that is truncated of its mGluR1/5-binding EVH1 domain (Homer1c-C) (*Shiraishi-Yamaguchi and Furuichi, 2007*) restored the spine density of *Snx6-/-* neurons (*Figure 5G,H*), indicating that SNX6-dependent dendritic distribution of Homer1b/c contributes to spine formation/stabilization, and that the cellular function of Homer1b/c as postsynaptic scaffold protein is required to restore the number of spines.

## Active transport of a fraction of Homer1b/c molecules in the dendritic shaft requires SNX6

Given that SNX6 is a cargo adaptor for the microtubule-based dynein−dynactin motor, we reasoned that SNX6 might mediate transport of Homer1b/c in dendrites. To this end, we transfected

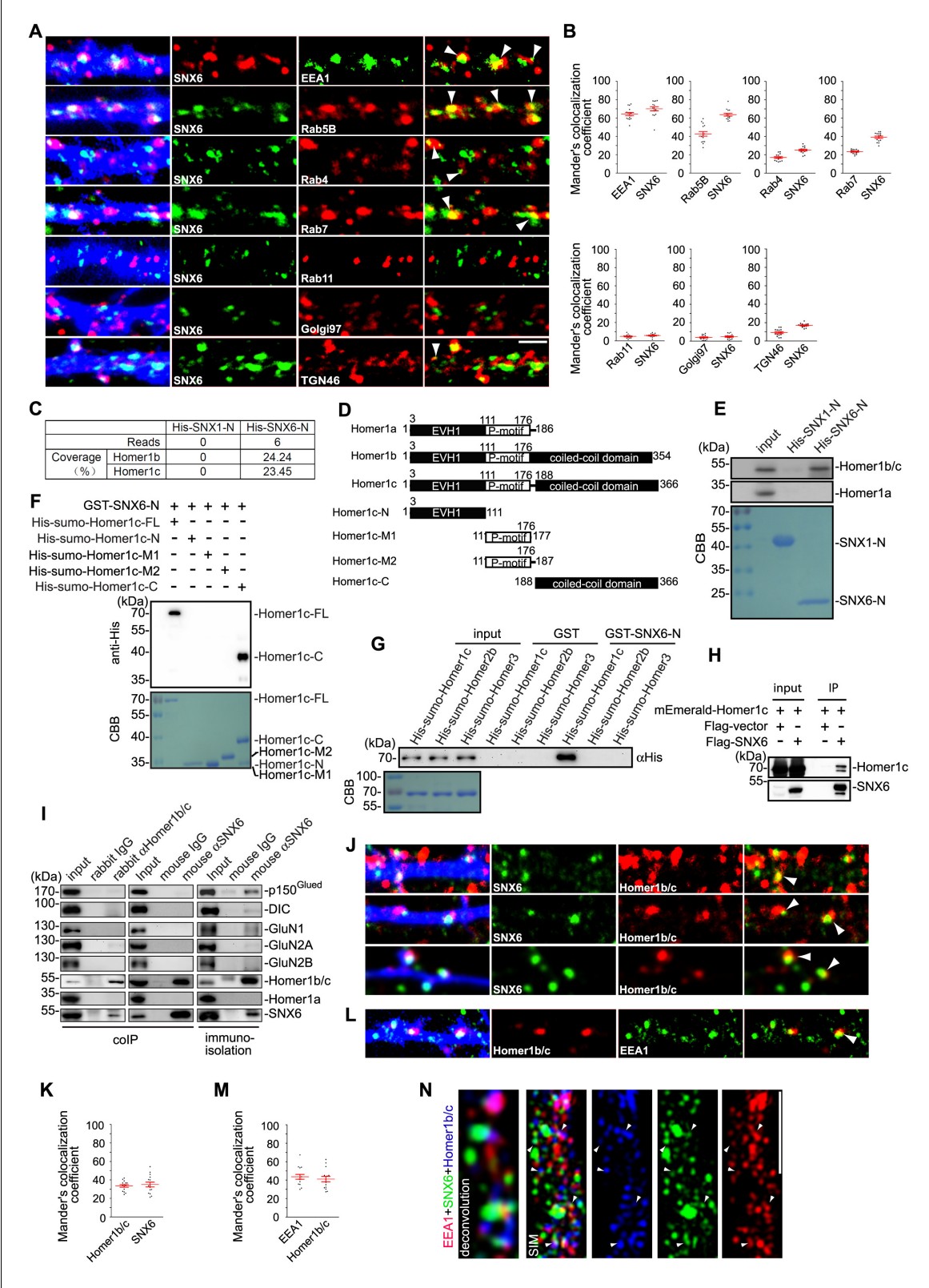

**Figure 4.** SNX6 interacts with Homer1b/c and colocalizes with Homer1b/c on endosomes. (**A**) Hippocampal neurons were transfected with pLL3.7.1 on DIV14 to express DsRed as volume marker, fixed on DIV17 and immunostained with antibodies to SNX6 and vesicular markers. DsRed is pseudocolored for presentation. White arrowheads indicate overlapped signals. (**B**) Quantification of colocalization in (**A**) from 45 dendritic segments of 15 neurons (mean ± SEM, N = 3. Total length of dendrites: 1568 μm for EEA1; 1447 μm for Rab5; 1637 μm for Rab4; 1489 μm for Rab7; 1319 μm for Rab11; 1207 μm

*Figure 4 continued on next page*

Figure 4 continued

for Golgi97 and 1462 μm for TGN46). (C) Mouse brain lysates were incubated with His-SNX1-N or His-SNX6-N immobilized on Ni-NTA agarose. Bound proteins were subjected to SDS-PAGE and mass spectrometry analysis. The table shows the number of Homer1b/c unique peptides identified by mass spec analysis and their sequence coverage. (D) Schematic representation of the domain structure of Homer1 isoforms and Homer1c fragments used in this study. (E) Upper panels: immunoblotting of bound proteins in (C). Lower panel: coomassie brilliant blue (CBB) stained SDS-PAGE gel shows purified recombinant proteins. (F) Mapping of SNX6-Homer1b/c interaction sites by in vitro binding assay. (G) In vitro binding assay of SNX6 and Homer family members. (H) Lysates from HEK293 cells overexpressing Flag-SNX6 and mEmerald-Homer1c were subjected to co-IP with Flag M2 beads, followed by immunoblotting with antibodies to Flag and Homer1b/c. (I) Total lysates and membrane fractions from mouse brain lysates were subjected to IP and immunoisolation with antibodies to Homer1b/c or SNX6, and antibodies to SNX6 coupled to Dynabeads Protein G, respectively. Shown are immunoblots probed with antibodies to SNX6, p150$^{Glued}$, DIC, GluN1, GluN2A, GluN2B, Homer1b/c and Homer1a. (J) DIV18 neurons were immunostained with antibodies to Homer1b/c and SNX6. (K) Quantification of colocalization in (J) from 45 dendritic segments of 15 neurons (mean ± SEM, N = 3 independent experiments. Total length 1677 μm). (L) DIV18 neurons were immunostained with antibodies to Homer1b/c and EEA1. (M) Quantification of colocalization in (L) from 45 dendritic segments of 15 neurons (mean ± SEM, N = 3. Total length 1459 μm). (N) DIV18 neurons were immunostained with antibodies to EEA1, Homer1b/c, and SNX6. Superresolution images were captured by structured illumination microscopy (SIM). White arrowheads indicate overlaps of signals from different channels. Bars: 2 μm.

The following figure supplement is available for figure 4:

**Figure supplement 1.** Colocalization analysis of superresolution images of triple-stained neurons captured by 3D-SIM (15 neurons for each immunostaining experiment).

hippocampal neurons with constructs expressing mEmerald-Homer1c and mCherry-SNX6 (*Figure 6— figure supplement 1*) and performed live-cell imaging by total internal reflection fluorescence microscopy (TIR-FM) to monitor their movement in dendrites. Indeed, we observed movement of SNX6-, Homer1c-double positive puncta in the shaft of both proximal and distal dendrites (*Figure 6A–C*, *Figure 6—figure supplement 2A,B*, *Videos 2* and *3*). A retrospective staining of MAP2 right after live imaging verified dendrite identity of the distal branch (*Figure 6C*). Quantitative analysis revealed that, similar to mobility characteristics of PSD95 clusters in dendrites (*Gerrow et al., 2006*), the majority (~90%) of Homer1c puncta (1022 out of 1128 puncta from 31 neurons) were stationary. Of note, the majority of motile SNX6-, Homer1c-positive puncta were smaller than 0.3 μm² in size (<600 nm in apparent diameter), whereas most of the immotile ones were larger (*Figure 6—figure supplement 2C*), suggesting that the moving structures were vesicles rather than large protein aggregates. The SNX6-, Homer1c-positive puncta moved bidirectionally in the dendritic shaft, with the mean velocity of 0.416 ± 0.037 μm/s, over distances ranging from 2.026 to 18.324 μm (*Figure 6—figure supplement 2D–F*). Movement of SNX6-, Homer1b-double positive puncta in the dendritic shaft was also observed by live imaging (*Video 4*). In contrast, in neurons overexpressing EGFP fusion of the GluN1 subunit of NMDAR, no comovement of SNX6- and GluN1-positive vesicles in dendrite was observed (*Video 5*). Moreover, we performed live imaging of mEmerald-Homer1c in *Snx6*$^{-/-}$ neurons and found that compared with wild-type, there was a dramatic decrease in the fraction of motile Homer1c fluorescent puncta (29 motile puncta out of 311 from 10 *Snx6*$^{+/+}$ neurons vs. 10 out of 1217 from 40 *Snx6*$^{-/-}$ neurons).

To verify that Homer1b/c-associated vesicles are of early endosome origin, we performed live imaging on neurons expressing fluorescently tagged EEA1. First, we imaged neurons coexpressing EEA1-YFP and mCherry-SNX6 and observed movement of EEA1-, SNX6-double positive vesicles in the dendritic shaft (*Video 6*). In neurons coexpressing EEA1-YFP and mCherry-Homer1c, more than 90% of EEA1 vesicles also contained Homer1c (461/482 = 95.6%from 20 *Snx6*$^{+/+}$ and 796/ 872 = 91.3 % from 26 *Snx6*$^{-/-}$ neurons), whereas the majority of Homer1c puncta were also EEA1-positive (461/690 = 66.8 % from 20 *Snx6*$^{+/+}$ and 796/1249 = 63.7% from 26 *Snx6*$^{-/-}$ neurons). Live imaging detected not only movement of EEA1-, Homer1c-double positive vesicles in the dendritic shaft (*Video 7*), but also a significant decrease in the fraction of motile EEA1-positive vesicles in *Snx6*$^{-/-}$ neurons (total EEA1-positive vesicles: 58/482 motile = 12% from 20 *Snx6*$^{+/+}$ neurons vs. 37/ 872 motile = 4.2% from 26 *Snx6*$^{-/-}$ neurons; EEA1-, Homer1c-double positive vesicles: 53/461 motile = 11.5% from 20 *Snx6*$^{+/+}$ neurons vs. 25/796 motile = 3.1% from 26 *Snx6*$^{-/-}$ neurons). In contrast, in dendrites of neurons coexpressing Homer1c and the late endosome marker Rab7, neither comovement of Rab7- and Homer1c-positive structures was observed (*Videos 8* and *9*), nor the fraction of motile Rab7-labeled structures changed significantly when SNX6 was ablated (331/601

**Table 1.** Quantitative analysis of colocalization of signals in superresolution images and statistical significance of colocalization (related to **Figures 4N** and **6D**, and **Figure 4—figure supplement 1**).

voxel colocolization values (%) / p value

**EEA1-SNX6-Homer1b/c**

| | | | | | | | | | | | | | | | |
|---|---|---|---|---|---|---|---|---|---|---|---|---|---|---|---|
| EEA1 with SNX6 and Homer1b/c | 15.68/ 0.00079 | 12.05/ 0.011 | 17.48/ 0.0086 | 15.18/ 0.0102 | 16.92/ 0.00864 | 16.48/ 0.0127 | 15.33/ 0.0092 | 18.86/ 0.0069 | 15.76/ 0.0089 | 11.96/ 0.0141 | 21.87/ 0.00053 | 15.98/ 0.00065 | 14.58/ 0.0096 | 16.76/ 0.001 | 19.04/ 0.0005 |
| SNX6 with EEA1 and Homer1b/c | 18.56/ 0.00079 | 13.12/ 0.011 | 17.28/ 0.0086 | 13.96/ 0.0102 | 17.44/ 0.00864 | 12.36/ 0.0127 | 15.92/ 0.0092 | 22.64/ 0.0069 | 15.08/ 0.0089 | 14.32/ 0.0141 | 16.34/ 0.00053 | 19.92/ 0.00065 | 15.12/ 0.0096 | 15.44/ 0.001 | 17.56/ 0.0005 |
| Homer1b/c with EEA1 and SNX6 | 14.24/ 0.00079 | 13.16/ 0.011 | 13.64/ 0.0086 | 10.22/ 0.0102 | 16.28/ 0.00864 | 16.84/ 0.0127 | 13.24/ 0.0092 | 13.92/ 0.0069 | 14.84/ 0.0089 | 14.52/ 0.0141 | 18.56/ 0.00053 | 15.36/ 0.00065 | 13.92/ 0.0096 | 13.44/ 0.001 | 17.63/ 0.0005 |

**p150$^{Glued}$-SNX6-Homer1b/c**

| | | | | | | | | | | | | | | | |
|---|---|---|---|---|---|---|---|---|---|---|---|---|---|---|---|
| p150$^{Glued}$ with SNX6 and Homer1b/c | 15.63/ 0.004 | 8.13/ 0.01 | 10.06/ 0.015 | 10.06/ 0.0102 | 10.06/ 0.0053 | 10.06/ 0.0047 | 10.06/ 0.003 | 10.06/ 0.0069 | 10.06/ 0.0187 | 7.62/ 0.020< p <0.0460 | 16.38/ 0.0031 | 9.35/ 0.0051 | 8.85/ 0.008< p <0.0340 | 9.97/ 0.00187 | 9.21/ 0.0071 |
| SNX6 with Homer1b/c and p150$^{Glued}$ | 14.57/ 0.004 | 12.89/ 0.01 | 10.41/ 0.015 | 13.02/ 0.0102 | 10.18/ 0.0053 | 12.67/ 0.0047 | 18.49/ 0.003 | 12.33/ 0.0069 | 9.51/ 0.0187 | 9.54/ 0.010< p <0.0250 | 23.55/ 0.0031 | 11.24/ 0.0051 | 9.39/ 0.008< p <0.0320 | 9.18/ 0.00187 | 10.7/ 0.0071 |
| Homer1b/c with SNX6 and p150$^{Glued}$ | 10.32/ 0.004 | 8.27/ 0.01 | 8.33/ 0.015 | 7.15/ 0.0102 | 9.66/ 0.0053 | 9.8/ 0.0047 | 13.16/ 0.003 | 8.49/ 0.0069 | 8.73/ 0.0187 | 8.69/ 0.0120< p <0.0366 | 15.07/ 0.0031 | 9.17/ 0.0051 | 8.24/ 0.0100< p <0.0370 | 8.43/ 0.00187 | 9.03/ 0.0071 |

**DIC-SNX6-Homer1b/c**

| | | | | | | | | | | | | | | | |
|---|---|---|---|---|---|---|---|---|---|---|---|---|---|---|---|
| DIC with SNX6 and Homer1b/c | 9.21/ 0.0067 | 9.65/ 0.0061 | 20.05/ 0.0049 | 7.9/ 0.007 | 13.41/ 0.006 | 15.25/ 0.0058 | 9.91/ 0.00187 | 10.88/ 0.0072 | 9.41/ 0.0079 | 8.53/ 0.0061 | 10.61/ 0.0071 | 9.64/ 0.0083 | 9.78/ 0.0082 | 10.98/ 0.0086 | 16.12/ 0.0042 |
| SNX6 with DIC and Homer1b/c | 12.00/ 0.0067 | 13.19/ 0.0061 | 14.91/ 0.0049 | 12.02/ 0.007 | 13.07/ 0.006 | 15.49/ 0.0058 | 11.32/ 0.00187 | 9.81/ 0.0072 | 10.1/ 0.0079 | 16.07/ 0.0061 | 10.79/ 0.0071 | 12.48/ 0.0083 | 11.77/ 0.0082 | 9.86/ 0.0086 | 23.11/ 0.0042 |
| Homer1b/c with DIC and SNX6 | 11.43/ 0.0067 | 9.81/ 0.0061 | 13.78/ 0.0049 | 9.82/ 0.007 | 10.71/ 0.006 | 10.86/ 0.0058 | 9.77/ 0.00187 | 9.82/ 0.0072 | 9.22/ 0.0079 | 10.36/ 0.0061 | 9.75/ 0.0071 | 7.87/ 0.0083 | 9.73/ 0.0082 | 10.21/ 0.0086 | 17.17/ 0.0042 |

motile = 55.1% from 27 $Snx6^{+/+}$ neurons vs. 341/647 motile = 52.7% from 22 $Snx6^{-/-}$ neurons). Together, these data indicate that SNX6 is required for the motility of EEA1-positive early endosomes or early endosome-derived vesicles carrying Homer1b/c.

Next we determined whether dynein−dynactin is involved in SNX6-regulated dendritic distribution of Homer1b/c. Consistent with the immunoisolation data (**Figure 4I**), superresolution microscopy analysis revealed that a significant fraction of both p150$^{Glued}$ and DIC colocalized with SNX6 and Homer1b/c on vesicular structures (**Figure 6D**, **Table 1**, **Figure 4—figure supplement 1** and **Videos 10** and **11**). Because dynein is essential for cell viability, we transiently treated neurons with Ciliobrevin D, an inhibitor of dynein activity (**Firestone et al., 2012**), and detected a decrease in both shaft and synaptic distribution of Homer1b/c in distal dendrites (**Figure 6E–H**). Further, overexpression of p150$^{Glued}$-N, a dominant negative mutant that disrupts the interaction between SNX6 and dynactin (**Hong et al., 2009**), also caused a decrease in Homer1b/c signals in distal dendrites and concurrent increase in the soma (**Figure 6I,J**), indicating a role for dynein−dynactin in the dendritic distribution of Homer1b/c.

Next we asked whether SNX6 serves as a linker between the vesicles carrying Homer1b/c and the dynein−dynactin motor complex. To determine whether association of Homer1b/c and the motor complex on vesicles requires SNX6, we attempted immunoisolation of Homer1b/c and dynein−dynactin-associated vesicles from mouse brain. Although antibodies to Homer1b/c failed to isolate Homer1b/c-positive vesicles from membrane fractions of mouse brain lysates, western blotting of vesicles immunoisolated with antibodies to both p150$^{Glued}$ and DIC detected Homer1b/c

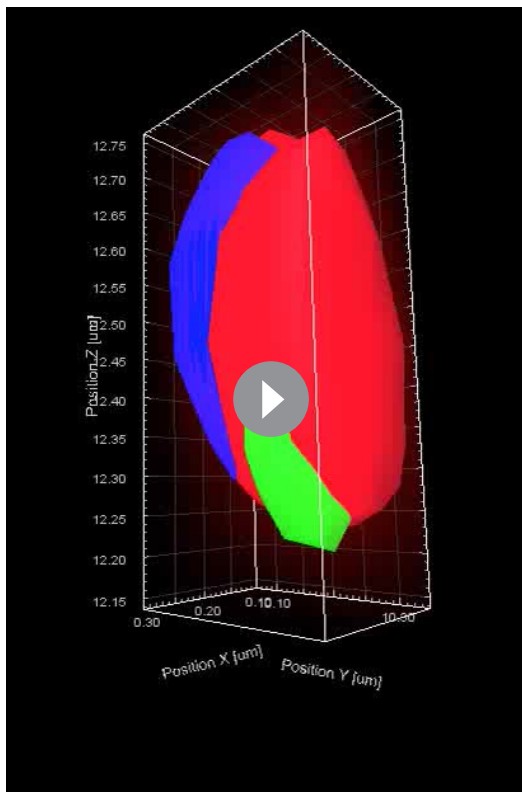

**Video 1.** 3D-SIM movie of an enlarged region of interest from a hippocampal neuron dendrite shows the association of EEA1 (red), SNX6 (green) and Homer1b/c (blue).

signals in wild-type but not CNS-SNX6 KO mice (*Figure 6K*). Moreover, we performed immunostaining and colocalization analysis of Homer1b/c on *Snx6*[+/+] and *Snx6*[-/-] neurons. Quantitative analysis of confocal images showed that there was a decrease in colocalization of Homer1b/c with p150[Glued] and DIC, but not with EEA1, in *Snx6*[-/-] neurons (*Figure 6L,M*). Together, these data suggest that SNX6 mediates dynein–dynactin-driven transport of a fraction of Homer1b/c molecules in the dendritic shaft.

### The Homer1b/c trafficking pathway is distinct from the PSD95 and secretory trafficking pathways in dendrite

Although SNX6 mediates association of Homer1b/c vesicles with dynein–dynactin, the majority of Homer1b/c puncta were immotile in steady-state neurons, suggesting that SNX6 might regulate Homer1b/c levels in distal dendrites via mechanisms other than vesicular transport. Moreover, ablation of SNX6 did not cause complete loss of Homer1/c from distal dendrites, suggesting that SNX6-independent mechanism(s) is required for Homer1b/c distribution in dendritic regions far from the cell body. Previous studies have found that the postsynaptic scaffolding proteins PSD95, guanylate kinase domain-associated protein (GKAP) and Shank are transported in a preformed protein complex in dendrite (*Gerrow et al., 2006*). To determine whether Homer1b/c shares the same trafficking pathway with PSD95, we performed live imaging of neurons expressing PSD95-RFP and determined association of SNX6 and Homer1b/c with motile PSD95 clusters by retrospective immunofluorescence staining. No SNX6/Homer1b/c signals were detected on motile PSD95 clusters (seven motile PSD95 puncta from five cells, *Figure 7A* and *Video 12*), indicating that Homer1b/c is not cotransported with PSD95.

Next we asked whether the secretory trafficking pathway in dendrite contributes to Homer1b/c distribution in dendritic shaft and spines. To address this question, we transfected neurons with plasmid overexpressing a kinase-dead version of protein kinase D1 (PKD1-K618N, or PKD-KD), which blocks secretory trafficking by preventing fission of transport carriers from the TGN (*Liljedahl et al., 2001*). Expression of PKD-KD did not affect either dendritic distribution or spine localization of Homer1b/c (*Figure 7B–E*), indicating that its dendritic trafficking and synaptic delivery does not rely on the secretory pathway.

### Translocation of Homer1b/c from dendritic shaft to spines is SNX6- and vesicular transport-independent

As localization of Homer1b/c to synaptic sites in dendritic spines is crucial for its function, next we asked the question whether its translocation from shaft to spines requires SNX6 and vesicular transport. First, we determined whether or not shaft-localized Homer1b/c puncta enter spines. In our live imaging experiments using mEmerald-Homer1c, we did not detect entry of Homer1c puncta into spines in either wild-type or *Snx6* KO neurons. We also imaged dendritic segments of neurons expressing mCherry-Homer1c (*Snx6*[+/+]: 30 cells, 50 dendritic segments, 791 spines; *Snx6*[-/-]: 30 cells, 50 dendritic segments, 547 spines). Under the experimental condition we used, most motile

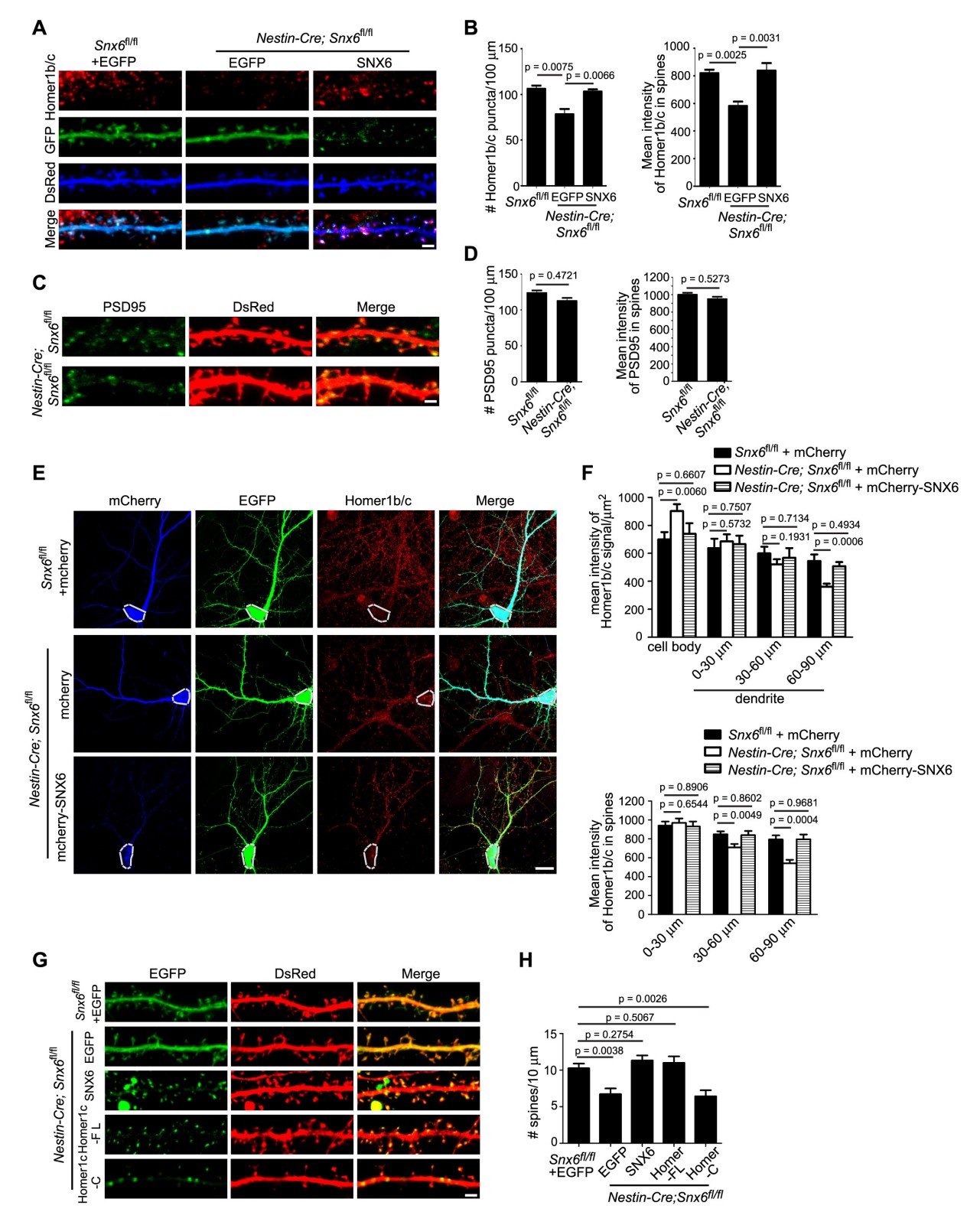

**Figure 5.** Partial loss of Homer1b/c from distal dendrites of $Snx6^{-/-}$ neurons. (**A**) Neurons were co-transfected with pLL3.7.1 and EGFP or EGFP-SNX6 construct on DIV13, fixed on DIV18 and immunostained with antibodies to Homer1b/c. Shown are representative confocal images of dendritic segments. (**B**) Quantification of puncta number per 100 μm dendrite length and mean intensity in spines for Homer1b/c (mean ± SEM, n = 30, N = 3). (**C**) Neurons were transfected with pLL3.7.1 on DIV13, fixed on DIV18 and immunostained with antibodies to PSD95. (**D**) Quantification of PSD95

*Figure 5 continued on next page*

*Figure 5 continued*

distribution in dendrites (mean ± SEM, n = 30, N = 3). (E) DIV14 neurons were co-transfected with constructs overexpressing EGFP and mCherry or mCherry-SNX6, fixed on DIV16 and immunostained with antibodies to Homer1b/c. shown are representative confocal images of transfected neurons. Dashed lines outline the cell bodies. (F) Quantification of Homer1b/c distribution in the cell body and dendrites, and its mean intensity in spines in (E) (mean ± SEM, n = 30, N = 3). (G) DIV13 neurons were co-transfected with constructs expressing DsRed and EGFP, EGFP-SNX6, mEmerald-Homer1c-FL or EGFP-Homer1c-C and fixed on DIV18. (H) Quantification of spine density in (G) (mean ± SEM, n = 30, N = 3). Bars: 20 μm in (E), 2 μm in other panels.

The following figure supplements are available for figure 5:

**Figure supplement 1.** Immunoblotting analysis of Homer1b/c and PSD95 in hippocampi from *Snx6^(fl/fl)* and *Nestin-Cre; Snx6^(fl/fl)* mice.

**Figure supplement 2.** Ablation of SNX6 causes decrease in spine density of CA1 but not CA3 neurons.

fluorescent puncta moved in shaft (*Figure 8A* and *Video 13*), only one event of Homer1c particle entry into spine was observed (*Video 14*). Since the plus ends of microtubules transiently invade dendritic spines (*Jaworski et al., 2009*) and dynein is a minus end-directed motor, it is conceivable that dynein-driven transport is not involved in transfer of Homer1b/c from shaft to spines. Moreover, we reasoned that if delivery of Homer1b/c from shaft to spines requires SNX6, there would be a decrease in spine distribution of Homer1b/c signals in dendrites of *Snx6* KO neurons. Quantitative analysis showed that, although both total and spine Homer1b/c signals decreased in distal dendrites (*Figure 5F*), its spine:shaft ratio remained constant throughout the dendrite and did not change in *Snx6^(-/-)* neurons (*Figure 8B*), indicating that in steady-state neurons, SNX6 is not involved in local trafficking of Homer1b/c from shaft to spines.

As a fraction of overexpressed fluorescently tagged Homer1c is cytosolic, next we tested the possibility that Homer1c enters spines via protein diffusion by fluorescence recovery after photobleaching (FRAP) assay on neurons expressing mEmerald-Homer1c. Consistent with previous findings on Homer1c turnover rates measured by FRAP using EGFP-Homer1c (*Kuriu et al., 2006*), following full synapse photobleaching of fluorescent signals, 7 ~ 9 % of recovery after 10 s was observed in both wild-type and *Snx6* KO neurons (8.09 ± 0.91% in *Snx6^(+/+)* and 8.14 ± 0.73% in *Snx6^(-/-)* neurons, *Figure 8C,D*), indicating that although the fraction of fast fluorescence recovery in spines attributable to entry of soluble cytosolic proteins (*Blanpied et al., 2008*; *Kerr and Blanpied, 2012*; *Kuriu et al., 2006*) is minor, diffusion of Homer1c protein molecules into spines does not require SNX6. After 10 min, about half of Homer1c fluorescence was recovered in both wild-type and *Snx6* KO neurons with similar recovery half-time (Recovery level $R_{final}$ = 54.55 ± 4.03%, $\tau_{1/2}$ = 108.4 ± 9.4 s in *Snx6^(+/+)* and $R_{final}$ = 52.63 ± 3.53%, $\tau_{1/2}$ = 107.2 ± 13.1 s in *Snx6^(-/-)*, *Figure 8C,D*). Further, similar results were obtained by fitting a single-exponential recovery curve to the average recovery time trace ($\tau_{1/2}$ = 117.6 ± 11.3 s, $R_{final}$ = 51.63 ± 0.86% in *Snx6^(+/+)* and $\tau_{1/2}$ = 108.9 ± 9.1 s, $R_{final}$ = 48.57 ± 0.73% in *Snx6^(-/-)*). These results are in good agreement with previous findings that there are immobile and mobile fractions of Homer1c in spines (*Kuriu et al., 2006*) and indicate that the dynamic turnover of Homer1c in spines is not affected by ablation of SNX6. Taken together, these data indicate that in steady-state neurons, neither SNX6 nor vesicular transport is required for recruitment of Homer1b/c from shaft to spines, and that synaptic Homer1b/c exchanges with the soluble protein pool that enters the spine by diffusion.

## Ablation of SNX6 causes impairment of AMPAR-mediated synaptic transmission and decrease in AMPAR surface expression

As ablation of SNX6 causes decrease in Homer1b/c distribution in distal dendrites, given the role of Homer1b/c in synaptic structure and function, next we sought to determine whether synaptic transmission is impaired in *Snx6^(-/-)* neurons by electrophysiological analysis. We eliminated the *Snx6* gene in a small subset of hippocampal neurons by injection of organotypic hippocampal slice culture from *Snx6^(fl/fl)* mouse with recombinant adeno-associated virus (AAV) coexpressing EGFP and the Cre recombinase (*Figure 9A* and *Figure 9—figure supplement 1*). By simultaneous recording the evoked EPSCs (eEPSCs) on infected and adjacent uninfected CA1 pyramidal neurons, we found that AMPAR-mediated eEPSCs were significantly impaired by about 50% with ablation of SNX6 (*Figure 9B*), whereas NMDAR-mediated eEPSCs and the pair-pulse ratio of AMPAR eEPSCs were

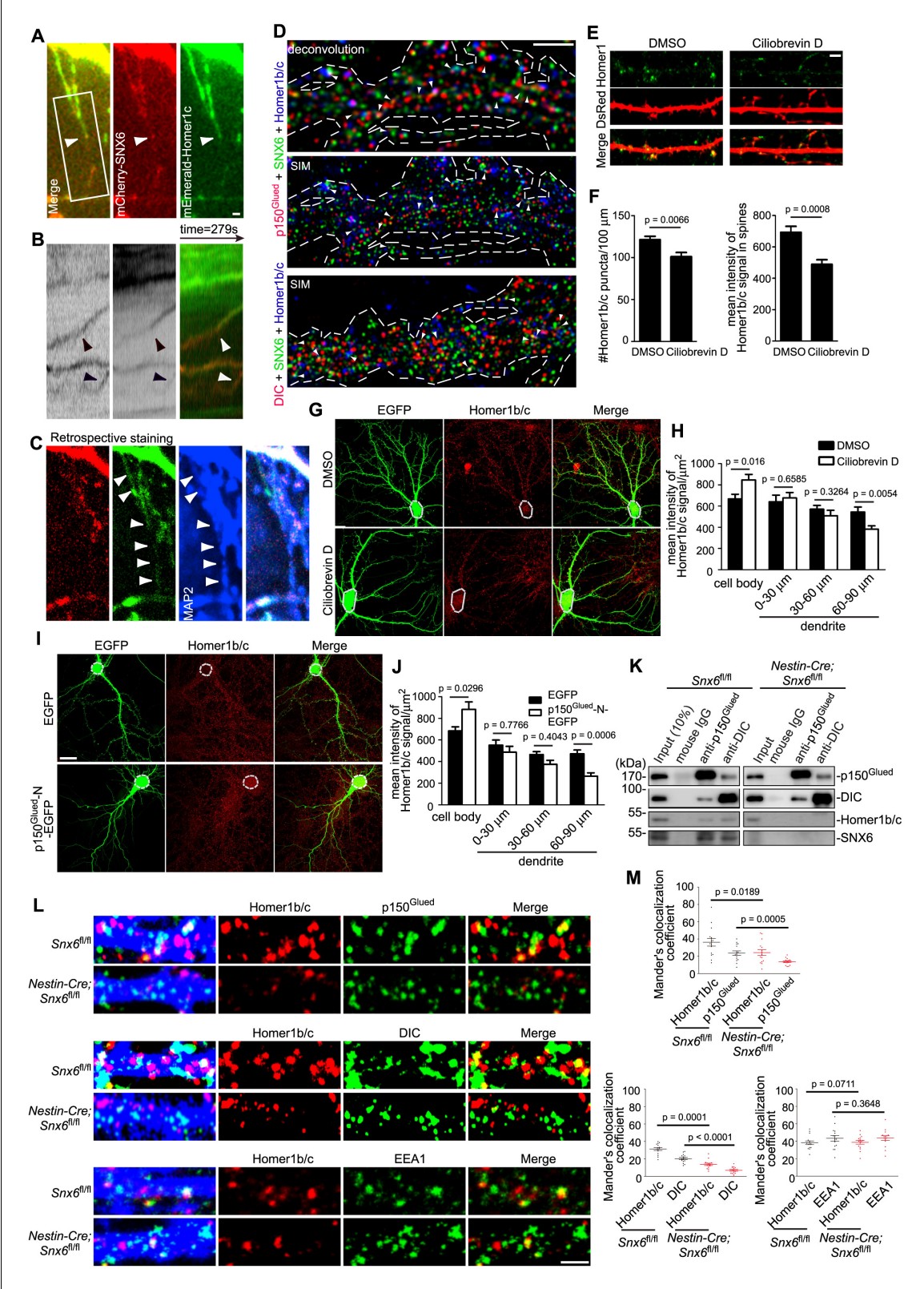

**Figure 6.** SNX6 is required for motility of Homer1b/c vesicles in dendritic shaft and their association with dynein–dynactin. (A–C) Dynamic behavior of mCherry-SNX6 and mEmerald-Homer1c in distal dendrite. The last frame of motile SNX6-, Homer1c-positive puncta (arrowhead) in dendrite (A) with the respective kymograph of boxed area (B) is shown. A retrospective staining of MAP2 after live imaging (C) illuminates dendrite identity. (D) Superresolution images of DIV18 neurons immunostained with antibodies to SNX6, Homer1b/c, and p150$^{Glued}$ or DIC. Shown are representative images

*Figure 6 continued on next page*

*Figure 6 continued*

of dendrites outlined with dashed lines. White arrowheads indicate overlapped signals. (**E**) DIV14 neurons were transfected with construct expressing EGFP, treated with DMSO or Ciliobrevin D on DIV16 for 2 hr and immunostained with antibodies to Homer1b/c. (**F**) Quantification of puncta number per 100 μm dendrite length and mean intensity in spines for Homer1b/c in (**E**) (mean ± SEM, n = 30, N = 3). (**G**) Same as (**E**), shown are representative confocal images of EGFP-expressing neurons. Dashed lines outline the cell bodies. (**H**) Quantification of Homer1b/c distribution in the cell body and dendrites in (**G**) (mean ± SEM, n = 30, N = 3). (**I**) DIV14 neurons were transfected with construct overexpressing EGFP or p150$^{Glued}$-N-EGFP, fixed on DIV16 and immunostained with antibodies to Homer1b/c. Dashed lines outline the cell bodies. (**J**) Quantification of Homer1b/c distribution in the cell body and dendrites in (**I**) (mean ± SEM, n = 30, N = 3). (**K**) Membrane fractions from mouse brain lysates were subjected to immunoisolation with antibodies to p150$^{Glued}$ or DIC coupled to Dynabeads Protein G, respectively. Shown are immunoblots probed with antibodies to SNX6, p150$^{Glued}$, DIC, and Homer1b/c. (**L**) Hippocampal neurons cultured from *Snx6$^{fl/fl}$* and *Nestin-Cre; Snx6$^{fl/fl}$* mice were transfected with pLL3.7.1 on DIV14, fixed on DIV17 and immunostained with antibodies to Homer1b/c and p150$^{Glued}$, DIC or EEA1. Shown are representative confocal images of dendritic segments. (**M**) Quantification of colocalization in (**L**) from 45 dendritic segments of 15 neurons (mean ± SEM, N = 2. Total length of dendrites: *Snx6$^{fl/fl}$*/ *Nestin-Cre; Snx6$^{fl/fl}$*: 1247 μm/1058 μm for p150$^{Glued}$; 1264 μm/1301 μm for DIC; 1244 μm/1291 μm for EEA1). Bars, 2 μm in (**A**), (**D**), (**E**) and (**L**), 20 μm in (**G**) and (**I**).

The following figure supplements are available for figure 6:

**Figure supplement 1.** Overexpressed mCherry-SNX6 and mEmerald-Homer1c recapitulate the distribution of endogenous proteins in hippocampal neurons.

**Figure supplement 2.** SNX6-, Homer1c-positive puncta move bidirectionally in the dendritic shaft.

unaffected (*Figure 9C,D*), indicating that the impairment of AMPAR eEPSCs is due to decrease in the number of AMPARs in the postsynaptic membrane but not reduction of presynaptic glutamate release. Indeed, no change in surface expression of the GluN2B subunit of NMDAR was detected in *Snx6$^{-/-}$* neurons by immunostaining and quantitative analysis (*Figure 9E,F*). In contrast, there was a decrease in the surface expression of GluA1 and GluA2, components of AMPAR, which was fully rescued by overexpressing SNX6 or Homer1c (*Figure 9G–J*).

Next we investigated mechanism(s) underlying reduced AMPAR surface expression in *Snx6$^{-/-}$* neurons. The longer isoforms of Homer inhibit not only trafficking to synaptic membrane but also constitutive activities of mGluR1/5 (*Ango et al., 2001*, *2002*; *Roche et al., 1999*). mGluR activation down-regulates surface expression of AMPAR by increasing its endocytosis rate (*Snyder et al., 2001*). Indeed, there was an increase in surface levels of mGluR5 in *Snx6$^{-/-}$* neurons (*Figure 9K*, *Figure 9—figure supplements 2* and *3*), which was restored by overexpression of SNX6 or Homer1c but not Homer1c-C (*Figure 9—figure supplement 2*). Moreover, ratiometric analysis of internalized and surface AMPAR detected an increase in the rate of AMPAR endocytosis in *Snx6$^{-/-}$* neurons (*Figure 9L,M*). To test the possibility that loss of Homer1b/c leads to decrease in surface AMPAR via increased surface levels and/ or constitutive activation of mGluRs, we treated neurons with inverse agonists for mGluR1/5 and determined surface levels of GluA1 and GluA2 by immunostaining. Quantitative analysis indicated that inhibition of mGluRs did rescue surface expression of AMPAR in *Snx6$^{-/-}$* neurons (*Figure 9N–Q*). Together, these data indicate that dendritic loss of Homer1b/c caused by ablation of SNX6 leads to increase in mGluR surface expression and mGluR-regulated AMPAR endocytic trafficking.

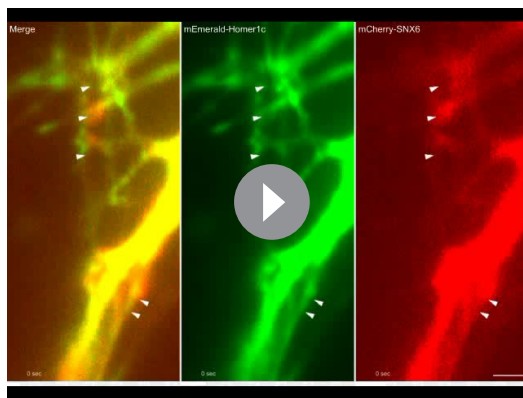

**Video 2.** Time-lapse live imaging showing movement of Homer1c-labeled and SNX6-labeled puncta in proximal dendrites. Hippocampal neurons co-transfected with Emerald-Homer1c and mCherry-SNX6 expressing constructs were imaged live by TIR-FM. The trajectories of two mobile Homer1c-, SNX6-positive puncta are indicated by white arrowheads. Images were acquired at 2 frames/s. Video plays at 50 frames/s. Bar: 2 μm.

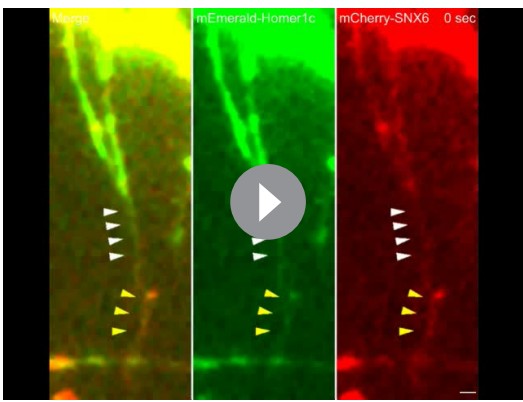

**Video 3.** Time-lapse live imaging showing movement of Homer1c-labeled and SNX6-labeled puncta in distal dendrites. Hippocampal neurons co-transfected with Emerald-Homer1c and mCherry-SNX6 expressing constructs were imaged live by TIR-FM. The trajectories of two mobile Homer1c-, SNX6-positive puncta are indicated by white and yellow arrowheads respectively. Images were acquired at 2 frames/s. Video plays at 50 frames/s. Bar: 2 µm.

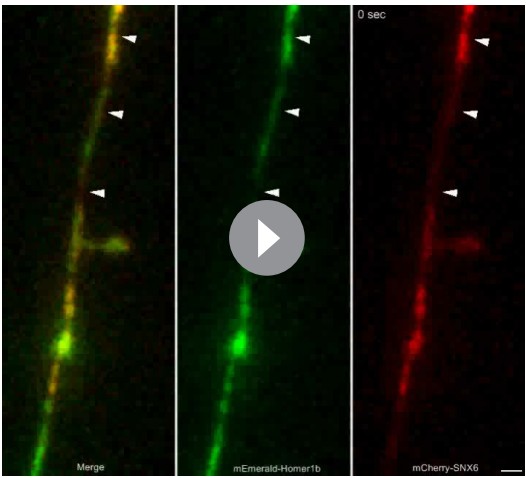

**Video 4.** Time-lapse live imaging showing movement of Homer1b-labeled and SNX6-labeled structures in a distal dendrite. Hippocampal neurons co-transfected with mEmerald-Homer1b and mCherry-SNX6 expressing constructs were imaged live by TIR-FM. The trajectory of a mobile Homer1b-, SNX6-positive structure is indicated by white arrowheads. Images were acquired at 2 frames/s. Video plays at 50 frames/s. Bar: 2 µm.

Another role of Homer1b/c in maintaining AMPAR surface levels is via the positioning of the endocytic zone (EZ) near the PSD (*Lu et al., 2007*). In the dendritic spine, an EZ adjacent to the PSD captures and retrieves AMPAR diffusing out of the synapse through local endocytic trafficking (*Blanpied et al., 2002*; *Lu et al., 2007*), which is crucial for the supply of a mobile pool of receptor molecules required for synaptic transmission and potentiation (*Lu et al., 2007*; *Petrini et al., 2009*). To determine whether ablation of SNX6 caused uncoupling of the EZ from the PSD, we performed co-immunostaining of PSD95 and clathrin heavy chain, a marker for the EZ. Indeed, ablation of SNX6 caused ~2 fold increase in the fraction of EZ (clathrin)-negative synapses (*Figure 9—figure supplement 4*), indicating that the coupling of the EZ to the PSD is impaired in *Snx6⁻/⁻* neurons, which might also contribute to lower surface levels of AMPAR in dendritic spines.

## Activity of the retromer core complex is not required for SNX6-regulated dendritic distribution of Homer1b/c

Previous studies have established a role for SNX6 as cargo adaptor in retromer-mediated vesicular transport (*Hong et al., 2009*; *Wassmer et al., 2009*). To test whether retromer is involved in SNX6-regulated dendritic distribution of Homer1b/c, we depleted VPS35, a core component of the retromer complex, by siRNA-mediated RNA interference in hippocampal neurons (*Figure 10A,B* and *Figure 10—figure supplement 1*). Immunofluorescence staining and

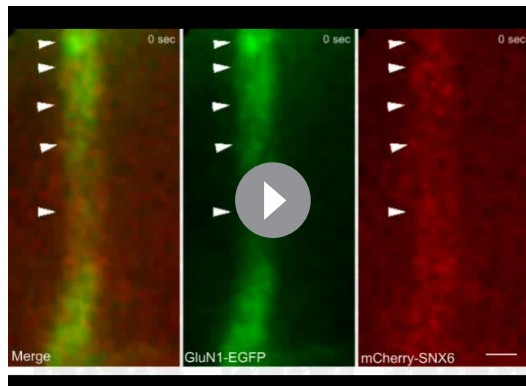

**Video 5.** Time-lapse live imaging showing movement of GluN1-labeled vesicles in dendrite. Hippocampal neurons co-transfected with mCherry-SNX6 and GluN1-EGFP expressing construct were imaged live by TIR-FM. The trajectory of the mobile GluN1-positive, SNX6-negative vesicle is indicated by white arrowheads. Images were acquired at 2 frames/s. Video plays at 50 frames/s. Bar: 1 µm.

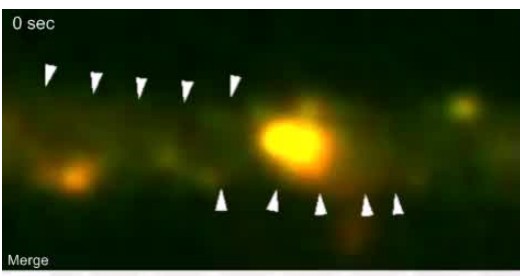

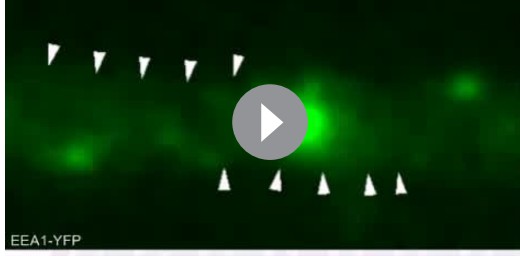

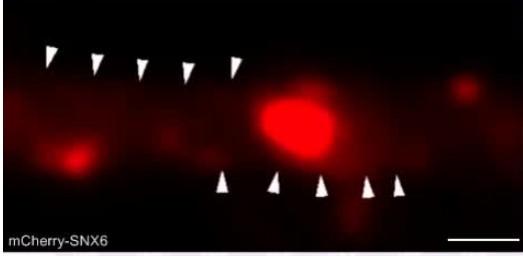

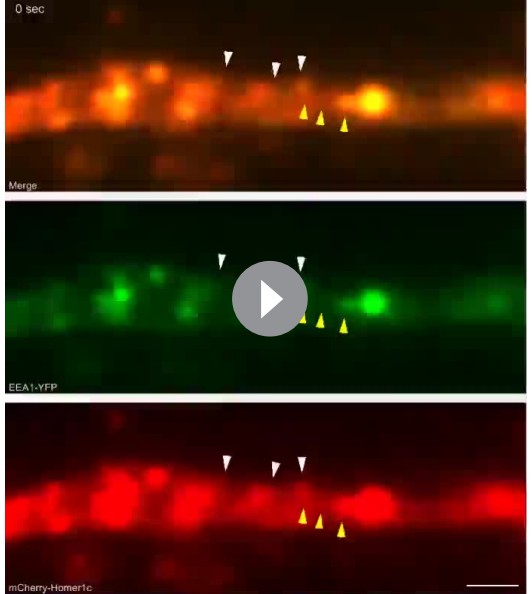

**Video 6.** Time-lapse live imaging showing movement of EEA1- and SNX6-labeled vesicles in dendrite. Hippocampal neurons co-transfected with EEA1-YFP and mCherry-SNX6 expressing constructs were imaged live by TIR-FM. The trajectories of two mobile EEA1-, SNX6-positive vesicles are indicated by white arrowheads. Images were acquired at 2 frames/s. Video plays at 50 frames/s. Bar: 1 μm.

**Video 7.** Time-lapse live imaging showing movement of EEA1- and Homer1c-labeled vesicles in dendrite. Hippocampal neurons co-transfected with EEA1-YFP and mCherry-Homer1c expressing constructs were imaged live by TIR-FM. Yellow arrowheads indicate the trajectory of an EEA1-, Homer1c-positive vesicle detaching and moving away from a large structure, suggesting fission and formation of transport carriers from early endosomes. White arrowheads indicate the trajectory of another vesicle moving in the dendritic shaft. Images were acquired at 2 frames/s. Video plays at 50 frames/s. Bar: 1 μm.

quantitative analysis of confocal images indicated that depletion of VPS35 in mature neurons caused a decrease not only in surface GluA1 but also in the number of Homer1b/c puncta in dendrites (*Figure 10C–E*), possibly resulted from reduced spine density as VPS35 has been found to promote spine formation and maturation (*Tian et al., 2015*; *Wang et al., 2012*). Nevertheless, compared with the phenotypes of *Snx6⁻/⁻* neurons, there was decrease in Homer1b/c signals throughout the cell body and dendrites, but no change in spine distribution of Homer1b/c or surface levels of mGluR5 when VPS35 was depleted (*Figure 10C and E–H*), indicating that it serves distinct function(s) from SNX6 in dendritic distribution of postsynaptic proteins. Moreover, live imaging of hippocampal neurons coexpressing mEmerald-Homer1c and VPS35-mCherry did not detect comovement of Homer1c- and VPS35-labeled vesicles in dendrite (*Figure 10I,J* and *Videos 15* and *16*). Further, retrospective staining of MAP2 and VPS35 right after live imaging confirmed the dendrite identity and absence of endogenous VPS35 at the base of the spine where a Homer1c-positive structure stopped (*Figure 10J*, rightmost panels). To further confirm that SNX6 does not cooperate with the retromer to regulate motility of Homer1b/c vesicles, we performed coimmunostaining of Homer1b/c and VPS35 on *Snx6⁺/⁺* and *Snx6⁻/⁻* neurons. Quantitative analysis showed that colocalization between Homer1b/c and VPS35 was very little compared with that between Homer1b/c and SNX6 or EEA1. Moreover, it was not affected by ablation of SNX6 (*Figure 10—figure supplement 2A,B*). Conversely, no change in colocalization between Homer1b/c and SNX6 was detected in VPS35-depleted

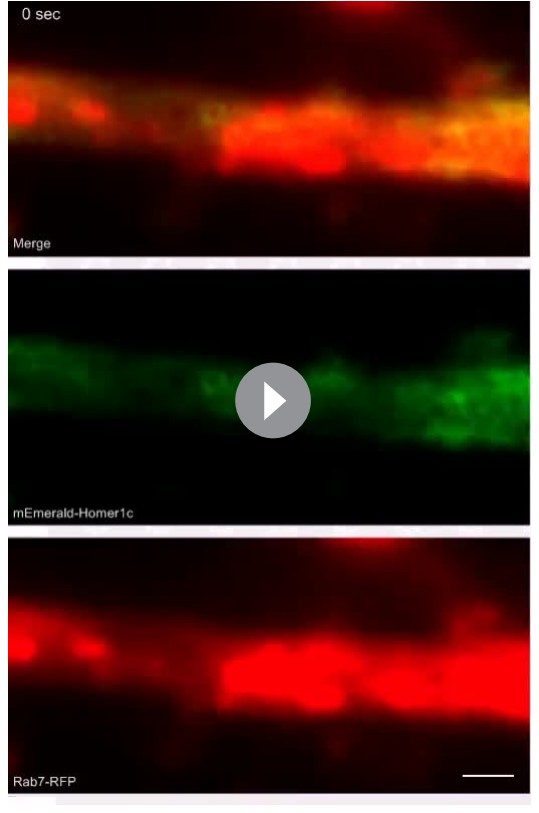

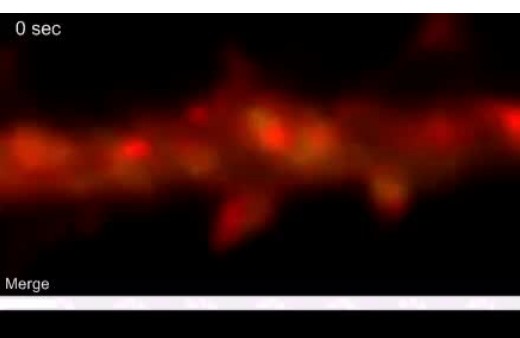

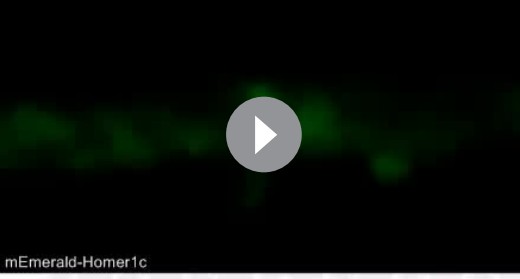

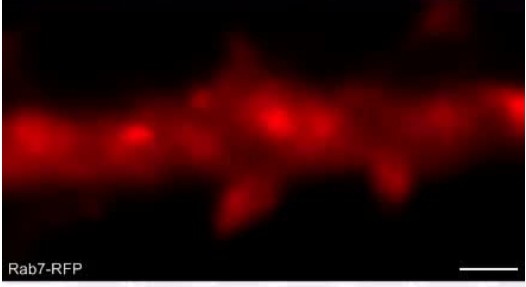

**Video 8.** Time-lapse live imaging showing movement of Rab7- and Homer1c-labeled structures in the dendrite of a wild-type neuron. *Snx6*<sup>+/+</sup> hippocampal neurons co-transfected with Rab7-RFP and mEmerald-Homer1c expressing constructs were imaged live by TIR-FM. Images were acquired at 2 frames/s. Video plays at 50 frames/s. Bar: 1 µm.

**Video 9.** Time-lapse live imaging showing movement of Rab7- and Homer1c-labeled structures in the dendrite of a *Snx6* KO neuron. *Snx6*<sup>-/-</sup> hippocampal neurons co-transfected with Rab7-RFP and mEmerald-Homer1c expressing constructs were imaged live by TIR-FM. Images were acquired at 2 frames/s. Video plays at 50 frames/s. Bar: 1 µm.

neurons either (*Figure 10—figure supplement 2C,D*). Taken together, these data indicate that SNX6 functions independent of retromer to regulate dendritic distribution of Homer1b/c.

## Discussion

In this study, we demonstrate that ablation of SNX6 in the CNS causes deficits in spatial learning and memory, a hippocampal-dependent brain function. At the cellular level, loss of SNX6 causes a decrease in spine density in the distal apical dendrites of CA1 hippocampal cells and impairment of their AMPAR-mediated synaptic transmission, indicating that SNX6 is required for synaptic structure and function of these excitatory neurons. We also show that SNX6 directly interacts with Homer1b/c, a PSD scaffolding protein crucial for the structural and functional integrity of dendritic spines, and that there is decrease in Homer1b/c distribution in distal dendrites in *Snx6*<sup>-/-</sup> neurons. Moreover, the spine density and surface AMPAR level phenotypes of *Snx6*<sup>-/-</sup> neurons could be rescued by overexpressing Homer1b/c or SNX6. These findings reveal an important physiological function of SNX6 in the CNS excitatory neurons.

The long isoforms of Homer1 are implicated in learning and memory. *Homer1* knockout mice exhibit deficits in spatial learning and a decrease in LTP in the CA1 region (*Gerstein et al., 2012*; *Jaubert et al., 2007*), which could be rescued by AAV-mediated expression of Homer1c in the

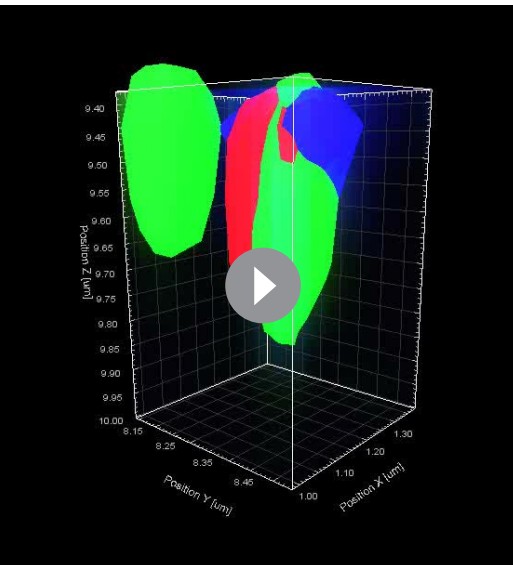

**Video 10.** 3D-SIM movie of an enlarged region of interest from a hippocampal neuron dendrite shows association of p150<sup>Glued</sup> (red), SNX6 (green) and Homer1b/c (blue).

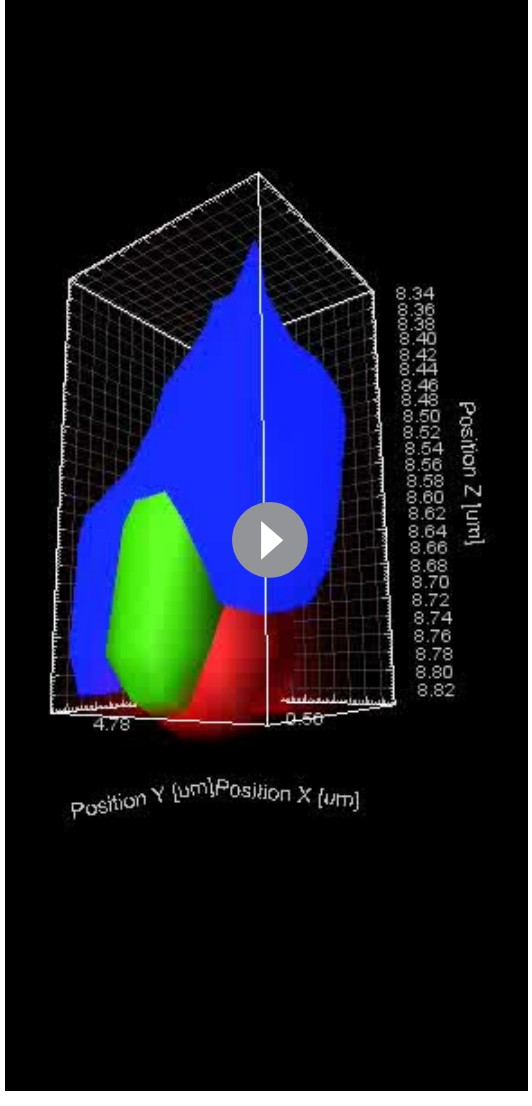

**Video 11.** 3D-SIM movie of an enlarged region of interest from a hippocampal neuron dendrite shows the association of DIC (red), SNX6 (green) and Homer1b/c (blue).

hippocampus (*Gerstein et al., 2012*). Although Homer1b/c is widely expressed in the brain, in the hippocampus it is predominantly distributed to the CA1 region (*Shiraishi et al., 2004*), which partially explains why ablation of SNX6 function causes a decrease in spine density of CA1 distal apical dendrites and impairs hippocampal-dependent spatial learning and memory that mainly involves the Schaffer collateral-CA1 synapses. Whether there are other SNX6-interacting proteins that are also required for CA1 neuron function awaits further investigation. As spatial learning and memory involve not only the hippocampus but also other cortical areas such as the entorhinal cortex and the medial prefrontal cortex (*Jo et al., 2007*; *Nagahara et al., 1995*; *Nakazawa et al., 2004*; *Steffenach et al., 2005*; *Zhou et al., 1998*), it also remains to be determined whether and how ablation of SNX6 affects the synaptic structure and function of neurons in other parts of the cortex.

PSD scaffolding proteins interact, anchor and stabilize glutamate receptors. Change in their protein content in dendritic shaft and spines influences synaptic transmission through receptor localization and distribution at synaptic sites. Among the PSD scaffolding proteins, PSD95 has been intensively studied. Dendritic trafficking and synaptic targeting of PSD95 requires a C-terminal tyrosine-based signal, palmitoylation of the N-terminus and its interaction with both Myosin V and dynein through binding of GKAP to these molecular motors (*Craven and Bredt, 2000*; *El-Husseini et al., 2000*; *Hruska et al., 2015*; *Naisbitt et al., 2000*). Synaptic localization of Shank also requires GKAP (*Naisbitt et al., 1999*; *Sala et al., 2001*), whilst recruitment of Homer1b to dendritic spines requires synaptically targeted Shank (*Sala et al., 2001*). We found that the motility of Homer1c-associated vesicles in dendritic shaft requires SNX6, and that ablation of SNX6 or inhibition of dynein-dynactin activity causes reduction in the amount of Homer1b/c in distal dendrites. Previously

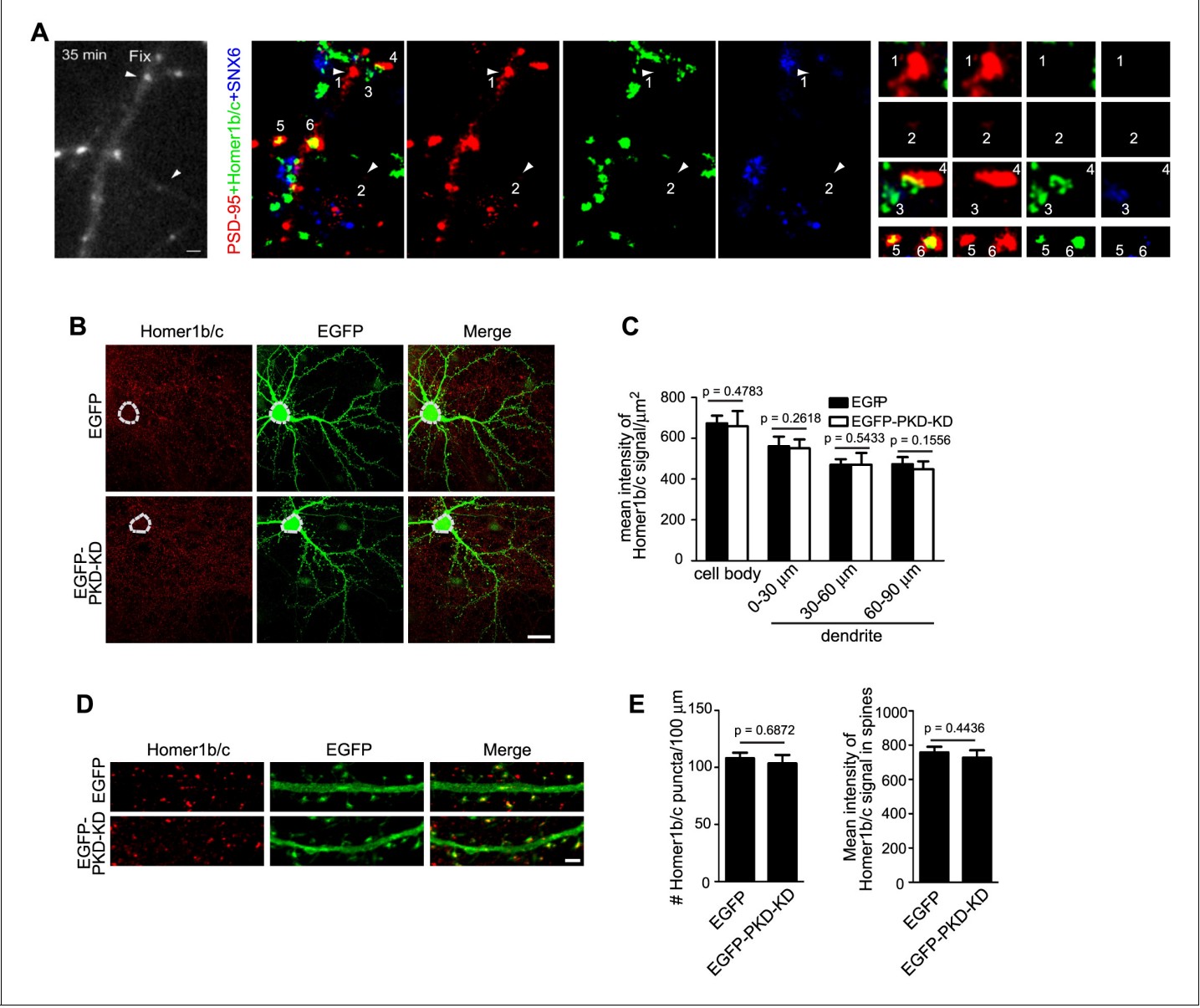

**Figure 7.** The dendritic trafficking pathway of Homer1b/c is distinct from the PSD95 and secretory trafficking pathways. (A) TIR-FM of hippocampal neurons transfected with PSD95-RFP expressing construct. Left panel: still image of the last frame of time lapse imaging. Arrowheads mark the final positions of two motile puncta. Center panels: confocal images of retrospective staining of endogenous SNX6 and Homer1b/c after live imaging. Right panels: enlargement of arrowhead-indicated, numbered puncta in the center panels. 1 and 2: motile PSD95 puncta lacking both SNX6 and Homer1b/c; 3: vesicle containing endogenous SNX6 and Homer1b/c, but not PSD95; 4: an immobile PSD95 punctum that contacts with SNX6 signal. 5: an immobile PSD95 punctum that colocalizes with Homer1b/c. 6: an immobile PSD95 punctum that colocalizes with both Homer1b/c and SNX6. (B) DIV14 hippocampal neurons were transfected with construct overexpressing PKD-KD and immunostained with antibodies to Homer1b/c on DIV16. Shown are representative confocal images. (C) Quantification of the mean intensity of Homer1b/c signals in the cell body and dendrites in (B) (mean ± SEM, n = 30, N = 3). (D) The effect of PKD-KD overexpression on the distribution of Homer1b/c in dendrites and spines. (E) Quantification of Homer1b/c distribution in (D) (mean ± SEM, n = 30, N = 3). Bars: 1 μm in (A), 20 μm in (B), 2 μm in (D).

imaging assays and quantitative modeling have established that dynein-driven bidirectional transport contributes to the polarized targeting of dendrite-specific cargo (*Kapitein et al., 2010*). Therefore, lack of dynein–dynactin-driven transport in the dendritic shaft provides a possible mechanism for the Homer1b/c distribution phenotype in $Snx6^{-/-}$ neurons. However, since the majority of Homer1c puncta are immobile in dendrites of steady-state neurons, and little is known about the cellular

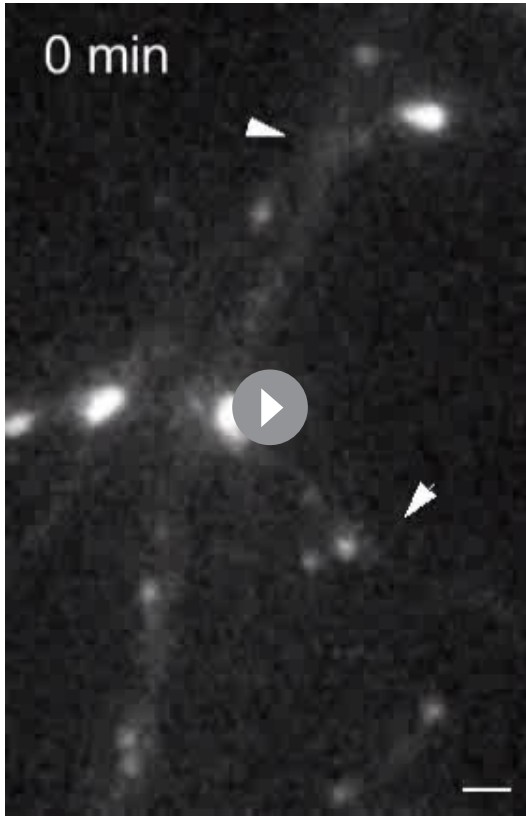

**Video 12.** Time-lapse live imaging showing movement of PSD95-RFP-labeled puncta in dendrites. Hippocampal neurons transfected with PSD95-RFP expressing construct were imaged live by TIR-FM. White arrowheads indicate two mobile puncta followed by fixation. Images were acquired at 2 frames/s. Video plays at 50 frames/s. Bar: 1 μm.

functions of SNX6 apart from its role as dynein cargo adaptor, it is also possible that SNX6 regulates the distribution of Homer1b/c in dendrites via mechanism(s) distinct from dynein–dynactin-driven transport. Moreover, ablation of SNX6 does not cause complete loss of Homer1b/c from distal dendrites, indicating that mechanism(s) other than SNX6-mediated transport contributes to its localization to dendritic shaft far from the cell body. Since disruption of the secretory pathway does not affect Homer1b/c localization to dendritic shaft and spines, alternative mechanisms for its distribution in dendrites include diffusion of free protein molecules, cotransport with protein(s) other than the PSD95-GKAP-Shank complex or vesicular transport mediated by different motor(s) and/or adaptor(s).

Notably, in $Snx6^{-/-}$ neurons, although there was a decrease in the amount of Homer1b/c in both shaft and spines of distal dendrites (*Figure 5F*), the spine:shaft ratio of its signals remained constant throughout the dendrite (*Figure 8B*), indicating that once in dendrite, Homer1b/c could enter the spines via SNX6-independent mechanism(s). In dendrites, several mechanisms exist for the transfer of postsynaptic components from shaft to synaptic sites in spines, including cytosolic diffusion, exocytosis of transmembrane proteins at the plasma membrane and their lateral diffusion to synaptic sites, and active transport by molecular motors. The AMPARs enter dendritic spines via both lateral diffusion and actin-based, Myosin V-driven transport of recycling endosomes (*Adesnik et al., 2005*; *Correia et al., 2008*; *Makino and Malinow, 2009*; *Wang et al., 2008*;

*Yudowski et al., 2007*). Since Homer1b/c is a scaffolding protein, its entry into spines might rely on diffusion of free molecules, possibly released from endosomal carriers or from the cytosolic pool in the shaft, or transport of the vesicular cargo directly from the shaft by a different motor. Our results from live imaging, FRAP and quantitative analyses show that direct spine entry of Homer1c puncta is an extremely rare event, and that the dynamic turnover of Homer1c in spines is not affected by ablation of SNX6. Collectively, these data indicate that in steady-state neurons, Homer1b/c enters spines by cytosolic diffusion, and SNX6 is not required for its spine localization.

In retromer-mediated vesicular transport of transmembrane proteins, although the VPS26–VPS29–VPS35 core complex of retromer has been shown to interact with cargo proteins in the endosomal membrane (*Arighi et al., 2004*; *Fjorback et al., 2012*; *Nothwehr et al., 2000*) and was hence termed the cargo selection complex (CSC), the SNX subunits of the retromer also contribute to cargo recognition (*Harterink et al., 2011*; *Strochlic et al., 2007*; *Temkin et al., 2011*; *Voos and Stevens, 1998*; *Zhang et al., 2011*). Our study not only shows that SNX6 directly interacts with Homer1b/c, but also provides evidence that SNX6-regulated distribution of Homer1b/c in dendritic shaft is retromer-independent. The CSC is involved in retrieval of APP from endosomes to the TGN and plasma membrane delivery and endocytic recycling of the AMPAR in dendritic spines (*Choy et al., 2014*; *Fjorback et al., 2012*; *Munsie et al., 2015*). SNX27, another member of the SNX family and a component of retromer (*Temkin et al., 2011*), interacts with both NMDAR and AMPAR and regulates their recycling to the cell surface (*Cai et al., 2011*; *Hussain et al., 2014*; *Loo et al., 2014*). It remains to be determined whether or not SNX6 functions in retromer-mediated

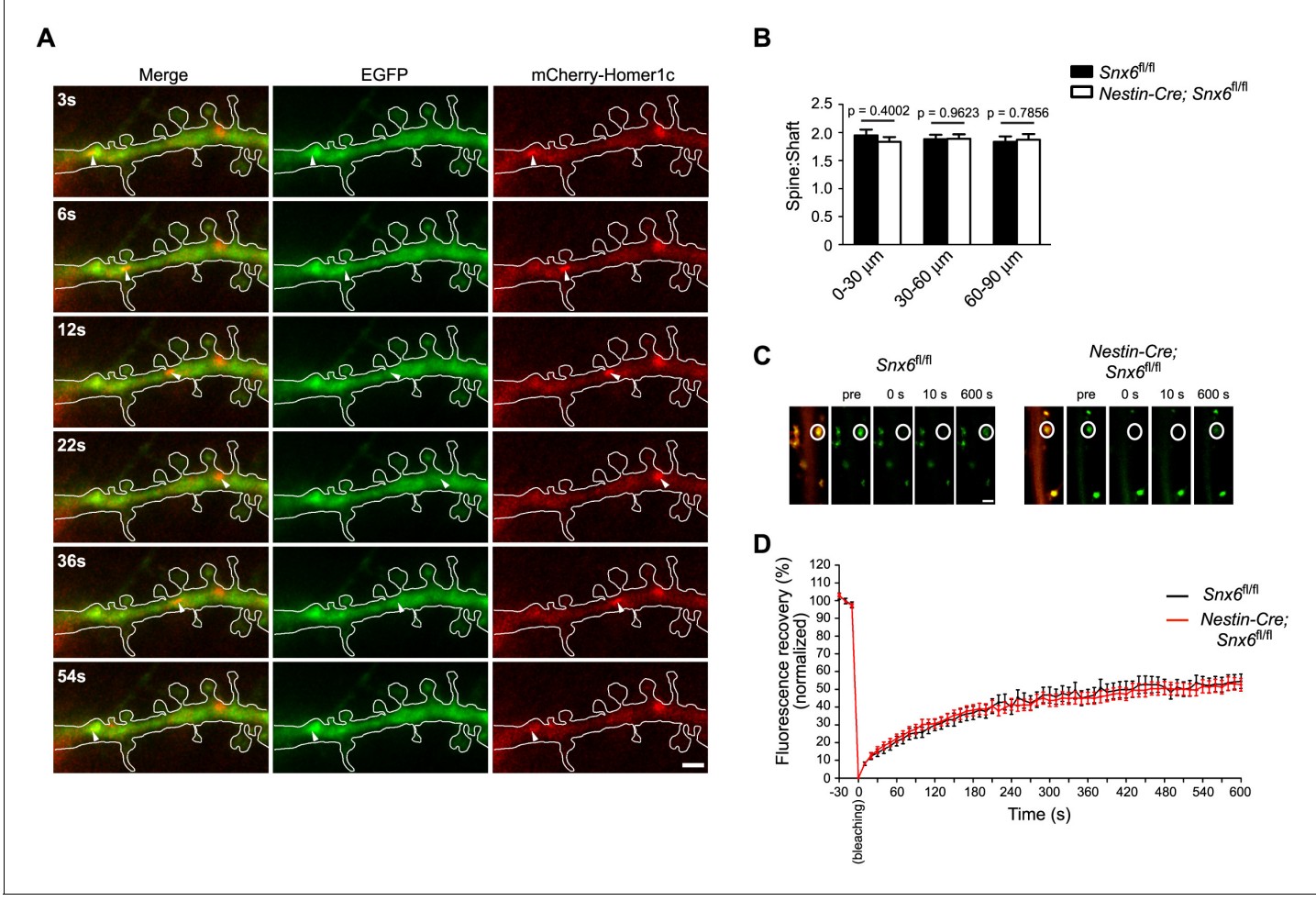

**Figure 8.** Homer1b/c enters spines by SNX6-independent protein diffusion. (**A**) Representative images from a time-lapse video (*Video 13*) of a wild-type neuron co-expressing EGFP and mCherry-Homer1c. White solid lines indicate outline of the dendritic shaft and spines. White arrowheads indicate positions of a Homer1c-labeled structure moving in the shaft. Bar: 2 μm. (**B**) Spine:shaft ratios of Homer1b/c fluorescence intensity over distance from the cell body. DIV14 neurons from *Snx6*<sup>fl/fl</sup> and *Nestin-Cre; Snx6*<sup>fl/fl</sup> mice were transfected with construct overexpressing EGFP as volume marker, fixed on DIV16 and immunostained with antibodies to Homer1b/c. Shown are the ratios of the mean intensity of Homer1b/c in spines to that in the corresponding shaft (mean ± SEM, 50 spines from 15 neurons/group, N = 2 independent experiments). (**C**) FRAP analysis of mEmerald-Homer1c in dendritic spines. Hippocampal neurons were co-transfected with constructs expressing mEmerald-Homer1c and DsRed on DIV13. FRAP analysis was performed on DIV16. Shown are examples of the fluorescence intensity of mEmerald-Homer1c before, immediately after, 10 s and 600 s after photobleaching of the spines indicated with white circles. Bar: 1 μm. (**D**) Averaged fluorescence recovery curves after photobleaching for mEmerald-Homer1c in spines of *Snx6*<sup>+/+</sup> (31 spines, 10 cells) and *Snx6*<sup>-/-</sup> (32 spines, 10 cells) neurons. Data represent mean ± SEM.

dendritic transport. Notably, SNX6 shares the highest sequence similarity (85%) with SNX32, another SNX with unknown function. The mild neurodevelopmental phenotype exhibited by *Nestin-Cre; Snx6*<sup>fl/fl</sup> mice suggests functional redundancy among the SNX family members. Further studies are needed to characterize the roles of evolutionarily conserved SNXs, including SNX6 and SNX32, in sorting and trafficking of neuronal proteins and their functions in synaptic development and activity.

## Materials and methods

### Animals

All experiments were performed in compliance with the guidelines of the Animal Care and Use Committee of the Institute of Genetics and Developmental Biology, Chinese Academy of Sciences. The *Nestin-Cre* transgenic C57BL/6 J mice were obtained from the Nanjing Biomedical Institution of

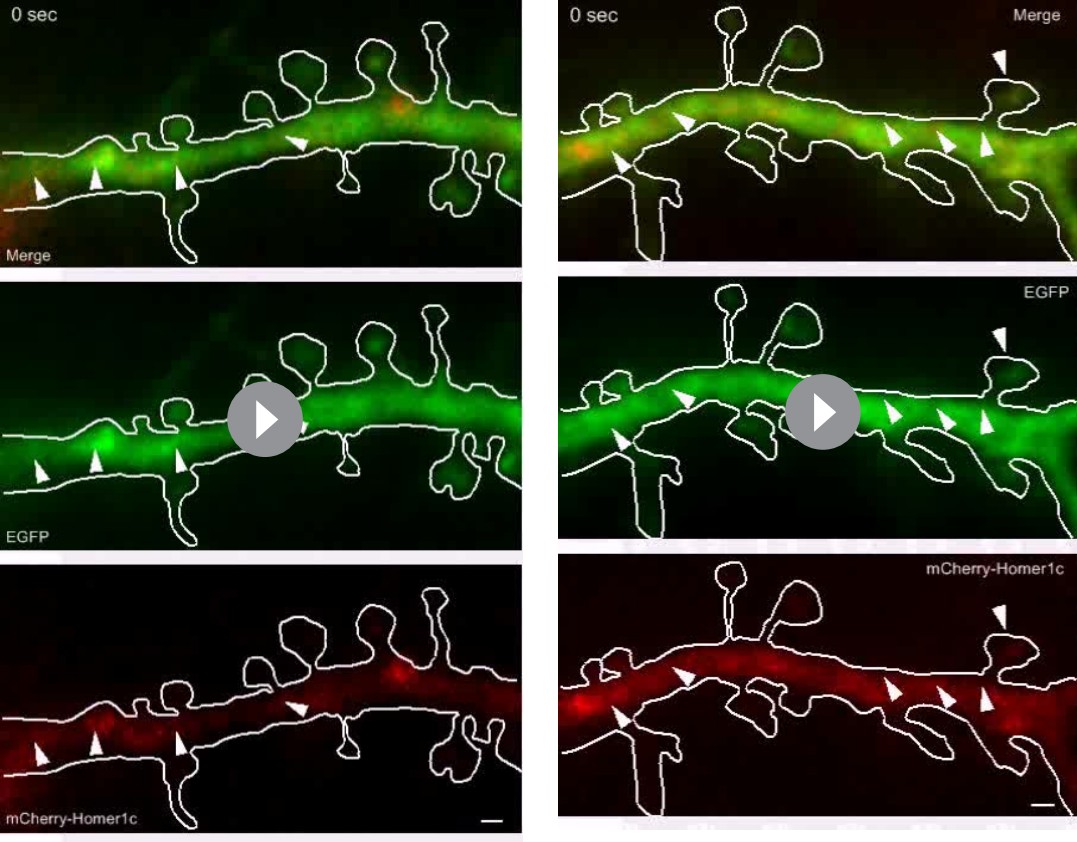

**Video 13.** A motile Homer1c-labeled structure in dendritic shaft did not enter spines. *Snx6*[+/+] hippocampal neurons co-transfected with EGFP and mCherry-Homer1c expressing constructs were imaged live by TIR-FM. Images were acquired at 2 frames/s. Video plays at 10 frames/s. White solid lines indicate outline of the shaft and spines. White arrowheads indicate the trajectory of a Homer1c-labeled structure moving in the dendritic shaft. Bar: 1 µm.

**Video 14.** A motile Homer1c-labeled structure in dendritic shaft entered a spine. *Snx6*[+/+] hippocampal neurons co-transfected with EGFP and mCherry-Homer1c expressing constructs were imaged live by TIR-FM. Images were acquired at 2 frames/s. Video plays at 10 frames/s. White solid lines indicate outline of the shaft and spines. White arrowheads indicate the trajectory of a Homer1c-labeled structure entering a spine. Bar: 1 µm.

Nanjing University (*Tronche et al., 1999*). The *Thy1-EGFP* transgenic mice were obtained from the Jackson Laboratory (*Feng et al., 2000*). The *Snx6*[fl/fl] mice were generated at the Nanjing Biomedical Institution of Nanjing University. SNX6 CNS-specific knockouts were generated by crossing *Snx6*[fl/fl] mice with *Nestin-Cre* mice.

## Generation of *Snx6* CNS-specific knockout mice

A 7.1 kb genomic fragment of *Snx6* containing exon 5, 6 and 7 was cloned from a bacterial artificial chromosome (BAC) clone (Clone name: RP23-422H22, BACPAC Resource Center, Oakland, CA, USA) to construct targeting vector. The first *lox*P site was inserted 248 bp upstream of exon 5, and a 2.1 kb FRT-neo-*lox*P-FRT cassette with the second *lox*P site was inserted 424 bp downstream of exon 5. The targeting vector was linearized with *Not*I and electroporated into C57BL/6NTac embryonic stem (ES) cells. ES cells were selected in G418- and ganciclovir-containing medium as described previously (*Liu et al., 2003*). Genomic DNA of 96 drug-resistant colonies were isolated, digested with *Xho*I or *Hpa*I and analyzed by southern blotting. Two independent ES cell clones (9 hr and 5D) were chosen to inject into C57BL/6 J blastocysts to obtain chimeric mouse. CNS-specific *Snx6* knockout mice were obtained by crossing *Snx6*[fl/fl] with *Nestin-Cre* mice. The *Snx6* frameshift mutation was generated by deleting exon five via Cre-loxp mediated site-specific recombination. Deletion of exon

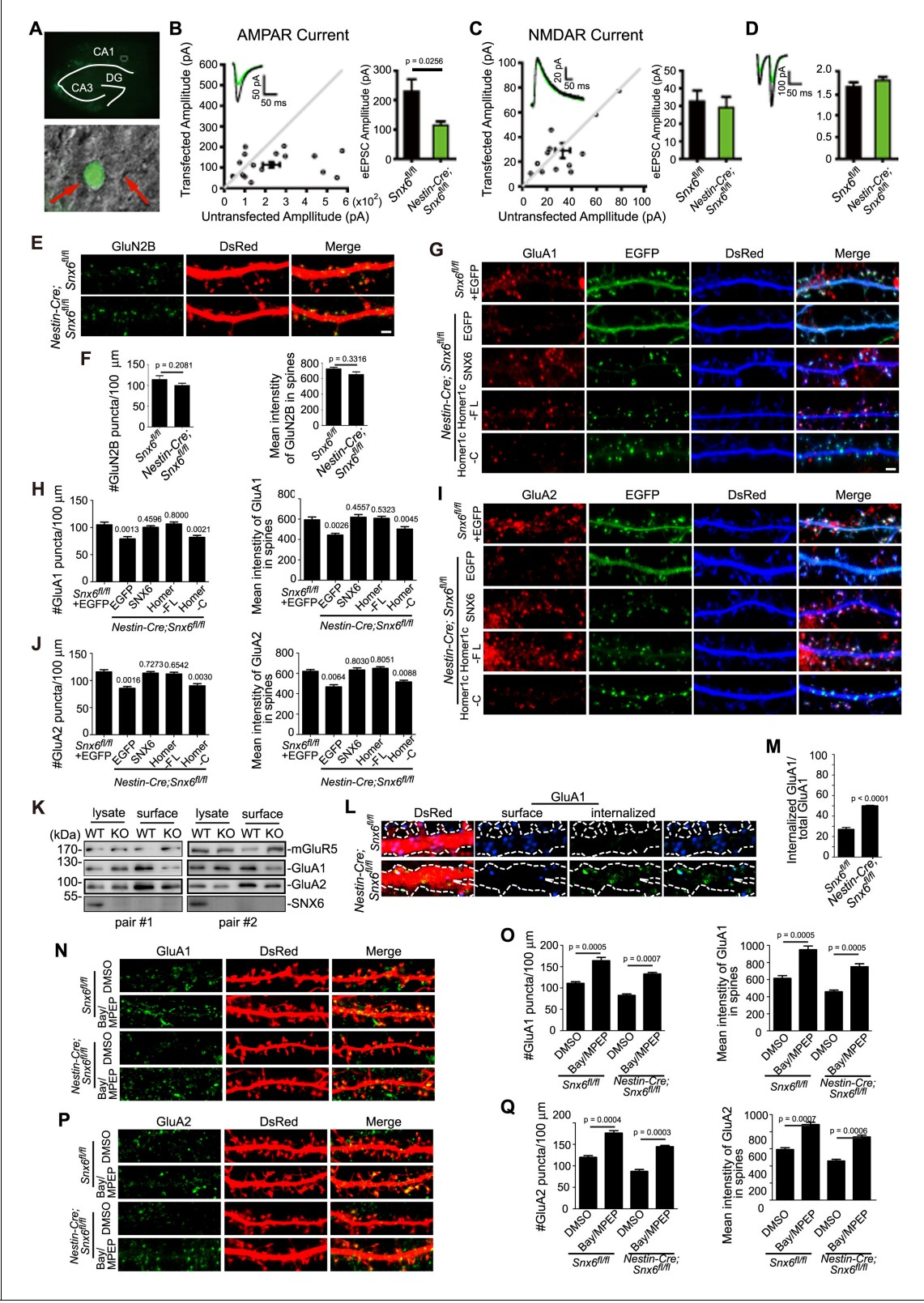

**Figure 9.** Impaired synaptic transmission and decreased surface AMPAR levels of *Snx6*[−/−] neurons. (**A**) Hippocampal slice culture from *Snx6*[fl/fl] mouse was partially infected with AAV-EGFP-2A-Cre. Lower panel shows a CA1 neuron infected with AAV-EGFP-2A-Cre and an adjacent control neuron (indicated with red arrows) that were chosen to be recorded simultaneously. (**B**) Dual recording analysis of AMPAR-mediated synaptic responses (n = 18 pairs). Scatterplots show amplitudes of AMPAR eEPSCs (absolute values) for single pairs of neurons (open circles) and mean ± SEM (filled circle)

*Figure 9 continued on next page*

*Figure 9 continued*

across all neuron pairs collected. The current amplitudes of infected neurons were plotted on the ordinate and the current amplitudes of the control neurons were plotted on the abscissa. Inset shows sample current traces from an infected (green) and a control (black) neurons. Bar graph shows mean ± SEM of AMPAR amplitudes represented in the scatterplots. (C) NMDAR-mediated eEPSCs (n = 12 pairs). (D) Paired-pulse recording of AMPAR eEPSCs (n = 11 pairs). Two identical stimulus pulses were delivered in an interval of 50 ms and AMPAR eEPSCs were recorded at −70 mV. Left are sample traces of eEPSCs from a pair of infected (green) and control neurons. The paired-pulse ratio (PPR) was the enhancement of the second eEPSC relative to the first eEPSC. Bar graph shows mean ± SEM of PPRs. (E) Hippocampal neurons transfected with pLL3.7.1 were fixed on DIV18 and immunostained with antibodies to surface GluN2B. (F) Quantification of puncta number per 100 μm dendrite length and fluorescence mean intensity in spines for surface GluN2B (mean ± SEM, n = 30, N = 3). (G) DIV13 neurons were co-transfected with constructs expressing DsRed and EGFP, EGFP-SNX6, mEmerald-Homer1c-FL or EGFP-Homer1c-C, fixed on DIV18 and immunostained with antibodies to surface GluA1. (H) Quantification of signal intensity and spine distribution of surface GluA1 (mean ± SEM, n = 32, N = 3). (I) Same as (G), except that neurons were immunostained with antibodies to surface GluA2. (J) Quantification of surface GluA2 (mean ± SEM, n = 33, N = 3). (K) Hippocampal neurons from *Snx6*^fl/fl (WT) and *Nestin-Cre; Snx6*^fl/fl (KO) littermates were cultured till DIV16. Surface levels of mGluR5, GluA1 and GluA2 were then measured by cleavable surface biotinylation followed by immunoblotting with antibodies to mGluR5, GluA1 and GluA2. SNX6 serves as a negative control for surface proteins. Shown are immunoblots from two pairs of *Snx6*^fl/fl and *Nestin-Cre; Snx6*^fl/fl littermates. (L) Antibody uptake assay was performed on neurons transfected with pLL3.7.1. Shown are representative images of dendrites immunostained for internalized and surface AMPAR signals. (M) Quantification of AMPAR endocytosis rate (mean ± SEM, 45 dendritic segments, n = 15, N = 3). (N) DIV14 neurons were treated with inverse agonists for mGluRs for 48 hr, fixed and immunostained with antibodies to surface GluA1. (O) Quantification of surface GluA1 (mean ± SEM, n = 30, N = 3). (P) Same as (N), except that neurons were immunostained with antibodies to surface GluA2. (Q) Quantification of surface GluA2 (mean ± SEM, n = 30, N = 3). Bar: 2 μm.

The following figure supplements are available for figure 9:

**Figure supplement 1.** Cre-mediated knockout of SNX6 in cultured neurons.

**Figure supplement 2.** Ablation of SNX6 causes an increase in surface expression of mGluR5 in hippocampal neurons, which is rescued by SNX6 or Homer1c.

**Figure supplement 3.** Homer1a overexpression increases surface mGluR5 expression.

**Figure supplement 4.** Ablation of SNX6 causes a decrease in the number of clathrin positive synapses.

five resulted in premature stop in mRNA translation at nt 270. Genotyping of mouse lines was performed by genomic PCR. PCR genotyping of tail prep DNA from offspring was performed with the following primer pairs:

FRTtF/loxtR:

```
5'-TTTGGCATTATCAAAACGTTGTTGTA-3'
5'-GAGATGCTCAGCACACTTTCTCTAC-3'
```

(PCR primer locations are shown in *Figure 1B* resulting in a PCR product of 295 base pairs in *Nestin-Cre; Snx6*^fl/fl mice and none in *Snx6*^fl/fl mice).

loxtF/loxtR:

```
5'-AGAGTCACTATCAGAGCCTCTTCAG-3'
5'-GAGATGCTCAGCACACTTTCTCTAC-3'
```

(PCR primer locations are shown in *Figure 1B* resulting in a PCR product of 366 base pairs in *Snx6*^fl/fl mice and none in *Nestin-Cre; Snx6*^fl/fl mice).

The *Nestin-Cre* transgene was detected using the following primer pairs:

```
5'-TGCCACGACCAAGTGACAGCAATG-3'
5'-ACCAGAGAGACGGAAATCCATCGCTC-3'
```

Mice with a sparsely distributed population of GFP-expressing neurons for analysis of single-cell morphology were generated by crossing *Nestin-Cre; Snx6*^fl/fl mice to *Thy1-EGFP* transgenic mice. *Thy1-EGFP* transgene was detected using the following primer pairs:

```
5'-TCTGAGTGGCAAAGGACCTTAGG-3'
5'-CGCTGAACTTGTGGCCGTTTACG-3'
```

## Constructs and viruses

The mEmerald-Homer1c construct was a generous gift from Michael Davidson (Addgene plasmid # 54120, Addgene, Cambridge, MA). The PSD-95-pTagRFP was a generous gift from Johannes Hell

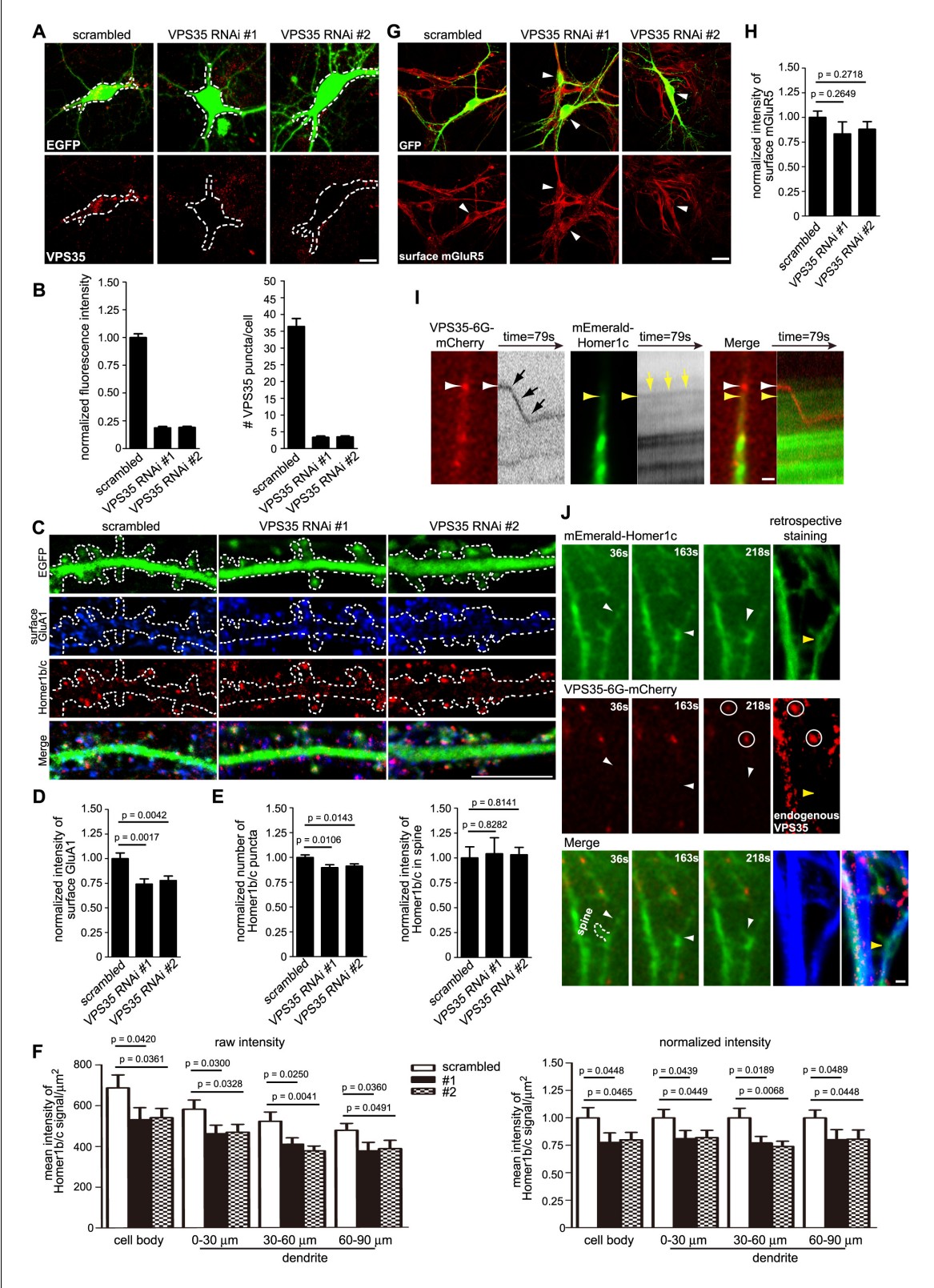

**Figure 10.** The retromer core complex is not required for SNX6-regulated dendritic distribution of Homer1b/c. (**A**) Hippocampal neurons were transfected with lentiviral vector expressing siRNA along with GFP at DIV12, fixed on DIV18 and immunostained with antibodies to VPS35. (**B**) Quantification of VPS35 signal intensity and puncta number in neurons in (**A**) (mean ± SEM, n = 10, N = 3). (**C**) Neurons transfected with siRNA constructs were immunostained with antibodies to surface GluA1 and Homer1b/c. (**D–E**) Quantification of surface GluA1 (**D**) (mean ± SEM, scrambled: *Figure 10 continued on next page*

*Figure 10 continued*

33 neurons; VPS35 RNAi #1: 30 neurons; VPS35 RNAi #2: 30 neurons, N = 3.) or Homer1b/c distribution in spines (**E**) (mean ± SEM, scrambled: 35 neurons; VPS35 RNAi #1: 32 neurons; VPS35 RNAi #2: 30 neurons. N = 3 ). (**F**) Quantification of Homer1b/c distribution in the cell body and dendrites of hippocampal neurons expressing scrambled or VPS35-targeting siRNA (mean ± SEM, n = 15, N = 3). The results show a decrease in signal intensity of Homer1b/c throughout the cell when VPS35 was depleted. (**G**) Same as (**C**), except that neurons were immunostained with antibodies to surface mGluR5. (**H**) Quantification of surface mGluR5 in (**G**) (mean ± SEM, scrambled: 35 neurons; VPS35 RNAi #1: 44 neurons; VPS35 RNAi #2: 52 neurons. N = 3 ). (**I–J**) TIR-FM of hippocampal neurons transfected with Homer1c and VPS35-expressing constructs. Shown in (**I**) is a VPS35-positive vesicle (white arrow) moving away from the cell body and bypassing a static Homer1c-positive structure (yellow arrow) with their respective kymographs to the right. Shown in (**J**) are still images of representative time points: a Homer1c-positive structure reached the base of spine and part of which entered the spine after a brief lag. White arrowheads indicate the mobile Homer1c structure. A retrospective staining of MAP2 and VPS35 after live imaging (right panels) illuminates the dendrite identity and the absence of endogenous VPS35 at the base of the spine. Yellow arrowhead indicates the position of the Homer1c-positive structure right before fixation. White circles indicate VPS35 puncta appearing in both retrospective staining and live imaging. Bars: 5 µm in (**A**) and (**C**), 20 µm in (**G**) and 1 µm in (**I**) and (**J**).

The following figure supplements are available for figure 10:

**Figure supplement 1.** VPS35 is efficiently knocked down in HEK293 cells.

**Figure supplement 2.** Colocalization of Homer1b/c with VPS35 does not require SNX6 and vice versa.

(Addgene plasmid # 52671). The FUmGW construct expressing membrane bound GFP (mGFP) was a generous gift from Connie Cepko (Addgene plasmid # 22479). The mCherry-Homer1C construct was generated by cloning full-length Homer1c amplified from mEmerald-Homer1c into pmCherry-C2 (Clontech Laboratories, Inc., Mountain View, CA). The mEmerald-Homer1b construct was derived from mEmerald-Homer1c by deletion of its aa 177–188. EGFP-Homer1C-N/C constructs were generated by cloning Homer1c N-terminus (aa1-187)/C-terminus (aa188-366) amplified from mEmerald-Homer1c into pEGFP-C2 (Clontech Laboratories, Inc.). His-sumo fusion of full-length Homer1c, Homer1c-N (aa1-111), Homer1c-M1 (aa111-177), Homer1c-M2 (aa111-187) and Homer1c-C (aa188-366) were generated by cloning fragments amplified from mEmerald-Homer1C into pET28a-sumo. His-sumo-Homer2b (NCBI Accession: AF093260.1) and Homer3 (NCBI Accession: NM_001146153.1) constructs were generated by PCR amplification of full-length Homer2b and Homer3 from mouse brain cDNA and cloning into pET28a-sumo. The GluN1-EGFP (NR1-EGFP) construct was a generous gift from A. Kimberley McAllister (Department of Neurology, University of California at Davis, USA). The EEA1-YFP construct was a generous gift from Li Yu (Tsinghua University, China). The Rab7-RFP construct was a generous gift from Hong Tang (Institute of Biophysics, Chinese Academy of Sciences, China). The pEGFP-PKD-K618N construct was a generous gift from Vivek Malhotra (Center for Genomic Regulation, Spain). pEGFP-N3-p150$^{Glued}$-N (aa 1–1060) and constructs expressing full-length SNX1, SNX2, SNX5, and SNX6 fused with EGFP, mCherry or FLAG tag were described previously (*Hong et al., 2009*). To obtain expression constructs for His-SNX1-N, His-SNX6-N, GST-SNX1-N and GST-SNX6-N, N termini of SNX1 (aa 1–271) and SNX6 (aa 1–181) were PCR-amplified from the full-length constructs and subcloned into pET28a and pGEX-4T-1, respectively. To avoid disrupting the subunit interactions of VPS35-VPS29-VPS26 (*Munsie et al., 2015*), VPS35-6G-EGFP was constructed by PCR amplifying human VPS35

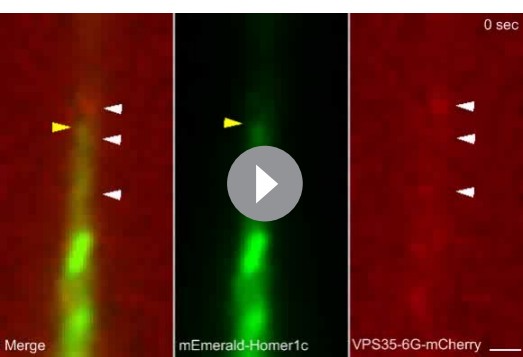

**Video 15.** Time-lapse live imaging showing movement of a VPS35 vesicle in dendrite. Hippocampal neurons co-transfected with mEmerald-Homer1c and VPS35-6G-mCherry expressing constructs were imaged live by TIR-FM. White arrowheads indicate the trajectory of a VPS35-positive vesicle moving away from the cell body and bypassing a static Homer1c-positive structure (indicated with yellow arrowhead). Images were acquired at 2 frames/s. Video plays at 50 frames/s. Bar: 1 µm.

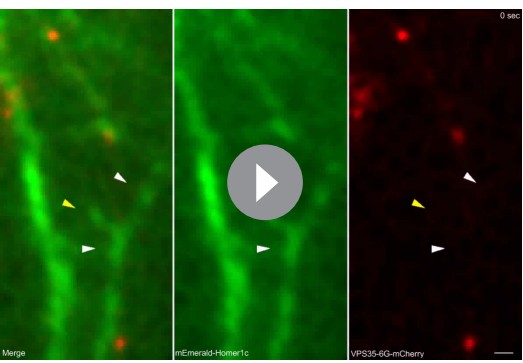

**Video 16.** Time-lapse live imaging showing movement of Homer1c and VPS35 puncta in dendrites. Hippocampal neurons co-transfected with mEmerald-Homer1c and VPS35-6G-mCherry expressing construct were imaged live by TIR-FM. White arrowheads indicate the initial and pausal sites of a Homer1c-positive, VPS35-negative structure. Yellow arrowhead indicates the spine position. Images were acquired at 2 frames/s. Video plays at 50 frames/s. Bar: 1 μm.

coding sequence from HeLa cDNA with a pair of primers in which nucleotides encoding six glycines were added to the reverse primer and inserting into pCMV-EGFPN3.To construct VPS35-6G-mCherry, EGFP sequence of VPS35-6G-EGFP was replaced by mCherry coding sequence from pCMV-mCherryC1. Viral particles of adeno-associated virus (AAV) carrying pAOV-CaMKIIα-EGFP-2A-Cre and the control construct pAOV-CaMKIIα-EGFP-2A-3FLAG were purchased from Obio Technology (Shanghai) Corp. Ltd., (Shanghai, China).

## Antibodies

Antibodies used in this study are: mouse anti-GluA1 (MAB2263), mouse anti-GluA2 (MAB397), mouse anti-Tau1 (MAB3420), and mouse anti-MAP2 (MAB3418) from Millipore (Billerica, MA); mouse anti-SYP (D-4) (sc-17750), mouse anti-PSD95 (6G6) (sc-32291), mouse anti-DIC (74-1) (sc-13524), goat anti-Homer1a (M-13) (sc-8922), rabbit anti-Homer1b/c (H-174) (sc-20807), mouse anti-SNX6 (D-5) (sc-365965), goat anti-SNX6 (N-19) (sc-8679), and rabbit anti-Rab5B (A-20) (sc-598) from Santa Cruz Biotechnology (Santa Cruz, CA); mouse anti-p150$^{Glued}$ (610474), mouse anti-EEA1 (610457) from BD Biosciences (San Diego, CA); rabbit anti-clathrin (ab21679), rabbit anti-TGN46 (ab50595), rat anti-CTIP2 (ab18465), rabbit anti-SNX1 (ab134126) and rabbit anti-Rab4 (ab109009) from Abcam; rabbit anti-GluN1 (D65B7) from Cell Signaling Technology (Mississauga, ON, Canada); rabbit anti-GluN2B (AGC-003) and rabbit anti-mGluR5 (AGC-007) from Alomone labs (Jerusalem, Israel); rabbit and mouse anti-GFP (MBL598, D153-3), rabbit and mouse anti-RFP (PM005, M155-3) from Medical and Biological Laboratories (Naka-kuNagoya, Japan); mouse anti-β-actin (A5441) (Sigma-Aldrich, St. Louis, MO); mouse anti-Golgi97 (A21270) from Invitrogen (Carlsbad, CA); goat anti-VPS35 (PAB7499) from Abnova (Taipei, Taiwan) and rabbit anti-GluN2A (612–401-D89) from Rockland Immunochemicals (Limerick, PA); rabbit anti-Rab7 (#9367) from Cell Signaling Technology (Mississauga, ON, Canada); rabbit anti-Rab11 (3H18L, 700184) from Life Technologies (Carlsbad, CA); Rabbit anti-SNX6 was described previously (*Hong et al., 2009*).

## Histology and immunohistochemistry

For SNX6 immunohistochemistry, 10 weeks-old mice were anesthetized and pre-fixed by perfusion with 4% paraformaldehyde (pH 7.4, Sigma-Aldrich) transcardially. Brains were removed, post-fixed overnight at 4°C followed by dehydration with gradient sucrose (30%, 40%, 50%), and sectioned at 25 μm on a LEIGA CM 1950 vibratome (Leica Biosystems, Germany). Braine slices were pasted on glass slides coated with gelatin/chromium potassium sulfate solution (gelatin, 3 g, and chromium potassium sulfate, 0.05 g from Sigma-Aldrich were dissolved in 200 ml sterile water) and antigen retrieval was performed in water bath with Tri-Sodium citrate (10 mM, pH 6.0, dihydrate, Sigma-Aldrich) at above 90°C for 20 min. Immunohistochemistry was performed following instructions from the manufacturer of polinker-2 plus polymer HRP detection system (GBI, Bothell, WA). Briefly, After permeabilization with 1% TritonX-100 for 15 min followed by an endogenous peroxidase activity quenching in 3% hydrogen peroxide, sections were rinsed with PBS, blocked in 1% BSA plus 5% normal goat serum, and incubated with anti-SNX6 primary antibody (Santa Cruz Biotechnology, diluted in PBS containing 1% BSA, 1% normal goat serum) overnight at room temperature (RT). Sections were then rinsed and incubated with polymer helper for 15 min at RT, poly-HRP anti-goat IgG for 20 min at 37°C followed by DAB color development, hematoxylin dyeing for the nucleus, and xylene clearing. Samples were analyzed using a Nikon ECLIPSE TE2000-U microscope.

## Behavioral analyses

All mice used for behavior analysis were 10 weeks old male with normal eating and movement in cages by eye observation. One day before test, mice were transferred into the room installed with test platform.

## Open field test

The mouse was fed in a separate cage one day before the test and was gently placed in an open-field test chamber and allowed to freely explore for 10 min. The locomotor activity (total distance traveled in the whole chamber) and the emotionality (the percentage of distance and time spent in the center area) were monitored and analyzed by an automated system (the Anilab System, AniLab Software and Instruments Co., Ltd, Ningbo, China).

## Rotarod

On the first training day, the mouse was placed on the rotating rod with straight line acceleration of 9.9 rpm/s from minimal (10 rpm) to maximal speed (30 rpm) (the acceleration process takes about 10 min) followed by fixed speed at 30 rpm for another 5 min. On the second day, the mouse was placed on the rotating rod with fixed speed (30 rpm), the motor function and coordination were determined by the latency to fall off the rod.

## Elevated plus maze

The mouse was placed on a platform consisting of four perpendicularly intersected arms (two open arms without walls and two arms enclosed by walls) 50 cm high from the ground and allowed to freely explore for 6 min after a 4 min pre-adaptation by allowing the mouse to move freely in an open chamber with high walls placed on the ground after leaving the feeding cage. The ration of retention time staying at and numbers entering the open arms to closed arms were monitored and analyzed by an automated system (the Anilab System, AniLab Software and Instruments Co., Ltd).

## Tail suspension test

The mouse was suspended by its tail above 50 cm high from the ground for 5 min. Due to innate aversion to this tail-up situation, the mouse will struggle until immobile after multiple failures. A camera was installed as closely as possible in order to obtain the highest possible resolution of the animals. The immobile time was quantified and used to evaluate the depression condition.

## Forced swimming test

On the first day, the mouse was gently and slowly placed into a round tank (height: 27 cm, diameter: 18.5 cm) of which two-thirds was filled with water to, typically, avoid the animal's head from being submerged under the water for 90 s for pre-adaptation. On the second day, the mouse was placed in the same round tank for 5 min. The immobile time was quantified and used to evaluate the depression condition.

## Three-Chamber test

Subject mouse was placed into the middle chamber and habituated for 5 min. Then, the wall between the chambers was removed to allow the mouse access freely to explore the three chambers with two empty wire cup-like cup housing in both left and right chambers for the first 10 min. The duration of subject mouse stretching into a 5 cm circular area around the cup is monitored as an active contact within 10 min. Then 'stranger 1' mouse was placed into cup housing in the left chamber for a second 10 min. For a third 10 min, 'stranger 2' mouse was placed into cup housing in the right chamber.

## Repetitive behavior test

The mouse was observed for a 10 min period in 20 cm × 30 cm quadrate cage with bedding. The duration of each mouse spending in the following behaviors was measured: cage-lid flipping/jumping, rearing, grooming and digging.

## Y maze spontaneous alternation

Testing was performed in a Y-shaped maze consisting of three radial arms at a 120° angle. The mouse was put in the center of the maze and allowed to explore freely its three arms for 6 min. Typically, the mouse prefers to explore a new arm rather than the one that was just visited. The percentage of alternation (the number of trials/the number of arm entries) is calculated for evaluating the spatial working memory.

## Shuttle box

The mouse was placed in a two-compartment shuttle box for about 5 ~ 10 min to make it quiet. A trial constituted of 5 s tone on followed by 5 s footshock (0.39 ~ 0.4 mA) was given to make mouse build the association between tone and footshock. The mouse can avoid to receive the shock by escaping to the opposite compartment during tone on (active escape), or can receive a foot shock shorter than 5 s during shock on after tone off (passive escape). Each day 25 trials were performed with 15 s interval for five days. The time latency of active or passive escape was monitored and analyzed by an automated system (the Anilab System, AniLab Software and Instruments Co., Ltd). The average time latency of 25 trials each day was used to evaluate conditioned memory.

## Morris water maze

The Morris water maze procedure consists of hidden platform acquisition training, probe trial testing and recall training. The water tank is a 120 cm diameter circular pool with a circular goal platform submerged 0.5 cm below the water surface. Water temperature was about 22°C to 24°C. Cues with different shapes and colors were pasted on the wall of the tank above the water surface in four different directions. A circular black curtain around the tank eliminated competing environmental cues. The mouse trajectory in the pool was monitored and analyzed by an automated system (Smart 3.0, Panlab SMART video tracking system, Barcelona, Spain). The day before the experiment, the mouse was gently placed into the pool without a platform to freely swim for 90 s for pre-adaptation. Acquisition training was then performed for eight days and four trials per day with different water-entering the site (at north, south, east, and west positions adjacent to the pool wall). During each trial, mouse must learn to use cues to navigate a path to the hidden platform within 90 s. If failed to find the platform, mouse will be led to the platform with a stick, and kept on it for 10 s. The escape latency (the average value of the time duration from entering the water to finding the platform of four trials per day) was calculated for each mouse. After acquisition training, the hidden platform was removed and probe testing was performed for five days and one trial each day at the distal water-entering site away from the platform. A 1.5x platform circle area where the platform was placed was monitored. The latency to first enter 1.5x area (time duration from entering the water to first enter the 1.5x area) and numbers of crossing 1.5x platform circle area of each mouse within 90 s were analyzed. For recall training, after a 20 day-rest, mouse was placed in the same pool without a platform to examine memory extinguishment. Similarly, the latency to first enter 1.5x area and numbers of crossing 1.5x area of each mouse within 90 s were analyzed. Afterward, the platform was placed back to the pool and recall training was performed for one day with four trials with different water-entering sites. The second day, the platform was removed again, and the mouse was placed in the pool at the farthest water-entering site away from the platform. The latency to first enter 1.5x area and numbers of crossing 1.5x area of each mouse was analyzed.

## Spine imaging, three-dimensional reconstructions and measurement of spine density in brain slices

The 100 µm-thick brain slices from $Snx6^{fl/fl}$; $Thy1$-GFP and $Nestin$-Cre; $Snx6^{fl/fl}$; $Thy1$-GFP mice were prepared as described in **Histology and immunohistochemistry** and mounted onto slides. For quantitative analysis of the morphology and density of dendrite spines, only those dendrite segments located in similar branches of the dendritic tree (oriens/distal or radiatum/thin branches for CA1 basal/apical dendrites (*Megías et al., 2001*), and secondary and tertiary branches in stratum oriens or stratum radiatum for CA3 basal/apical dendrites (*Baker et al., 2011*) from GFP-expressing and relatively isolated dorsal hippocampal CA1 or CA3 pyramidal neurons were selected for imaging by z-stack sectioning with a 0.40 µm interval using a Nikon confocal microscopy (EZ-C1, 100x oil, 4x optical zoom) with the same acquisition parameters. Three-dimensional reconstructions were

performed and spine density was quantified using the Filament module of IMARIS software as described previously (*Shen et al., 2008*). The parameters were: the minimums pine head diameter (thinnest diameter) was $\geq$ 0.1 μm, the ratio of branch length to trunk radius was $\geq$ 1.5 μm, and the branch length $\geq$ 0.5 μm. The spine numbers each segment was further verified by manual counting. Spine density was defined as spine numbers/3D-segment length.

## Transmission electron microscopy (TEM)

Tissue preparation and electron microscopy were conducted as described (*Chen et al., 2008*) with slight modification. Briefly, eight-week-old mice (*Snx6*$^{fl/fl}$ and *Nestin-Cre;Snx6*$^{fl/fl}$) were anesthetized with 2% pentobarbital sodium and perfusion-fixed with cold phosphate buffer (PB, 0.1 M, pH 7.4) containing 2.5% glutaraldehyde (Electron Microscopy Sciences, Hatfield, PA) and 1% PFA (Electron Microscopy Sciences). Following removal of mouse brain, the hippocampal CA1 and CA3 regions were sliced into 1 mm-thickness sections transverse to its longitudinal axis. Sections were fixed in the same fixative overnight at 4℃, rinsed with PB and postfixed with 1% osmium tetroxide ($OsO_4$) for 1 hr at 4℃ in dark. Sections were then rinsed with distilled water and dehydrated in an ascending series of acetone (50%, 70%, 80%, 90% and 100%, 15 min per dilution). Samples were embedded in Embed 812 (Electron Microscopy Sciences) and polymerized at 60℃ for 48 hr. Ultra-thin sections (60 nm) were mounted on carbon-coated copper grids, stained with 2% uranyl acetate for 15 min, rinsed with distilled water and stained with lead citrate for 5 min, rinsed with distilled water and air dried. Sections were imaged on a JEM-1400 electron microscope (JEOL) at 80 kV. Electron micrographs were captured with a Gatan 832-CCD (4 k x 3.7 k pixels, Gatan Inc., Pleasanton, CA) at 30,000$\times$ magnification. All image analysis was conducted blind to the genotype.

## Primary culture of mouse hippocampal neurons

Hippocampi from C57BL/6 J mouse embryos (E17.5) were removed and trypsinized (0.125% trypsin, 15 min at 37℃). Dissociated cells were suspended in Dulbecco's modified Eagle medium (DMEM, Hyclone, Logan, UT) supplemented with 10% horse serum and 10% F12, then plated on coverslips pre-coated with poly-D-lysine (100 μg/ml, Sigma-Aldrich) in 24-well plates at a density of 2000 ~ 5000 cells/well. Four hours later, the medium was replaced with neuronal culture medium (Neurobasal medium, 2% B27, 1% Glutamax). Half of the media were changed every three days until use.

## Primary neuron transfection and siRNA

Neurons were transfected at DIV12 ~14 using Lipofectamine 2000 (Invitrogen) or Lipofectamine LTX (Invitrogen, Carlsbad, CA) according to the manufacturer's recommendations. For VPS35 knockdown, Lentiviral vectors coexpressing VPS35-targeting siRNA and EGFP were purchased from Applied Biological Materials Inc., USA. Target sequences for VPS35: siRNA #1: 5'-GGTGTAAATG TGGAACGTTACAAACAGAT-3'; siRNA #2: 5'-AGCTGTTATGTGCTTAGTAATGTTCTGGA-3'. Scrambled (non-targeting) siRNA: 5'-GGGTGAACTCACGTCAGAA-3'. Neurons were fixed for immunostaining analysis six days after siRNA transfection.

## Immunofluorescence staining, confocal image acquisition and analysis

The procedures for immunofluorescence staining and confocal microscopy were performed essentially as previously described (*Hong et al., 2009*). Briefly, hippocampal neurons were fixed at DIV 16 ~ 18 with 4% paraformaldehyde supplied with 4% sucrose in phosphate-buffered saline (PBS) for 10 min at RT, then permeabilized and blocked in PBS containing 1% BSA, 0.4% Triton X-100 for 15 min at RT. Then primary and secondary antibodies conjugated with Alexa Fluor 488/555/647 were used for detection. For goat anti-SNX6 and rabbit anti-Rab5B (Santa Cruz Biotechnology), antigen retrieval was performed as described in **Histology and Immunohistochemistry** for optimal staining. Surface staining of GluA1 and mGluR5 was performed as previously described by *Peebles et al. (2010)*. Surface staining of GluN2B and GluA2 was performed as previously described by *Swanger et al. (2013)*. For digitonin extraction of cytosolic proteins before immunostaining, neurons were rinsed with KHM buffer (20 mM HEPES (pH 7.4), 110 mM $CH_3CO_2K$, and 2 mM Mg $(CH3COO)_2$) and then treated with KHM buffer containing 25 μg/ml digitonin for 5 min on ice. Neurons were then rinsed once with KHM buffer and fixed with 4% paraformaldehyde (PFA) in PBS for 10 min at RT, blocked with PBS containing 5 % BSA and 0.2% Triton X-100 for 15 min followed by

overnight incubation with primary antibody. Images were acquired by confocal microscopy (EZ-C1, 100x oil, 4x optical zoom) and analyzed with the NIS-Elements AR3.1 software. Some confocal images (*Figure 5A,C,E,G*; *Figure 6E,G and I*; *Figure 7B,D*; *Figure 9E,G,I,N,P*; *Figure 5—figure supplement 2A*; *Figure 9—figure supplement 2*; *Figure 9—figure supplement 3A*; *Figure 9—figure supplement 4*) were collected using z-stack with a 0.40 μm interval and analyzed with ImageJ. Confocal imaging after applying 31 × 31 median followed by Costes' auto-threshold subtraction was done to quantify colocalization (Mander's colocalization coefficient (MCC), values are %; as previously described [*Dunn et al., 2011*]). Control and experimental group neurons which were to be directly compared were imaged with the same acquisition parameters. To reduce variability, only segments of the secondary and tertiary dendrites (distance from the cell body: 30–120 μm; length: 30–40 μm /segment) were imaged. Ten to fifteen neurons each group in each independent experiment, and 90–120 μm dendrites per neuron were analyzed. Two-dimensional, background-subtracted, maximum projection reconstructions of Z-stack images were used for morphologic analysis and quantification. To examine the size, number, and fluorescence intensity of signal puncta in shaft and spines, the EGFP- or DsRed-labeled dendrites or spines were outlined manually. Numbers of puncta were measured manually and the size of Homer1 puncta as well as the mean intensity of fluorescent signals in individual spines was measured using the ImageJ function 'Analyze > measure'. Quantification of spine:shaft ratios of Homer1b/c was conducted as described (*Smith et al., 2014*). Briefly, the dendritic shaft values of Homer1b/c signals was calculated as the mean fluorescence intensity of three regions of shaft along the dendritic region within 1–2 μm of analyzed spines, and used with corresponding spine values to produce spine:shaft ratios. All image analyses were conducted blind to the genotype. For quantitative analysis of VPS35 knockdown in neurons, the fluorescence intensity or VPS35 puncta from EGFP-expressing neurons in each group was normalized by that of untransfected neurons in the same group. The normalized values in each VPS35 siRNA group were further presented as relative values to the scrambled siRNA group.

## Protein identification with mass spectrometry

For identification of SNX6-interacting proteins, mouse brain was homogenized with lysis buffer (20 mM Tris.HCl, 10 mM HEPES, pH 7.4, 150 mM NaCl, 1 mM EDTA, 20 mM imidazole, 1% Triton) supplemented with protease inhibitors. Lysates were centrifuged at 12,000 × g for 20 min. The supernatants were incubated with recombinant His-tagged SNX1-N (aa 1–271) or SNX6-N (aa 1–181) immobilized on Ni-NTA agarose overnight at 4°C. Beads were rinsed with wash buffer (150 mM NaCl, 20 mM Tris.HCl, 10 mM Hepes, 1 mM EDTA, 100 mM imidazole, 1% Triton) for five times and bound proteins were resolved by SDS-PAGE. The SDS-PAGE gel containing the protein sample was cut into pieces and destained with 25 mM ammonium bicarbonate/50% acetonitrile. Proteins in the sliced gels were reduced with 10 mM DTT at 37°C for 1 hr, and then alkylated with 25 mM iodoacetamide at RT for 1 hr in the dark before digested with trypsin (Sigma T1426; enzyme-to-substrate ratio 1:50) in 25 mM ammonium bicarbonate at 37°C overnight. Tryptic peptides were extracted from gel by sonication with a buffer containing 5% trifluoroacetic acid and 50% acetonitirile. The liquid was dried by SpeedVac, and peptides were resolubilized in 0.1% formic acid and filtered with 0.45 μm centrifugal filters before analysis with a TripleTOF 5600 mass spectrometer (AB SCIEX, Canada) coupled to an Eksigent nanoLC. Proteins were identified by searching the MS/MS spectra against the *Mus musculus* SwissProt database using the ProteinPilot 4.2 software. Carbamidomethylation of cysteine was set as the fixed modification. Trypsin was specified as the proteolytic enzyme with a maximum of 2 miss cleavages. Mass tolerance was set to 0.05 Da and the false discovery rates for both proteins and peptides were set at 1%.

## Co-immunoprecipitation (IP) and GST-pull down

For immunoprecipitation, HEK293T cells transfected with constructs expressing Flag-SNX6 and mEmerad-Homer1c were washed with ice-cold PBS and lysed with lysis buffer 1 (0.5% [vol/vol] NP-40, 50 mM Tris-HCl, pH 7.4, 150 mM NaCl, 1 mM EDTA) supplemented with protease inhibitors. The following steps were performed as previously described (*Hong et al., 2009*). HEK293T (ATCC, Manassas, VA) used for protein overexpression and immunoprecipitation in this study were negative for mycoplasma. Cells were cultured in DMEM (HyClone) supplemented with 10% fetal bovine serum (FBS) (HyClone), penicillin and streptomycin (HyClone). For endogenous IP, whole brains of 10 week-

old C57 mice were homogenized with 10 times volume of lysis buffer 2 (150 mM NaCl, 20 mM Tris-HCl, 10 mM Hepes, 1 mM EDTA, 1% TritonX-100, pH 7.4) supplemented with protease inhibitors and rotated for 30 min at 4°C. Lysates were centrifuged at 12,000 × g for 20 min. The supernatants were collected and incubated with 5 μg rabbit IgG (control) or mouse anti-SNX6 antibody bound to 20 μL of pre-washed Protein A/G Sepharose beads overnight at 4°C. Beads were washed five times with lysis buffer and bound proteins were eluted with 2× loading buffer and subjected to SDS-PAGE and immunoblotting. For GST-pull down assays, 2 μg of recombinant His-sumo-Homer1c-FL, His-sumo-Homer2b, His-sumo-Homer3, His-sumo-Homer1c-N, His-sumo-Homer1c-M1, His-sumo-Homer1c-M2 or His-sumo-Homer1c-C was incubated with GST, GST-SNX1-N or GST-SNX6-N immobilized on glutathione-Sepharose beads overnight at 4°C and proceeded as described for endogenous IP. For membrane IP/immunoisolation assays, mouse brain was homogenized with lysis buffer (20 mM Tris-HCl, 10 mM HEPES, pH 7.4, 150 mM NaCl, 1 mM EDTA, 0.25 M sucrose) supplemented with protease inhibitors and centrifuged at 800 × g for 15 min. The supernatants were collected and subjected to high-speed centrifugation at 100,000 × g for 1 hr (TLS-55 rotor, Optima MAX Ultracentrifuge, Beckman Coulter, USA). The pellets (p100, the membrane fraction) were resuspended in lysis buffer and subjected to immunoisolation with Dynabeads Protein G (Invitrogen, Carlsbad, CA, USA) coupled with 3–5 μg of mouse anti-SNX6 antibody. Then the beads were eluted by boiling in 2× SDS gel loading buffer and bound proteins were subjected to SDS-PAGE and immunoblotting.

## 3D-SIM superresolution microscopy and image analysis

3D-SIM images of immunostained neurons were acquired as previously described (*Niu et al., 2013*) on the DeltaVision OMX V4 imaging system (Applied Precision Inc, USA) with a 100 × 1.4 oil objective (Olympus UPlanSApo), solid state multimode lasers (488, 593 and 642 nm) and EMCCD cameras (Evolve 512 × 512, Photometrics, USA). For quantitative analysis of 3D-SIM images with three colors/triple channels, first raw images were committed to costes auto-threshold subtraction for unbiased background subtraction as previously described (*Dunn et al., 2011*). Then Biobprob ImageJ plugin was used to measure the colocalilzation of triple channels (*Fletcher et al., 2010*). The statistical significance of triple colocalization was determined by comparing the values of voxel colocalization of the original images with those of images generated by randomizing spatial locations of signals in the original ones (i.e., colocalization occurred by chance), which was illustrated with *p* values produced by Biobprob ImageJ plugin (parameters: voxel size: 40, 40, 125 nm; confidence interval: 95; p<0.05 means significantly more colocalization than chance).

## Live imaging acquisition and analysis

Hippocampal neurons isolated from embryonic day 17.5 (E17.5) or newborn (P0) mice were transfected at DIV 10 ~ 13 with Lipofectamine LTX (Invitrogen) and imaged live by TIR-FM (Nikon TE2000-E equipped with 488 and 561 nm solid laser, 473/543 filter, 60 × 1.49 oil objective (Nikon CFI Apochromat TIRF) and EMCCD camera (iXon Ultra 897, ANDOR, UK)) at DIV 11 ~ 16 at 37°C with 5% $CO_2$based on the optimal fluorescent protein expression. Image acquisition (512 × 512 pixels, two frames/sec for 5–10 min) was controlled by μManager software. Kymographs were prepared using NIH ImageJ functions: 'reslice' with one pixel Z-spacing (pixels) and 'Z projection' with 'Standard deviation' type. The distance between start position of each track in Kymographs and cell body was recorded and used for aligning these tracks of the motile SNX6-, Homer1c- positive vesicles along dendrites. Average velocities (run length / [ total time- pause time]) were acquired with ImageJ plugins 'Macros > read velocities from tsp'. For retrospective staining, cells were fixed with 4% PFA immediately after live imaging, and immunostained with antibodies to MAP2 or endogenous SNX6, Homer1b/c and VPS35 as previously described (*Hong et al., 2009*). For size measurement of fluorescent puncta, the shape of puncta was distinguished by the naked eye from the diffuse cytosolic signal and the area value was obtained by the ImageJ function 'Analyze>measure'.

## Fluorescence recovery after photobleaching

FRAP was performed as described by *Kerr and Blanpied (2012)*. Briefly, hippocampal neurons were transfected with constructs expressing mEmerald-Homer1c and pLL3.7.1-DsRed as spine volume marker on DIV13 and FRAP was performed on DIV15 ~ 16. Photobleaching of entire spines with

synaptic Homer1c signals was achieved using 50 ~ 60% 488 nm laser power, while the following acquisition immediately after bleaching was achieved using 5 ~ 6% 488 nm laser power. To analyze the recovery of fluorescence, raw images were background subtracted frame-by-frame. The bleached spine and an additional 'control' spine were targeted as ROI. The recovery rate was calculated as R = (I(t)-I(0))/(I(before bleaching)-I(0)), with I(0) being the intensity immediately after bleaching. After normalization using the 'control' spine, recovery trace of each bleached spine over time was drawn. The fluorescence recovery at the end of recording time was determined as the $R_{final}$. After averaging the intensity of five time points, the first time point at which the intensity recovered to the half of $R_{final}$ was determined as the half-time. By averaging all individual recovery traces and fitting a single exponential recovery curve using 'Graphpad Prism 5>analysis>fit>exponential>one-phase association>least squares fit', the value of $R_{final}$ and half-time obtained through the exponential fitting curve are similar to the average values obtained from each individual recovery trace.

## Brain slice culture and electrophysiology

Cultured hippocampal slices were prepared from P6-P8 $Snx6^{fl/fl}$ male mice as previously described (*Herring et al., 2013*). After one day in culture (DIV1), the slices were injected with adeno-associated virus (AAV) carrying EGFP-2A-Cre driven by the CaMKIIα promoter. Infected slices were cultured for an additional two weeks before recording. For recording, cultured slices were perfused with artificial cerebrospinal fluid (ACSF), containing (in mM) NaCl 119, KCl 2.5, NaHCO₃ 26, NaH₂PO₄ 1, glucose 11, CaCl₂ 4, MgCl₂ 4, 2-chloroadenosine 0.01 (to dampen epileptiform activity) and saturated with 95% $O_2$/5% $CO_2$. Isolation of currents from glutamatergic (AMPA and NMDA) receptors was achieved by adding picrotoxin (0.1 mM) to the ACSF to block GABAA receptors. CA1 pyramidal cells were visualized by infrared differential interference contrast microscopy. The intracellular solution contained (in mM) CsMeSO4 135, NaCl 8, HEPES 10, Na₃GTP 0.3, MgATP 4, EGTA 0.3, QX-314 5, and spermine 0.1. Cells were recorded with 3–5 MΩ borosilicate glass pipettes, following stimulation of Schaffer collaterals with bipolar metal electrodes placed in the stratum radiatum of the CA1 region. Series resistance was monitored and not compensated, and cells in which series resistance varied by 25% during a recording session were discarded. Synaptic responses were collected with a Multiclamp 700B-amplifier (Axon Instruments, Foster City, CA), filtered at 2 kHz and digitized at 10 kHz. GFP-positive neurons were identified by epifluorescence microscopy. All paired recordings involved simultaneous whole-cell recordings from one GFP-positive neuron and a neighboring GFP-negative neuron. The stimulus was adjusted to evoke a measurable, monosynaptic eEPSC in both cells. AMPAR eEPSCs were measured at a holding potential of −70 mV, and NMDAR eEPSCs were measured at +40 mV 150 ms after the stimulus, at which point the AMPAR eEPSC has completely decayed. All paired recording data were analyzed statistically with a Wilcoxon Sign Rank Test for paired data. A p value of <0.05 was considered statistically significant. Error bars represent standard error measurement.

## CiliobrevinD and mGluR inverse agonist treatment

At DIV 16, cultured hippocampal neurons were treated with 20 μM Ciliobrevin D (EMD Chemicals, Gibbstown, NJ) for 2 hr and fixed for immunostaining analysis. For mGluR inverse agonist treatment, experiments were performed as previously (*Hu et al., 2010*). Briefly, DIV14 neurons were treated with Bay 36–7620 (Bay, 10 μM, Sigma-Aldrich) and MPEP (5 μM, Sigma-Aldrich) dissolved in DMSO for 48 hr and fixed for the subsequent assay.

## GluA1 endocytosis assay

Endocytosis assay of GluA1 in steady state neurons was performed essentially as previously described (*Lu et al., 2007*). Briefly, DIV16 hippocampal neurons were pre-chilled on ice for 5 min and incubated with GluA1 N-terminal antibody (1:200, Millipore, Billerica, MA) for 30 min. After antibody washout, neurons were transferred to 37°C for 30 min before fixation in PFA-sucrose. Surface-bound GluA1 was immunostained by using Alexa Fluor 647-conjugated goat anti-mouse secondary antibodies (1:1000, Invitrogen) followed by blocking surface-remaining GluA1 with unconjugated goat anti-mouse secondary antibodies (undiluted stock, Jackson ImmunoResearch, West Grove, PA). Then neurons were post-fixed with PFA and permeabilized with blocking buffer containing 0.4%Triton-100. The internalized GluA1 was immunostained with Alexa Fluor 488-conjugated goat anti-

mouse antibody (1:1000, Invitrogen). Confocal images were captured and used for calculating the ratio of internalized GluR1 as follows after applying median and costes auto-threshold subtraction: mean intensity of internalized GluA1/(mean intensity of surface GluA1 + mean intensity of internalized GluA1).

## Biotinylation assay of surface proteins

Biotinylation assay was performed as previously described (*Fu et al., 2011*). Briefly, DIV16 hippocampal neurons cultured from newborn (P1) mice were washed twice with ice-cold PBS containing 1 mM $CaCl_2$ and 0.5 mM $MgCl_2$ (PBS-Ca-Mg), then incubated with PBS-Ca-Mg supplemented with 0.25 mg/ml EZ-link Sulfo-NHS-LC-biotin (Pierce, Thermo Fisher Scientific, Rockford, IL) for 1 hr at RT with mild shaking. The biotinylation reaction was quenched and unbound biotin was removed by washing the cells twice with PBS-Ca-Mg containing 100 mM glycine for 15 min at 4°C. Neurons were then lysed in lysis buffer (50 mM Tris-Cl, 10 mM HEPES, pH 7.4, 150 mM NaCl, 0.5 mM EDTA, 1% Triton) supplemented with protease inhibitors. After centrifugation at 8000 $\times$ g at 4°C for 10 min, the supernatants were collected and incubated with streptavidin beads (Thermo Fisher Scientific, Rockford, IL) overnight at 4°C. Beads were washed five times with lysis buffer. Bound proteins were eluted with 2$\times$ loading buffer and subjected to SDS-PAGE and immunoblotting.

## Statistical analysis

All data are presented as mean ± SEM. GraphPad Prism 5 (GraphPad Software) was used for statistical analysis. The two-tailed unpaired t-test was used for statistical analysis of immunoblotting data and to evaluate statistical significance of two groups of samples. One-way analysis of variance with a Tukey post hoc test was used to evaluate statistical significance of three or more groups of samples. For TEM data, non-parametric two-sided statistical testing Mann–Whitney was used to avoid the restriction of sample sizes without a normal distribution (*Morris, 2000*). A p value of less than 0.05 was considered statistically significant.

# Acknowledgements

We thank Drs. Michael Davidson, Johannes Hell, Connie Cepko, A Kimberley McAllister, Vivek Malhotra, Li Yu and Hong Tang for providing reagents, Dr. Jian-Jun Yang (Jinling Hospital, School of Medicine, Nanjing University, China) for technical advices and Dr. You-ming Lu (Huazhong University of Science and Technology, China) for valuable advice in carrying out the Morris water maze tests. This work was supported by funding from the National Natural Science Foundation of China (31530039, 31325017 and 31471334 to J-J Liu), the Ministry of Science and Technology of China (2014CB942802 and 2016YFA0500100 to J-J Liu and 2014CB942804 to YS Shi) and State Key Laboratory of Molecular Developmental Biology (2014-MDB-TS-01 to J-J Liu).

# Additional information

### Funding

| Funder | Grant reference number | Author |
| --- | --- | --- |
| National Natural Science Foundation of China | 31530039 | Yang Niu<br>Zhonghua Dai<br>Cheng Zhang<br>Yanrui Yang<br>Zhenzhen Guo<br>Xiaoyu Li<br>Jia-Jia Liu |
| Ministry of Science and Technology of the People's Republic of China | 2014CB942802 | Yang Niu<br>Zhonghua Dai<br>Wenxue Liu<br>Cheng Zhang<br>Yanrui Yang<br>Zhenzhen Guo<br>Xiaoyu Li<br>Yun S Shi<br>Jia-Jia Liu |

| | | |
|---|---|---|
| State Key Laboratory of Molecular Developmental Biology, China | 2014-MDB-TS-01 | Yang Niu<br>Zhonghua Dai<br>Cheng Zhang<br>Yanrui Yang<br>Zhenzhen Guo<br>Xiaoyu Li<br>Xiahe Huang<br>Yingchun Wang<br>Jia-Jia Liu |
| National Natural Science Foundation of China | 31325017 | Yang Niu<br>Zhonghua Dai<br>Cheng Zhang<br>Yanrui Yang<br>Zhenzhen Guo<br>Xiaoyu Li<br>Jia-Jia Liu |
| National Natural Science Foundation of China | 31471334 | Yang Niu<br>Zhonghua Dai<br>Cheng Zhang<br>Yanrui Yang<br>Zhenzhen Guo<br>Xiaoyu Li<br>Jia-Jia Liu |
| Ministry of Science and Technology of the People's Republic of China | 2016YFA0500100 | Yang Niu<br>Zhonghua Dai<br>Wenxue Liu<br>Cheng Zhang<br>Yanrui Yang<br>Zhenzhen Guo<br>Xiaoyu Li<br>Yun S Shi<br>Jia-Jia Liu |
| Ministry of Science and Technology of the People's Republic of China | 2014CB942804 | Yang Niu<br>Zhonghua Dai<br>Wenxue Liu<br>Cheng Zhang<br>Yanrui Yang<br>Zhenzhen Guo<br>Xiaoyu Li<br>Yun S Shi<br>Jia-Jia Liu |

The funders had no role in study design, data collection and interpretation, or the decision to submit the work for publication.

## Author contributions

YN, ZD, Data curation, Formal analysis, Investigation, Visualization, Methodology, Writing—original draft, Writing—review and editing; WL, Data curation, Formal analysis, Validation, Investigation, Visualization, Methodology, Writing—original draft, Writing—review and editing; CZ, Formal analysis, Investigation, Visualization, Writing—original draft; YY, Data curation, Formal analysis, Investigation, Methodology, Writing—original draft, Writing—review and editing; ZG, XL, XH, Data curation, Formal analysis, Investigation, Writing—original draft; CX, Formal analysis, Investigation, Methodology, Writing—original draft; YW, Formal analysis, Supervision, Funding acquisition, Investigation, Methodology, Writing—original draft; YSS, Conceptualization, Data curation, Formal analysis, Supervision, Funding acquisition, Investigation, Visualization, Methodology, Writing—original draft, Project administration, Writing—review and editing; J-JL, Conceptualization, Data curation, Formal analysis, Supervision, Funding acquisition, Investigation, Visualization, Methodology, Writing—original draft, Project administration, Writing—review and editing

## Author ORCIDs

Jia-Jia Liu, http://orcid.org/0000-0002-6099-1059

## Ethics

Animal experimentation: This study was performed in compliance with the guidelines of the Animal Care and Use committee of the Institute of Genetics and Developmental Biology, Chinese Academy

of Sciences (Permit Number: AP2013001). All surgery was performed under sodium pentobarbital anesthesia, and every effort was made to minimize suffering.

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
