## [Decision Letter]

[Editors’ note: this article was originally rejected after discussions between the reviewers. Further to an appeal, the editors stood by their decision, however offering the option of a resubmission.]

Thank you for submitting your work entitled "Synaptic Delivery of Homer1b/c by SNX6-mediated Dendritic Vesicular Transport" for consideration by *eLife*. Your article has been reviewed by three peer reviewers, and the evaluation has been overseen by a Reviewing Editor and a Senior Editor. The reviewers have opted to remain anonymous.

Our decision has been reached after consultation between the reviewers. Based on these discussions and the individual reviews below, we regret to inform you that your work will not be considered further for publication in *eLife*.

All the reviewers found that the paper contains a significant amount of novel data, that the description of the SNX6 knockout mice is interesting and solid, and that the biochemical link between SNX6 and Homer1b/c is well described and relevant. However, two of the three reviewers found that the major conclusion of the paper on SNX6 mediating a novel dynein-dependent dendritic transport pathway responsible for the synaptic delivery of Homer1b/c is not sufficiently supported by data. In particular, the reviewers had major concerns about the quality of the live imaging data and the rather weak biochemical and functional evidence supporting a specific connection to dynein. Furthermore, the data proving that the proposed novel transport pathway is retromer-independent were found not be sufficiently strong. The third reviewer has pointed out that the behavioral data were not complete and the data that support the increased level of surface mGlu5 were not convincing. Based on these major criticisms concerning the most important conclusions of the paper, we cannot proceed with this manuscript in e*Life*. However, we hope that you will find the comments of the reviewers useful for further developing your study.

*Reviewer #1:*

This paper describes the neuronal function of the sorting nexin family member SNX6. The authors show that SNX6 knockout mice exhibit defects in learning and memory, a phenotype that they connect to the reduction in the number of dendritic spines in a subpopulation of hippocampal pyramidal cells. Further, they uncover a biochemical connection between SNX6 and the postsynaptic scaffolding protein Homer1b/c, and show that SNX6 knockout leads to a reduction in the intensity of Homer1b/c in dendrites and affects surface levels of AMPA receptors and synaptic transmission. They also demonstrate co-motility of SNX6 and Homer1b/c, and propose that SNX6 mediates dynein-dynactin mediated transport and synaptic delivery of Homer1/b. Finally, the authors provide some data suggesting that this function of SNX6 is independent of other components of the retromer complex, a part of which SNX6 forms.

This paper is very rich in data, and provides solid new information on the function of SNX6 in neurons. However, the data supporting the major mechanistic conclusion of the authors on SNX6 mediating the dendritic transport and synaptic delivery of Homer1b/c to dendritic spines is not very strong.

Major comments:

1) For the proposed model to work, SNX6 should be able to bind to Homer1b/c and dynactin at the same time. However, the PX domain of SNX seems to be necessary for binding to both Homer1b/c and to dynactin, and it should be clarified whether both interactions can occur simultaneously.

2) Figure 5 demonstrates a dramatic reduction of Homer1b/c intensity throughout the dendrite in a SNX6 knockout neuron. It should be investigated whether the expression level of Homer1b/c is affected by SNX6 knockout, and if not, in which part of the neuron Homer1b/c accumulates. Is it possible that SNX6 regulates not the transport but stability of Homer1b/c?

3) The authors do not show that the motility of Homer1b/c is affected by the loss of SNX6. If the authors would like to make conclusion about SNX6 as a factor involved in Homer1b/c transport, such data should be included.

4) Triple colocalizations of EEA1/SNX6/Homer1b/c (Figure 4), and p150 or DIC/ SNX6/Homer1b/c (Figure 5) show very little actual colocalization between the three markers, and could just as well reflect spurious overlap between abundant punctate labeling patterns. Quantifications should be provided illustrating the significance of colocalizations by using shifted images as controls.

5) When the authors discuss Homer1b/c transport, do they mean transport from the cell body, or redistribution along the dendrite? If they mean transport from the cell body, why would an early endosome serve as a carrier? If relocalization of Homer1b/c along the dendrite is affected, why is the Homer1b/c intensity reduced throughout the dendrite? More clear data and a better discussion are needed here, including alternative models of the functional connection between SNX6 and Homer1b/c.

6) The strong conclusion on retromer-independent function of SNX6 in Homer1b/c transport seems to be mostly based on the observation that while the loss of both SNX6 and VPS35 reduces the number of Homer1b/c puncta, only the knockout of SNX6 affect Homer1b/c intensity is spines. However, this comparison is not fair, because a SNX6 knockout and VPS35 knockdown are used, and the data are thus not directly comparable. The observation that VPS35 and Homer1b/c do not colocalize may be due to the fact that the two proteins do not bind directly but only through SNX6, which becomes limiting when both VPS35 and Homer1b/c are overexpressed. Is colocolalization of VPS35 and Homer1b/c observed when SNX6 is overexpressed as well? Altogether, the authors should either provide additional data showing that SNX6-dependent distribution of Homer1b/c does not require retromer, or remove this conclusion.

*Reviewer #2:*

In this paper Niu et al. demonstrate that the CNS-specific Snx6 knockout mice exhibit deficits in spatial learning and memory, accompanied with loss of spines from apical dendrites of hippocampal CA1 pyramidal cells.

Interestingly they found that SNX6 functions independently of retromer to mediate vesicular transport of Homer1b/c to PSD. Indeed with a number of experiments they showed that the ablation of SNX6 causes loss of Homer1b/c from spines as well as decreases in surface levels of AMPAR and AMPAR-mediated synaptic transmission. Thus this paper potentially identifies a novel dendritic transport pathway that contributes to synaptic structure and function. This is a nice and well presented paper, however the following experiments and controls are absolutely required to really complete the paper.

Major points:

– Alteration of mGlu5 signaling and altered Homer1 localization to synapses has been associated to impaired social behavior in mice. Thus it will be interesting to characterize the social and repetitive behavior in the SNX6 KO mice.

– Dendrite spine morphology and synapse structure should be also analyzed in cortex.

– The quality of EM images is poor and should be improved substantially.

– The surface staining of mGlu5 is not very convincing. The higher expression of surface mGlu5 in the SNX6 KO mice should be proved also with biochemical experiments.

*Reviewer #3:*

Niu et al., investigates the physiological role of SNX6, which is a known component of the retrograde complex. Loss of SNX6 caused deficits in a/ spine density in hippocampal CA1 neurons, b/ hippocampal-dependent spatial learning and memory, c/ synaptic homer and AMPA receptor levels, d/ AMPA receptor-mediated synaptic transmission and e/ coupling of the endocytic zone to PSD. These data suggest an important role for SNX6 in synaptic structure and function. The biochemical, electrophysiology and behavioral experiments in this paper are well executed and are high quality but the cell biological part is poorly developed. The cell biological / localization analysis / live imaging data presented here (Figure 4, Figure 5 and some panels in Figure 6 and Figure 7) are insufficient to support the major conclusions of the paper. It is possible that SNX6 directly transport homer1 but this is not shown by the data presented in this manuscript. Moreover, the paper lacks evidence for the claim that SNX6 mediates dynein-driven long range dendritic transport of Homer1b/c,

Major concerns:

1) From the current data it cannot be concluded that there is a "SNX6-mediated Homer1b/c transport pathway (Video 2–Video 4)". Overexpression of the fluorescently tagged SNX6 and homer1 look completely different than the endogenous staining patterns and do agree with previous reported subcellular distributions. Most of the SNX6 and homer1 clusters are static and do not move and are not present at synaptic sites.

2) There is no evidence in this paper that "SNX6 mediates vesicular transport of Homer1b/c to synaptic sites in dendrites". Other than previous data on an interaction between dynein and SNX6 and the pulldown experiments shown in Figure 4 there is no functional evidence for the role dynein in this transport pathway. The authors should provide functional data on the link between dynein and SNX6 and homer in dendritic transport.

3) There is no evidence in this paper that there is "a novel dendritic transport pathway that contributes to synaptic structure and function" (last sentence in the Abstract). Authors should first perform high-quality imaging experiments to exclude that other 'dendritic' trafficking pathways are not involved. Such as secretory pathways (work from Mike Ehlers) and other routes transporting mobile scaffolding proteins (Gerrow et al., Neuron, 2016). Moreover, the data do not convincingly show that the phenotype in SNX6 knockout mice is retromer independent.

[Editors’ note: what now follows is the decision letter after the appeal].

Thank you for choosing to send your work entitled "Synaptic Delivery of Homer1b/c by SNX6-mediated Dendritic Vesicular Transport" for consideration at *eLife*. Your article and your letter of appeal have been considered by a Senior Editor and a Reviewing Editor, and we regret to inform you that we are upholding our original decision. The concerns raised by the reviewers about the strength of the evidence supporting your major conclusions were very serious, and therefore, the paper in its current form is not suitable for publication in *eLife*. However, should you be able to thoroughly address the issues raised by the three reviewers with additional experiments, we will be prepared to consider a new submission for a formal review by the same reviewers.

If you choose to resubmit your paper, please give particular consideration to the following points:

1) It would be important to improve the quality of live imaging data by showing that the fluorescent proteins used for these experiments faithfully recapitulate the distribution of endogenous proteins.

2) It would also be important to provide clear proof for the idea that SNX6 indeed serves as a linker between Homer1b/c and cytoplasmic dynein. Please note that various strategies of inhibiting dynein are by themselves not sufficient here, as the inhibition of dynein is well known to affect most microtubule-based transport pathways independent of their identity. One would like to see proof that SNX6 really connects Homer1b/c to dynein, for example, by showing a reduced colocalization or a reduced biochemical interaction between Homer1b/c and the dynein complex in SNX6 knockout. Alternatively, you might consider a possibility that SNX6 participates in transporting Homer1b/c by simply recruiting it to endosomes, which then bind to microtubule motors independently of SNX6. It would also be useful to consider alternative ways of how SNX6 could regulate the abundance of Homer1b/c at the synapses.

3) On the technical side, please note that the use of SIM microscopy is an asset, but since the improvement in resolution provided by this technique is modest, its use does not overcome the need for proper unbiased quantifications, especially when the analyzed samples are highly crowded.

4) For proving that the pathway you are analyzing is indeed retromer-independent, it would be important to demonstrate better the efficiency of the knockdown of VPS35 in neurons, because if the knockdown is only partial and not complete, this might explain the differences between VPS35 knockdown and the SNX6 knockout phenotype, especially as the loss VPS35 does affect the number of Homer1b/c puncta in dendrites. Please also refer to the comments of reviewer 1 on this issue.

[Editors’ note: Further to the previous decision, the authors submitted a new version of the manuscript. The decision letter after another round of full peer review follows.]

Thank you for resubmitting your work entitled "Synaptic Delivery of Homer1b/c by SNX6-mediated Dendritic Vesicular Transport" for further consideration at *eLife*. Your revised article has been evaluated by Anna Akhmanova (Senior editor), a Reviewing editor, and two reviewers.

The reviewers agree that the part of the paper describing the mouse phenotype is well done. However, although the revisions have alleviated some technical concerns on the cell biological part, the evidence about SNX6 and Homer1 participating in a novel dendritic transport pathway remains insufficiently convincing.

First, if Homer1b/c localizes to early endosomes and is transported with them, then it is not a novel pathway. Moreover, as the authors show in Figure 6, SNX6 knockout does not affect the 40% colocalisation between Homer1b/c and an early endosome marker. But then, surprisingly, SNX6 knockout does affect Homer1 motility and colocalization with dynein. Do the authors actually mean that the complex of SNX6 and Homer1b/c is required to transport early endosomes into dendrites? The authors should be much more explicit on this issue, as other molecules have been implicated in endosome transport. If Homer1b/c and SNX6 are transported on early endosomes, then SNX6 knockout should affect endosome motility. This should be easy to measure, and such data should be included. Alternatively, if SNX6 in complex with dynein transports the non-endosomal Homer1 population, what kind of vesicles are these, and what is then the relevance of colocalization of Homer1 with endosomes?

Furthermore, the new data provided by the authors show that fluorescently tagged Homer1b/c is present in large structures, many of which do not colocalize with a PSD marker. In contrast, the endogenous Homer1b/c is well known to show a synaptic localization. It is thus possible that the non-synaptic fluorescently tagged Homer 1b/c is present in aggregates, and that loss of SNX6 affects the very infrequent motility of these aggregates. The motility of fluorescent Homer1b/c particles might then be irrelevant to the distribution of synaptic Homer1. It is possible that the synaptic localization of Homer1b/c actually depends on protein diffusion and not on microtubule-based transport, an option that is not even properly discussed. The authors should use FRAP to investigate the turnover of the synaptic population of Homer1b/c in order to find out whether synaptic Homer1 exchanges with the soluble cytosolic pool of the protein or traffics into synapses as particles, and whether any of these processes are affected by the loss of SNX6.

If the authors cannot satisfactory uncover the nature of the "new dendritic transport pathway" that they propose or if it turns out that Homer1b/c accumulates in the synapses by exchanging with the soluble pool of the protein, the title, Abstract, the text and the conclusions of the paper will need to be very thoroughly revised accordingly.

[Editors' note: further revisions were requested prior to acceptance, as described below.]

Thank you for resubmitting your work entitled "Dendritic Delivery of Homer1b/c by SNX6-mediated Long-range Vesicular Transport" for further consideration at *eLife*. Your revised article has been favorably evaluated by Anna Akhmanova (Senior editor), a Reviewing editor, and two reviewers.

The manuscript has been improved but there are some remaining issues that need to be addressed before acceptance, as outlined below:

In the revised version of the paper, the authors performed the experiments suggested by the reviewers. The results most relevant to the overall message of the paper are that the labeled Homer1c structures only very rarely enter spines and that photobleached Homer1c in spines shows SNX6-independent recovery by 50% within 10 minutes through exchange with the soluble pool. These are very significant results, because together, they provide strong support for the view that Homer1c is not delivered into spines on endosomes but simply exchanges with the soluble pool in a SNX6-independent manner, which is not unexpected. Since most of dendritic Homer1c is present in spines, but vesicular transport is not responsible for delivering Homer1c into spines, the role of SNX6-dependent vesicle transport in "dendritic delivery of Homer1b/c" becomes confusing. Is it needed to redistribute soluble Homer1c through dendrites, so that it can then diffuse into spines? Or does Homer1c on endosomes represent part of a Homer1c pathway distinct from Homer1c function in dendritic spines? Based on the data presented in the paper, these questions are impossible to address. However, the data as they stand now certainly bring the importance of the observed SNX6-dependent transport of Homer1c on endosomes into question.

Given the large amount of work the authors have invested in revising the paper, the good quality of much of the data and the complexity of the problem, the reviewers felt that the paper can still be published. However, it would be essential to very thoroughly revise the text of the paper in a way that would do justice to the data shown, and not just to the model that the authors were trying to prove. To achieve this, it would be essential to very strongly downplay the "SNX6-mediated transport and delivery" angle, particularly by removing it from the title, re-writing the Abstract and Introduction, adding the data on FRAP and the failure to detect vesicle-based delivery of Homer1c into spines into the main figures, and writing a very balanced discussion. The revised version of the paper should include all the data but should not attempt to create an impression that the authors have proven that the phenotypes of SNX6 knockout mice are due to altered transport and delivery of Homer1c on SNX6-positive endosomes into dendritic spines, thus explaining the spine phenotypes observed. The reviewers would like to emphasize that "cosmetic" changes to the writing will not be sufficient in this case.

It is possible that the authors might disagree with the opinion of the reviewers, and in this case they are advised to seek publication in another journal.

Should you decide to resubmit, please address this additional point: the data of panel K of Figure 8 on the surface staining of mGlu5 do not look convincing. Please move them to the Supplement and move the supplementary biochemical data (Figure 8—figure supplement 1) to the main figure.

[Editors' note: further revisions were requested prior to acceptance, as described below.]

Thank you for choosing to send your work entitled "A Neuronal Role for SNX6 in Dendritic Trafficking of the Postsynaptic Scaffold Protein Homer1b/c" for consideration at *eLife*. The new version of your manuscript was reviewed by the Editor and one of the previous reviewers and, once again, they are not satisfied with the limited revisions you have made in response to their criticisms. Their initial reaction was to reject your paper with no chance for reconsideration but I requested they give you one last chance and here is their firmly written opinion that you simply must take seriously if you wish to have this work published in *eLife*:

Specifically, in the previous decision letters we have very clearly indicated that we cannot offer to publish your paper in the current form, because the reviewers found that your major claim that " SNX6 mediates long-range vesicular transport of Homer1b/c to distal dendrites" is not sufficiently supported by data. With other words, it is possible that SNX6 indeed mediates long-range transport of Homer1b/c, but it is also possible that Homer1b/c arrives to the dendritic spines by diffusion alone, and the interactions that you described serve another function (if any). Given the complexity of the subject and the large amount of work that you have put into revising this manuscript, we have offered you an opportunity to completely and thoroughly revise the whole paper in a way that would make it clear to the reader from the start to the end that your transport-related conclusions are not more than one of the possible explanations of the mouse phenotypes that you nicely describe. Such re-writing would entail not a minor but a major revision of your paper. Given your new suggestion for the title and the minor edits you have done, it is very clear to us that you are not willing to perform such a major revision. We of course fully respect your decision, because this is your paper, and you should write it in the way you find most appropriate. We would like to emphasize that your English writing is perfectly fine, and that our negative decision is only due to the fact the reviewers were not convinced that your major conclusion is sufficiently supported by data. We would also like to emphasize that since the reviewers are not co-authors of your paper, it is not their task to search for "sentences or words in the manuscript still misleading or overstated" – it is up to authors to make such changes.

We are willing to give you one more chance to re-write the paper. However, you would need to really thoroughly rethink and revise the whole manuscript to provide a more balanced view of different ways in which their data can be explained. You are very much in favor of pushing through the model that SNX6 mediates transport of Homer1b/c on vesicles, but the experimental support for this model is insufficient and the FRAP data directly contradict it. Therefore, the paper needs to be re-written in a way that would make it clear to the reader that transport of Homer1b/c on endosomes is not more than one of the possible explanations of the observed mouse phenotypes. Trying to solve the problem by adding a few disclaimers here and there is not enough, and also not what we have asked for. For example, in the latest version, you proposed to revert to the title with transport being the main message, while we asked for this change one revision before.

---

## [Author Response]

[Editors’ note: this article was originally rejected after discussions between the reviewers. The authors’ rebuttal follows.]

*Reviewer #1:*

*This paper describes the neuronal function of the sorting nexin family member SNX6. The authors show that SNX6 knockout mice exhibit defects in learning and memory, a phenotype that they connect to the reduction in the number of dendritic spines in a subpopulation of hippocampal pyramidal cells. Further, they uncover a biochemical connection between SNX6 and the postsynaptic scaffolding protein Homer1b/c, and show that SNX6 knockout leads to a reduction in the intensity of Homer1b/c in dendrites and affects surface levels of AMPA receptors and synaptic transmission. They also demonstrate co-motility of SNX6 and Homer1b/c, and propose that SNX6 mediates dynein-dynactin mediated transport and synaptic delivery of Homer1/b. Finally, the authors provide some data suggesting that this function of SNX6 is independent of other components of the retromer complex, a part of which SNX6 forms.*

*This paper is very rich in data, and provides solid new information on the function of SNX6 in neurons. However, the data supporting the major mechanistic conclusion of the authors on SNX6 mediating the dendritic transport and synaptic delivery of Homer1b/c to dendritic spines is not very strong.*

*Major comments:*

*1) For the proposed model to work, SNX6 should be able to bind to Homer1b/c and dynactin at the same time. However, the PX domain of SNX seems to be necessary for binding to both Homer1b/c and to dynactin, and it should be clarified whether both interactions can occur simultaneously.*

We thank the reviewer for the very insightful comment. Two proteins binding to the same domain of another protein does not necessarily mean they bind to the same amino acid residues. By immunoisolation, we detected SNX6, Homer1b/c and dynein−dynactin on SNX6-positive vesicles (Figure 4 in the original manuscript). Although the interaction between SNX6 and Homer1b/c is pretty strong in both in vitro binding assay (Figure 4 in the original manuscript) and coIP with overexpressing proteins (Figure 4 in the original manuscript), given the fact that the interaction between dynein−dynactin and SNX6 is weak and transient, and can only be detected when both proteins (endogenous) are associated with membranes, which is well documented in our previous studies (Hong et al., 2009; Niu et al., 2013), it is not practical to coIP all three proteins with buffer containing detergent. Therefore, we decided to take an alternative approach to determine whether SNX6 is required for association of Homer1b/c cargo with dynein−dynactin: we performed immunoisolation experiment of membrane fractions from mouse brain with antibodies to Homer1b/c, p150^Glued^ or DIC and examined whether these proteins still associate with the same vesicles in the absence of SNX6. Although the anti-Homer1b/c antibody failed to immunoisolate/enrich Homer1b/c-positive vesicles, we are very pleased that Homer1b/c was detected on p150Glued- or DIC-positive vesicles isolated from wild-type but not SNX6 knockout mouse brain (Figure 6). This piece of data indicates that association of Homer1b/c and the motor on the same vesicles requires SNX6.

Moreover, we also performed co-immunostaining of neurons and found that the colocalization between Homer1b/c and p150^Glued^/DIC decreased in SNX6 knockout neurons, whereas its colocalization with the endosome marker EEA1 did not change (Figure 6). Together these biochemical and cell biological data indicate that SNX6 serves as linker between Homer1b/c and the dynein−dynactin motor complex. These results are now incorporated in the text and the data is shown in new Figure 6.

*2) Figure 5 demonstrates a dramatic reduction of Homer1b/c intensity throughout the dendrite in a SNX6 knockout neuron. It should be investigated whether the expression level of Homer1b/c is affected by SNX6 knockout, and if not, in which part of the neuron Homer1b/c accumulates. Is it possible that SNX6 regulates not the transport but stability of Homer1b/c?*

We thank the reviewer for the very thoughtful comments. First we determined whether the expression level of Homer1b/c is affected by SNX6 knockout by immunoblotting of hippocampal lysates. Quantitative analysis of results from 3 pairs of wild-type and knockout littermates indicate that there was no significant difference in Homer1b/c levels, which means that ablation of SNX6 did not affect its protein stability. Second, in the original manuscript we only quantified the mean intensity of Homer1b/c signals in dendritic segments 30-120 μm from the cell body of Snx6+/+ and Snx6-/- neurons. In light of the reviewer’s comments, we quantified the dendritic distribution of Homer1b/c signals relative to the cell body. We are very pleased to find out that consistent with the phenotype caused by inhibition of dynein−dynactin activity (new Figure 6), there is a decrease in Homer1b/c signals in distal dendrite with concurrent increase in the cell body, indicating that indeed loss of SNX6 impairs long-range transport of Homer1b/c from the cell body to dendrite. These results are incorporated as Figure 5—figure supplement 3 and Figure 5 in the manuscript.

*3) The authors do not show that the motility of Homer1b/c is affected by the loss of SNX6. If the authors would like to make conclusion about SNX6 as a factor involved in Homer1b/c transport, such data should be included.*

Thanks a lot for the very insightful comments. To address this issue, we performed live imaging of mEmerald-Homer1c in dendrites of SNX6 knockout neurons. Quantification of motile Homer1c puncta indicates that compared with wild-type neurons, in which 29 out of 311 puncta from 10 neurons are motile, only 10 out of 1217 puncta from 40 knockout neurons are motile, indicating that indeed dendritic transport of Homer1b/c requires SNX6 activity. These results are incorporated in the text under the subtitle “Dendritic vesicular transport and spine localization of Homer1b/c require SNX6”.

*4) Triple colocalizations of EEA1/SNX6/Homer1b/c (Figure 4), and p150 or DIC/ SNX6/Homer1b/c (Figure 5) show very little actual colocalization between the three markers, and could just as well reflect spurious overlap between abundant punctate labeling patterns. Quantifications should be provided illustrating the significance of colocalizations by using shifted images as controls.*

Thanks a lot for the constructive comment on colocalization analysis of superresolution images. In the original manuscript we used the conventional method to quantify the overlap between three fluorescent labels using Mander’s coefficient as a measure of colocalization. We redid the colocalization analysis on the images (15 neurons/group) and assessed the statistical significance of the data with the methodology developed by Costes et al. (Automatic and Quantitative Measurement of Protein-Protein Colocalization in Live Cells. Biophys. J. 86, 3993-4003.) (Costes et al., 2004) and Fletcher et al. (Multi-Image Colocalization and Its Statistical Significance. Biophys. J. 99, 1996-2005) (Fletcher et al., 2010). Basically we evaluated the statistical significance of the values of voxel (corresponds to pixel in 2D images) colocalization by comparing them with those of images generated by randomizing spatial locations of signals in original images (i.e., colocalization occurred by chance). The results show that the voxel colocalization of EEA1-SNX6-Homer1b/c, p150Glued-SNX6-Homer1b/c and DIC-SNX6-Homer1b/c in the 3D-SIM images are significantly more than chance. The quantification data are added to the manuscript as Table 1 and Figure 4—figure supplement 1.

*5) When the authors discuss Homer1b/c transport, do they mean transport from the cell body, or redistribution along the dendrite? If they mean transport from the cell body, why would an early endosome serve as a carrier? If relocalization of Homer1b/c along the dendrite is affected, why is the Homer1b/c intensity reduced throughout the dendrite? More clear data and a better discussion are needed here, including alternative models of the functional connection between SNX6 and Homer1b/c.*

We thank the reviewer for the thoughtful comments. The question as to whether or not Homer1b/c is transported from the cell body is similar to that raised in Comment #2. First, it has been well established that early endosomes serve as carrier that is transported from the cell body to dendrites to provide membrane supply and other materials for dendrite development (Satoh et al., 2008; Zheng et al., 2008). As we have discussed in the original manuscript, microtubules in the dendrite are of mixed polarity and dynein serves as the motor to transport cargoes from the cell body to dendrite, our data that Homer1b/c transport is driven by dynein is in full agreement with previous studies (Satoh et al., 2008; Zheng et al., 2008; Kapitein et al., 2010). Second, in our initial analysis of Homer1b/c distribution in dendrite, we quantified mean intensity of Homer1b/c in dendritic segments ranging from 30 to 120 μm from the cell body, and focused on spine localization of Homer1. In light of reviewer’s comments we have performed quantitative analysis of confocal images to determine whether its dendritic distribution decreases towards the distal end of dendrites. We are very pleased to find that indeed, similar to phenotypes caused by inhibition of dynein−dynactin activity (new Figure 6), the signal intensity of Homer1b/c decreases in distal dendrites in SNX6 knockout neurons, accompanied with its accumulation in the cell body, which was rescued by overexpression of mCherry-SNX6 (new Figure 5, please refer to Response to Comment #2). These data together indicate that transport of Homer1b/c from the cell body to dendrite requires dynein−dynactin, which support and greatly strengthen our original conclusions and are added to the manuscript as Figure 5.

*6) The strong conclusion on retromer-independent function of SNX6 in Homer1b/c transport seems to be mostly based on the observation that while the loss of both SNX6 and VPS35 reduces the number of Homer1b/c puncta, only the knockout of SNX6 affect Homer1b/c intensity is spines. However, this comparison is not fair, because a SNX6 knockout and VPS35 knockdown are used, and the data are thus not directly comparable. The observation that VPS35 and Homer1b/c do not colocalize may be due to the fact that the two proteins do not bind directly but only through SNX6, which becomes limiting when both VPS35 and Homer1b/c are overexpressed. Is colocolalization of VPS35 and Homer1b/c observed when SNX6 is overexpressed as well? Altogether, the authors should either provide additional data showing that SNX6-dependent distribution of Homer1b/c does not require retromer, or remove this conclusion.*

Thanks for the very insightful and constructive comments on VPS35. Because VPS35 knockout is lethal (Wen et al., 2011), in previous studies researchers studied its cellular and molecular functions by either using heterozygous mice (Vps35+/m, expression of which is ~50% of wild-type, (Wen et al., 2011)), suppressing its expression by RNA interference (Wang et al., 2012) or overexpressing the Parkinson’s Disease-related D620N point mutant (Munsie et al., 2015; Tang et al., 2015). Since we do not have the Vps35+/m mice, based on the previous findings on VPS35 haploinsufficiency and the role of the retromer core complex in neuronal survival and degeneration (Wen et al., 2011; Liu et al., 2014; Wang et al., 2014; Tang et al., 2015), i.e., even ~50% reduction in VPS35 protein levels causes severe phenotypes in neuronal function and survival, we chose to transiently knockdown its expression in hippocampal neurons to assess its role in dendritic transport of Homer1b/c. Quantitative analysis shows that the VPS35 knockdown efficiency is 70-80% (Figure 11). In fact, hippocampal neurons transfected with VPS35 siRNA-expressing construct were not as healthy as those expressing the scrambled siRNA, which is in good agreement with previous findings that VPS35 function is essential for cell survival.

Author response image 1.Quantitative analysis of VPS35 knockdown efficiency.(**A**) HEK 293 cells were transfected with lentiviral vector expressing siRNA along with EGFP followed by immunoblotting with antibodies to VPS35 and β-actin after 72 h. Right panel shows quantification of immunoblots. Relative amount of VPS35 was determined with NIH ImageJ (N = 3 experiments). (**B**) Hippocampal neurons were transfected with lentiviral vector expressing siRNA along with GFP at DIV12, fixed on DIV18 and immunostained with antibodies to VPS35. Right panels are quantification of VPS35 signals in the outlined area of neurons (mean ± SEM, n = 10 neurons from three independent experiments). Scale bar, 5 μm.**DOI:**
http://dx.doi.org/10.7554/eLife.20991.042

To determine whether VPS35 and Homer1b/c bind through SNX6, we performed co-immunostaining of endogenous proteins (to avoid potential artifacts caused by overexpression of fluorescent fusion proteins) and found that, first of all, colocalization between Homer1b/c and VPS35 in dendrite is low compared with colocalization of Homer1b/c with SNX6 and EEA1; second of all, it was not affected by ablation of SNX6 (Figure 9—figure supplement 2). Consistently, no change was detected in Homer1b/c-SNX6 colocalization in VPS35-depleted neurons (Figure 9—figure supplement 2). Together these data indicate that VPS35 is not involved in SNX6-mediated Homer1b/c transport.

Moreover, to determine whether VPS35 depletion causes defect in Homer1b/c transport from the cell body to dendrites, we also performed quantitative analysis of Homer1b/c distribution in VPS35-depleted neurons and found that, in contrast to accumulation of Homer1b/c signals in the cell body and decrease in distal dendrite in SNX6 knockout neurons, the signal intensity of Homer1b/c decreases throughout the cell when VPS35 is suppressed. The decrease in Homer1b/c distribution in dendrites of VPS35-depleted neurons might be due to impaired cell viability/function and lower spine density as VPS35 has been found to promote spine formation and maturation (Wang et al., 2012; Tian et al., 2015). Together these data indicate that SNX6 and VPS35 function in synaptic localization of postsynaptic proteins via different mechanisms. These results are now incorporated in the manuscript as Figure 9 and Figure 9—figure supplement 1 and Figure 9—figure supplement 2.

*Reviewer #2:*

In this paper Niu et al. demonstrate that the CNS-specific Snx6 knockout mice exhibit deficits in spatial learning and memory, accompanied with loss of spines from apical dendrites of hippocampal CA1 pyramidal cells.

*Interestingly they found that SNX6 functions independently of retromer to mediate vesicular transport of Homer1b/c to PSD. Indeed with a number of experiments they showed that the ablation of SNX6 causes loss of Homer1b/c from spines as well as decreases in surface levels of AMPAR and AMPAR-mediated synaptic transmission. Thus this paper potentially identifies a novel dendritic transport pathway that contributes to synaptic structure and function. This is a nice and well presented paper, however the following experiments and controls are absolutely required to really complete the paper.*

*Major points:*

*– Alteration of mGlu5 signaling and altered Homer1 localization to synapses has been associated to impaired social behavior in mice. Thus it will be interesting to characterize the social and repetitive behavior in the SNX6 KO mice.*

Thanks for the very insightful comments and suggestions. Since Homer and Shank are binding partners in the PSD, and genes encoding Shank family members are causative genes for autism spectrum disorders (ASD) (Hulbert and Jiang, 2016), it is natural to reason that changes in Homer1 functions/levels/distribution might also cause autism-like phenotypes in mice. We searched literature and found that intriguingly, although mutations or deletion of Shank family members causes abnormal social and repetitive behaviors in mouse models (Jiang and Ehlers, 2013), no link between Homer and human ASD has been reported (Hulbert and Jiang, 2016). Interestingly, previous studies report that Homer1 knockout mice that lack both long and short isoforms in the brain spent more time in social interaction with a naïve WT stranger in the social dyad test than wild-type and no difference in repetitive behaviors (Yuan et al., 2003; Jaubert et al., 2007). Though counterintuitive, this might be explained by functional redundancy among the Homer family members and/or distinct molecular and cellular functions of Homer and Shank besides their common role in PSD scaffolding. Nevertheless we performed social interaction and repetitive behavior assays on the SNX6 KO mice. The results show that the KO mice did not behave significantly different from the wild-type in these assays (Figure 2), indicating that ablation of SNX6 did not affect exactly the same higher brain functions involving Homer1 and its interaction partner Shank in brain regions other than the hippocampus. As we have pointed out in the Discussion, among the Homer family members, Homer1b/c is predominantly expressed in the hippocampal CA1 region (Shiraishi et al., 2004). Homer1 knockout mouse exhibited similar phenotypes in spatial learning and memory as SNX6 knockout, which could be rescued by AAV-mediated Homer1c expression in the hippocampus (Gerstein et al., 2012). Furthermore, AAV-mediated delivery of Homer1c also improved the spatial learning deficits in a rat model of cognitive learning (Gerstein et al., 2013). Based on the previous findings on Homer1 involvement in hippocampal-dependent spatial learning and memory and our data, we conclude that ablation of SNX6 specifically impairs Homer1 functions in the hippocampus.

*– Dendrite spine morphology and synapse structure should be also analyzed in cortex.*

Thanks for the very insightful comments. Our behavior screen detected deficits in hippocampal-dependent spatial learning and memory, but not in motor coordination, mood levels and social or repetitive behaviors, indicating that ablation of SNX6 did not cause major structural/functional changes in other circuits or regions of the brain. Since our behavior tests identified impairment in hippocampal –dependent function but not functions involving other brain regions (please refer to behavior screen results shown in Figure 2 and Response to major point 1), in the current study we focused on analyses of changes in the hippocampal region at cellular and subcellular levels. We agree with the reviewer that to thoroughly investigate physiological changes in the CNS caused by SNX6 knockout, we should analyze spine morphology and synapse structure in other regions of the brain including the cerebral cortex. However, technically it is very difficult because currently we have no clue other than deficits in spatial learning and memory to help us locate the exact regions or neural circuits in the cortex for analyses at cellular and subcellular levels. Like identifying other vesicular cargoes mediated by SNX6 or other neural circuits affected by SNX6 knockout, it is our long-term goal to analyze brain regions of the KO mice other than the hippocampus and is beyond the scope of this manuscript.

*– The quality of EM images is poor and should be improved substantially.*

We apologize for the quality of the EM images. We have now replaced them with better ones, which clearly show not only excitatory synapses, but also membranous structures such as mitochondria and lysosomes.

*– The surface staining of mGlu5 is not very convincing. The higher expression of surface mGlu5 in the SNX6 KO mice should be proved also with biochemical experiments.*

Thanks for the constructive comments. To verify that surface levels of mGluR5 are increased in SNX6 KO neurons, we cultured hippocampal neurons from two pairs of WT and KO mice and performed biotinylation assay of surface proteins. We are very pleased that the results are fully consistent with the IF staining data, i.e., there is increase in surface mGluR5 and decrease in surface GluA1 and GluA2. These data are incorporated in the manuscript as Figure 8—figure supplement 1.

*Reviewer #3:*

*Niu et al., investigates the physiological role of SNX6, which is a known component of the retrograde complex. Loss of SNX6 caused deficits in a/ spine density in hippocampal CA1 neurons, b/ hippocampal-dependent spatial learning and memory, c/ synaptic homer and AMPA receptor levels, d/ AMPA receptor-mediated synaptic transmission and e/ coupling of the endocytic zone to PSD. These data suggest an important role for SNX6 in synaptic structure and function. The biochemical, electrophysiology and behavioral experiments in this paper are well executed and are high quality but the cell biological part is poorly developed. The cell biological / localization analysis / live imaging data presented here (Figure 4, Figure 5 and some panels in Figure 6 and Figure 7) are insufficient to support the major conclusions of the paper. It is possible that SNX6 directly transport homer1 but this is not shown by the data presented in this manuscript. Moreover, the paper lacks evidence for the claim that SNX6 mediates dynein-driven long range dendritic transport of Homer1b/c,*

*Major concerns:*

*1) From the current data it cannot be concluded that there is a "SNX6-mediated Homer1b/c transport pathway (Video 2–Video 4)". Overexpression of the fluorescently tagged SNX6 and homer1 look completely different than the endogenous staining patterns and do agree with previous reported subcellular distributions. Most of the SNX6 and homer1 clusters are static and do not move and are not present at synaptic sites.*

We thank the reviewer for the thoughtful comments. Please note that since Homer1b/c is a scaffold protein, the cystosolic distribution of overexpressed protein generates diffuse signals in the cell in addition to puncta, which could be extracted with the mild detergent digitonin before IF staining, while transmembrane proteins or proteins associated with membranous structures remain in the cytoplasm. First of all, to determine whether the fluorescent proteins overexpressed in cultured neurons faithfully recapitulate the distribution of endogenous proteins, we transfected neurons with corresponding constructs and performed immunofluorescence staining. To remove overexpressed proteins in cytosol we conducted digitonin extraction of live neurons before fixation and staining (please see Immunofluorescence staining, confocal image acquisition and analysis in Materials and methods for details). Confocal images indicate that mEmerald-Homer1c indeed partially colocalizes with both the endosomal marker EEA1 and endogenous SNX6 in dendrites; conversely, mCherry-SNX6 colocalizes with EEA1 and endogenous Homer1b/c. Moreover, both mEmerald-Homer1c and mCherry-SNX6 also partially overlapped with PSD95, indicating that like the endogenous proteins, the overexpressed fluorescent fusion proteins could localize to dendritic spines. These data are added to the revised manuscript as Figure 5—figure supplement 1.

Second of all, we compared the motility of mEmerald-Homer1c to that of PSD95-GFP (Gerrow et al., 2006) and found that similar to PSD95, the majority of Homer1c are immotile in dendrite. Unlike mitochondria (~30% are motile) or endosomes (Yap et al., 2008), which are actively transported in neuronal cells to fulfill their cellular functions, both Homer1c and PSD95 are scaffold proteins that stay anchored to the postsynaptic cytomatrix once they have been delivered to the postsynaptic site. Since we performed live imaging of neurons in which the fusion protein has been overexpressed to levels high enough to detect the fluorescent signals, by the time of imaging most of the protein molecules might have already been delivered to the spines. Nevertheless, in light of reviewer’s comments, we performed live imaging of mEmerald-Homer1c in Snx6+/+ and Snx6-/- neurons. We are very pleased to find out that compared with wild-type neurons, there was approximately 10-fold decrease in the fraction of motile Homer1c puncta in Snx6-/- neurons (29 out of 311 mEmerald-Homer1c puncta from 10 Snx6+/+ neurons are motile, whereas only 10 out of 1217 puncta from 40 Snx6-/- neurons are motile). These data indicate that the dendritic vesicular transport of Homer1c is indeed SNX6-dependent, which further strengthened our original conclusion. This piece of data is incorporated in the text under the subtitle “Dendritic vesicular transport and spine localization of Homer1b/c require SNX6”.

*2) There is no evidence in this paper that "SNX6 mediates vesicular transport of Homer1b/c to synaptic sites in dendrites". Other than previous data on an interaction between dynein and SNX6 and the pulldown experiments shown in Figure 4 there is no functional evidence for the role dynein in this transport pathway. The authors should provide functional data on the link between dynein and SNX6 and homer in dendritic transport.*

We thank the reviewer for the very insightful comments. However, we would like to respectfully point out that we have addressed the issue already in the original manuscript with significant amount of experimentation. We determined whether SNX6-mediated dendritic transport of Homer1b/c is dynein-dependent with two approaches: 1) we treated hippocampal neurons with ciliobrevin D, an inhibitor of dynein activity, and analyzed dendritic distribution of Homer1b/c by IF staining and confocal microscopy; 2) we overexpressed the N-terminal fragment of p150Glued, a dominant negative mutant that disrupts SNX6-p150Glued interaction, in hippocampal neurons and analyzed distribution of Homer1b/c in dendrites. Quantification results indicate that indeed, inhibition of dynein activity caused decrease in synaptic distribution of Homer1b/c (Figure 5), and disruption of motor-adaptor interaction caused accumulation of Homer1b/c in the cell body and its decrease in distal dendrites (Figure 5). Moreover, in the revised manuscript, to further confirm that dynein−dynactin drives Homer1b/c transport from the cell body to dendrites, we repeated the ciliobrevin D treatment experiment and compared the subcellular distribution of Homer1b/c in the cell body and dendritic segments. The results show that, consistent with data in the original manuscript, inhibition of dynein activity causes accumulation of Homer1b/c in the cell body and its decrease in the distal region of dendrites. Further, we performed two new experiments to determine whether SNX6 serves as cargo adaptor to link the dynein-dynactin motor complex to Homer1b/c vesicular cargo: 1) colocalization analysis of Homer1b/c and dynein−dynactin in SNX6 knockout neurons; 2) immunoisolation of Homer1b/c- and dynein−dynactin-double positive vesicles from SNX6 knockout mouse brain. The results show that 1) the colocalization between Homer1b/c with DIC and p150Glued is decreased when SNX6 is ablated, whereas the colocalization between Homer1b/c and EEA1 is not affected; 2) although the antibodies against Homer1b/c did not work for immunoisolation, antibodies to both p150Glued and DIC worked and western blotting detected Homer1b/c on DIC- or p150Glued-positive vesicles from wild-type but not SNX6-/- mouse brain. All together, these cell biological and biochemical data indicate that SNX6-mediated dendritic transport of Homer1b/c requires dynein−dynactin activity. To make it clear, we have reorganized the data, combined Figure 5 with the new data and made them into the new Figure 6.

*3) There is no evidence in this paper that there is "a novel dendritic transport pathway that contributes to synaptic structure and function" (last sentence in the Abstract). Authors should first perform high-quality imaging experiments to exclude that other 'dendritic' trafficking pathways are not involved. Such as secretory pathways (work from Mike Ehlers) and other routes transporting mobile scaffolding proteins (Gerrow et al., Neuron, 2016). Moreover, the data do not convincingly show that the phenotype in SNX6 knockout mice is retromer independent.*

We thank the reviewer for the very insightful comments and information on the previous findings. However, we would like to respectfully point out that we have already addressed the first issue in the original manuscript with experimental data. We have shown that 1) neither SNX6 nor Homer1b/c signals were detected on motile PSD95 clusters in dendrites; 2) block of the secretory pathway from the TGN in neurons by overexpressing the PKD-KD dominant mutant (Liljedahl et al., 2001) did not affect dendritic/synaptic distribution of Homer1b/c. We dedicated a section (subtitled “The SNX6-mediated Homer1b/c transport pathway is distinct from other trafficking pathways in dendrite”) and a whole figure to this question (Figure 6) in the original manuscript. It is now presented as Figure 7 in the revised version.

For the second question, we have also performed significant amount of experiments to verify that the phenotype in SNX6 knockout is retromer independent (please refer to our response to comment 6 by reviewer #1). Because VPS35 knockout is lethal (Wen et al., 2011), and we do not have the Vps35+/m mice in which the expression level of VPS35 is ~50% of wild-type, we chose to transiently knockdown its expression in hippocampal neurons (knockdown efficiency is 70-80%, Figure 9 and Figure 9—figure supplement 1) to assess its role in dendritic transport of Homer1b/c.

In the revised manuscript, taking into account that SNX6 mediates transport of Homer1b/c from the cell body to dendrites, we also performed quantitative analysis of Homer1b/c distribution in VPS35-depleted neurons and found that, in contrast to accumulation of Homer1b/c signals in the cell body and decrease in distal dendrites in SNX6 knockout neurons, the signal intensity of Homer1b/c decreases throughout the cell when VPS35 is suppressed (Figure 9). Please note that this phenotype is distinct from that caused by not only ablation of SNX6 (Figure 5), but also inhibition of dynein activity or disruption of motor-cargo interaction (Figure 6).

Further, to address reviewer #1’s concern whether VPS35 and Homer1b/c bind through SNX6, we performed co-immunostaining and confocal microscopy of endogenous proteins and found that, the colocalization between VPS35 and Homer1b/c (~10%) was not affected at all by ablation of SNX6 (Figure 9—figure supplement 2). Colocalization between Homer1b/c and SNX6 (~30%) was not affected by VPS35 knockdown either (Figure 9—figure supplement 2). Together these data indicate that VPS35 is not involved in SNX6-mediated dendritic transport of Homer1b/c that contributes to its synaptic localization. These results are now incorporated in the manuscript as Figure 9 and Figure 9—figure supplement 1 and Figure 9—figure supplement 2.

[Editors’ note: what now follows is the authors’ response after rejection of the appeal.]

*If you choose to resubmit your paper, please give particular consideration to the following points:*

*1) It would be important to improve the quality of live imaging data by showing that the fluorescent proteins used for these experiments faithfully recapitulate the distribution of endogenous proteins.*

Thanks for the great suggestion. Since Homer1b/c is a scaffold protein, the cystosolic distribution of overexpressed protein generates diffuse signals in the cell in addition to puncta, which could be extracted with the mild detergent digitonin before IF staining, while transmembrane proteins or proteins associated with membranous structures remain in the cytoplasm. To determine whether the fluorescent proteins overexpressed in cultured neurons faithfully recapitulate the distribution of endogenous proteins, we performed immunofluorescence staining of digitonin-extracted neurons transfected with corresponding constructs. Confocal images indicate that mEmerald-Homer1c indeed partially colocalizes with both the endosomal marker EEA1 and endogenous SNX6 in dendrites; conversely, mCherry-SNX6 colocalizes with both EEA1 and endogenous Homer1b/c (Please see below). Moreover, since digitonin extraction removes cytosolic proteins, we could not use overexpressed cytosolic GFP or RFP as volume marker to determine whether or not mEmerald-Homer1c and mCherry-SNX6 also distribute in dendritic spines. Instead, we performed immunostaining of the postsynaptic marker PSD95 on transfected neurons. We are very pleased to report that both mEmerald-Homer1c and mCherry-SNX6 signals partially colocalize with PSD95 (Please see below), indicating that like the endogenous proteins, the overexpressed fluorescent proteins localize in both dendritic shaft and spines. These data are added to the revised manuscript as Figure 5—figure supplement 1.

*2) It would also be important to provide clear proof for the idea that SNX6 indeed serves as a linker between Homer1b/c and cytoplasmic dynein. Please note that various strategies of inhibiting dynein are by themselves not sufficient here, as the inhibition of dynein is well known to affect most microtubule-based transport pathways independent of their identity. One would like to see proof that SNX6 really connects Homer1b/c to dynein, for example, by showing a reduced colocalization or a reduced biochemical interaction between Homer1b/c and the dynein complex in SNX6 knockout. Alternatively, you might consider a possibility that SNX6 participates in transporting Homer1b/c by simply recruiting it to endosomes, which then bind to microtubule motors independently of SNX6. It would also be useful to consider alternative ways of how SNX6 could regulate the abundance of Homer1b/c at the synapses.*

Thanks a lot for the excellent suggestions on the linker role of SNX6 in dynein-driven transport of Homer1b/c. We followed the suggestions for cell biological and biochemical assays to determine whether the association between Homer1b/c and dynein−dynactin is impaired in SNX6 knockout: 1) co-IF staining followed by colocalization analysis of confocal images shows that there is indeed a decrease in the colocalization of Homer1b/c and p150^Glued^/DIC in SNX6 knockout neurons, whereas the colocalization of Homer1b/c and EEA1 was not affected; 2) immunoisolation of membranous structures from mouse brain extracts with antibodies to p150^Glued^ or DIC detects Homer1b/csignals in wild-type but not in SNX6 knockout mice, indicating that Homer1b/c and dynein-dynactin are not on the same vesicles when SNX6 is ablated. Please note that we attempted immunoisolation with antibodies to Homer1b/c but unfortunately they did not work for this type of experiment. Moreover, we performed live imaging of mEmerald-Homer1c in dendrites of *Snx6^-/-^* neurons and found that there was approximately 10-fold decrease in the fraction of motile Homer1b/c puncta compared with that in wild-type neurons (29 out of 311 puncta from 10 *Snx6^+/+^* neurons are motile, whereas only 10 out of 1217 puncta from 40 *Snx6^-/-^* neurons are motile). These data together indicate that SNX6 serves as cargo adaptor to link Homer1b/c vesicular cargo and the motor complex, which strongly support and strengthen our original conclusion that SNX6 mediates dynein-driven vesicular transport of Homer1b/c. The results are now incorporated in the text and figures as new Figure 6.

*3) On the technical side, please note that the use of SIM microscopy is an asset, but since the improvement in resolution provided by this technique is modest, its use does not overcome the need for proper unbiased quantifications, especially when the analyzed samples are highly crowded.*

To confirm our quantification results of the 3D-SIM images (EEA1-SNX6-Homer1b/c in Figure 4, and p150^Glued^/DIC-SNX6-Hoemr1b/c in Figure 5), in which we quantified the overlap among three fluorescent labels using Mander’s coefficient as a measure of colocalization, we redid the colocalization analysis and assessed the statistical significance of the data with the methodology developed by Costes et al. (Automatic and Quantitative Measurement of Protein-Protein Colocalization in Live Cells. Biophys. J. 86, 3993-4003.) (Costes et al., 2004) and Fletcher et al. (Multi-Image Colocalization and Its Statistical Significance. Biophys. J. 99, 1996-2005) (Fletcher et al., 2010). Basically we evaluated the statistical significance of the values of voxel (corresponds to pixel in 2D images) colocalization by comparing them with those of images generated by randomizing spatial locations of signals in original images (i.e., colocalization occurred by chance). The results show that the voxel colocalization of EEA1-SNX6-Homer1b/c and that of p150^Glued^/DIC-SNX6-Homer1b/c are significantly more than chance. The quantification data are added to the manuscript as Table 1 and Figure 4—figure supplement 1 and shown below.

*4) For proving that the pathway you are analyzing is indeed retromer-independent, it would be important to demonstrate better the efficiency of the knockdown of VPS35 in neurons, because if the knockdown is only partial and not complete, this might explain the differences between VPS35 knockdown and the SNX6 knockout phenotype, especially as the loss VPS35 does affect the number of Homer1b/c puncta in dendrites. Please also refer to the comments of reviewer 1 on this issue.*

Because VPS35 knockout is lethal (Wen et al., 2011), in previous studies researchers studied its cellular and molecular functions by either using heterozygous mice (Vps35^+/m^, expression of which is ~ 50% of wild-type, (Wen et al., 2011)), suppressing its expression by RNA interference (Wang et al., 2012) or overexpressing the Parkinson’s Disease-related D620N point mutant (Munsie et al., 2015; Tang et al., 2015). Based on the previous findings on VPS35 haploinsufficiency and the role of the retromer core complex in neuronal survival and degeneration (Wen et al., 2011; Liu et al., 2014; Wang et al., 2014; Tang et al., 2015), we chose to transiently knockdown its expression in hippocampal neurons to assess its role in dendritic transport of Homer1b/c. We performed quantitative analysis of both western blots and confocal images, the results show that VPS35 knockdown efficiency is 70-80% in hippocampal neurons, which are incorporated in the manuscript as Figure 9 and Figure 9—figure supplement. Please note that when we were analyzing the fluorescent signals of VPS35, we not only quantified the mean intensity of VPS35 fluorescent signals but also the number of VPS35 puncta in the somatodendritic area of each neuron (Choy et al., 2014).

Further, we also quantified the subcellular distribution of Homer1b/c in VPS35 knockdown neurons to determine whether VPS35 acts in SNX6-mediated transport pathway. Quantification results show that VPS35 knockdown causes decrease in Homer1b/c signals throughout the cell (Figure 9), whereas the signals accumulate in the soma and decrease in dendritic segments distal to the cell body of SNX6 knockout neurons. Please note that this phenotype is distinct from that caused by not only ablation of SNX6, but also inhibition of dynein activity or disruption of motor-cargo interaction shown in Figure 6.

Moreover, we performed confocal microscopy and colocalization analysis of Homer1b/c and VPS35 in wild-type and SNX6 knockout neurons. The results show that the colocalization of Homer1b/c with VPS35 in dendrite, which is much lower than its colocalization with SNX6 and EEA1, is not affected by SNX6 knockout (Figure 9—figure supplement 2). Conversely, colocalization of Homer1b/c with SNX6 is not affected by VPS35 knockdown either (Figure 9—figure supplement 2). Collectively these results indicate that VPS35 is not involved in dynein-driven, SNX6-mediated vesicular transport of Homer1b/c from the hippocampal cell body to dendrites.

Taken together, these results are supportive of the findings reported in the original manuscript. Based on the extended amount of experimental data we re-arranged the figures to better support the flow of the story. In total, this improved manuscript contains one new figure, one table, six new supplementary figures, and numerous modifications/improvements to the existing figures, all of which support and greatly strengthen our original conclusions. Since we could not find enough room for new data in the original figures, we have reorganized the figures to accommodate the new results in the revised manuscript as the following: 1) we split Figure 5 into two, the new Figure 5 and Figure 6, so Figure 5–Figure 8 are now Figure 5–Figure 9, respectively; 2) we combined Figure 4, Figure 4—figure supplement 1 to make it the new Figure 4; 3) we added immunostaining and confocal data for mEmerald-Homer1c and mCherry-SNX6 as Figure 5—figure supplement 1; 4) we added data from quantitative analysis of 3D-SIM images as Table 1 and new Figure 4—figure supplement 1;,2) we added data for subcellular distribution of mEmerald-Homer1c and mCherry-SNX6 as Figure 5—supplement 1; 6) we added data for biotinylation assay of surface receptors in wild-type and KO neurons as Figure 8—figure supplements 1;,2) we added data for VPS35 knockdown efficiency as Figure 9—figure supplement 1;,5) we added data for Homer1b/c colocalization with SNX6 (in VPS35 KD) and VPS35 (in *Snx6^-/-^*) as Figure 9—figure supplement 2.

[Editors’ note: the authors’ response after a second round of full peer review follows].

*First, if Homer1b/c localizes to early endosomes and is transported with them, then it is not a novel pathway. Moreover, as the authors show in Figure 6, SNX6 knockout does not affect the 40% colocalisation betweev n Homer1b/c and an early endosome marker. But then, surprisingly, SNX6 knockout does affect Homer1 motility and colocalization with dynein. Do the authors actually mean that the complex of SNX6 and Homer1b/c is required to transport early endosomes into dendrites? The authors should be much more explicit on this issue, as other molecules have been implicated in endosome transport. If Homer1b/c and SNX6 are transported on early endosomes, then SNX6 knockout should affect endosome motility. This should be easy to measure, and such data should be included. Alternatively, if SNX6 in complex with dynein transports the non-endosomal Homer1 population, what kind of vesicles are these, and what is then the relevance of colocalization of Homer1 with endosomes?*

We appreciate the comments of the reviewers. It looks like the reviewers get the impression that SNX6 transports all early endosomes into dendrites. We apologize for the confusion/misunderstanding that might have been caused by our failure to make this aspect of the paper clearer. Based on the data that Homer1b/c and SNX6 colocalize on early endosomes and dendritic transport of Homer1b/c requires SNX6 and dynein‒dynactin, we propose that SNX6 serves as dynein cargo adaptor to mediate vesicular transport of Homer1b/c. In the absence of SNX6, the link between the motor and Homer1b/c-associated vesicles is missing, so that there is a decrease in colocalization between Homer1 and dynein. Please note that we did not claim that SNX6 links Homer1 to endosomes. Instead, current data support that SNX6 recognizes the endosomal carrier for Homer1b/c via its direct interaction with the cargo molecule. Indeed, it is an excellent question how active transport of endosomes carrying different cargoes is regulated by their cargo adaptors. In fact, during the study on SNX6’s role in dendrite we searched the literature on dendritic trafficking. We found that 1) most studies focus on recycling endosomes and receptor trafficking to the postsynaptic site, mechanisms for long-range transport of endosomal cargoes and their adaptors for molecular motors remain to be identified; 2) the motility of endosomes labeled by different cargoes is not the same. Research conducted by Dr. Bettina Winckler and her colleagues at University of Virginia has identified Neep21 as a regulator for early endosomal sorting and trafficking in dendrites (Yap et al., 2008). Intriguingly, they found by live imaging that only 7% of EEA1-positive endosomes are motile, compared with 63% of Neep21-positive endosomes (Lasiecka et al., 2014). Moreover, the run length and velocity of these two types of vesicles are also quite different from each other (Lasiecka et al., 2014). These findings suggest that there are multiple mechanisms underlying endosomal transport. To make sure we have not missed any most recent progress on this topic, we consulted an expert about endosome motility in dendrites, who was very nice to provide us the following information based on their published and unpublished imaging data:

1) Motility of vesicles labeled with various endosomal markers vary a lot.

2) There are many types of endosomes, so the motor adaptor for one specific cargo might not control the behavior of all endosomes equally and it might be difficult to see an effect of gene knockout with just any endosomal marker.

3) If less than 10% of vesicles are motile, then a decrease in motility will be hard to observe.

In retrospect, given the role of early endosomes as sorting station for a wide variety of cargoes, and the ~ 15% colocalization among EEA1, Homer1 and SNX6 (Figure 4—figure supplement 1, measured with more stringent method for quantification of images obtained by 3D-SIM, rather than conventional confocal microscopy), it is highly possible that the motile SNX6-, Homer1-positive vesicles are transport carriers derived from EEA1-positive endosomes and they might have lost the marker(s) upon sorting/fission from the endosomal membrane or en route to their final destination. That is to say, they do not represent all early endosomes or endosome-derived vesicles carrying different cargoes. Given the heterogeneity of early endosomes in composition and motility and the various transport pathways they might adopt (recycling to the plasma membrane, retrograde transport to the TGN, and trafficking to, conversion to late endosome and fusion with lysosomes, etc.), even if we carry out the live imaging experiment with a well-known endosomal marker protein, e.g., EEA1 or Rab5, questions still remain as to whether EEA1- or Rab5-labeled vesicles represent a proportion or all types of endosomal carriers derived from early endosomes, and whether SNX6 mediates transport of all endosomal carriers derived from EEA1- or Rab5-positive endosomes. Therefore, we would appreciate your opinion on this issue. To avoid confusion, we will tone down the statement about novel transport pathway (How about “previously unknown” instead of “novel”?), rewrite the sentences in the Results and Discussion sections to make it clearer and modify the Abstract accordingly.

*Furthermore, the new data provided by the authors show that fluorescently tagged Homer1b/c is present in large structures, many of which do not colocalize with a PSD marker. In contrast, the endogenous Homer1b/c is well known to show a synaptic localization. It is thus possible that the non-synaptic fluorescently tagged Homer 1b/c is present in aggregates, and that loss of SNX6 affects the very infrequent motility of these aggregates. The motility of fluorescent Homer1b/c particles might then be irrelevant to the distribution of synaptic Homer1. It is possible that the synaptic localization of Homer1b/c actually depends on protein diffusion and not on microtubule-based transport, an option that is not even properly discussed. The authors should use FRAP to investigate the turnover of the synaptic population of Homer1b/c in order to find out whether synaptic Homer1 exchanges with the soluble cytosolic pool of the protein or traffics into synapses as particles, and whether any of these processes are affected by the loss of SNX6.*

*If the authors cannot satisfactory uncover the nature of the "new dendritic transport pathway" that they propose or if it turns out that Homer1b/c accumulates in the synapses by exchanging with the soluble pool of the protein, the title, Abstract, the text and the conclusions of the paper will need to be very thoroughly revised accordingly.*

First, we would respectfully disagree with reviewer’s comment on large structures of Homer1c-EGFP as “aggregates”. As those fluorescent puncta are digitonin resistant, they are either membrane- or PSD-bound (shown by confocal images of colocalization between Homer1c-EGFP and PSD95, Figure 5—figure supplement 1), making them no longer cytosolic and easily removed by detergent extraction. Second, we very much appreciate the wish to fully understand the mechanism of Homer1b/c delivery from dendritic shaft to spines and the role of SNX6 in this process. We apologize for not being able to explain explicitly that our current findings do not suggest that SNX6 is directly responsible for transport of Homer1b/c, whether it associates with vesicles or not, from dendritic shaft to spines. Besides the live imaging data showing that the motility of Homer1b/c vesicles is greatly impaired in SNX6 knockout neurons, IF staining and microscopy analysis show that Homer1b/c distribution in distal dendrites is reduced in the absence of SNX6, indicating that long-range vesicular transport of Homer1b/c from the cell body to dendrite is SNX6-dependent. To verify that failure of Homer1b/c transport to distal dendrites leads to its loss from dendritic spines and decrease in spine density of distal dendrites, we went back to the original confocal images and quantified spine localization of Homer1b/c over distance from the soma. The results are as expected and are in good agreement with a decrease in Homer1b/c signals in distal dendrite (Figure 12), which we would incorporate in the manuscript. Although we did detect SNX6 signals in spines by IF staining, we have not obtained any direct evidence showing that transport of Homer1b/c into spines, whether microtubule- and dynein-dependent or not, is mediated by SNX6. Therefore, our main conclusion in this study is that SNX6 mediates long-range transport of Homer1b/c in dendritic shaft, which is required for its synaptic localization.

Author response image 2.DIV14 neurons were cotransfected with constructs overexpressing EGFP and mCherry or mCherry-SNX6, fixed on DIV16 and immunostained with antibodies to Homer1b/c.Shown is quantification of Homer1b/c signal intensity in dendritic spines over distance from the cell body (mean ± SEM, n = 30, N = 3).**DOI:**
http://dx.doi.org/10.7554/eLife.20991.043

To determine whether SNX6 is also required for delivery of Homer1b/c from shaft to spines, which is short-distance transport compared with long-range movement on microtubule tracks, we need to test the role of microtubules and actin filaments as well as dynein and the actin-based motor myosin, which is beyond the scope of this manuscript. We appreciate the editor/reviewer suggesting the FRAP experiment to determine whether Homer1b/c enters dendritic spine by active transport or diffusion. However, since overexpressed Homer1c-EGFP is in excess, in addition to its presence on vesicles, lots of the molecules are cytosolic, and those that have been delivered to spine incorporate into the PSD, forming higher-order polymerized complex with Shank that is necessary for the structural and functional integrity of spines (Bosch et al., 2014; Hayashi et al., 2009; Meyer et al., 2014), As a consequence of overexpression, there would be three pools of Homer1c-EGFP, namely vesicular, cytosolic and PSD-associated. Given that Homer1 in spines complexes with other PSD scaffold proteins to form a matrix structure, for neurons at steady state, it is highly unlikely the PSD-associated Homer1 exchanges with the free molecules in the cytosolic pool. Instead, we would rather expect addition of Homer1 to the expanding PSD during synaptic plasticity when the spine head expands. However, testing this idea is beyond the scope of this manuscript. Besides, if there are plenty of proteins in the cytosol, most likely those free molecules enter and leave spines by diffusion, which might just be an overexpression artifact. So we would expect that FRAP measures the diffusion rate of cytosolic Homer1 in spine and shaft area adjacent to the spine, rather than exchange between the synaptically incorporated and the cytosolic pools. As for vesicular Homer1, since we have already found that only ~ 10% of Homer1 puncta are motile (similar to PSD95 puncta in the 2006 Gerrow paper), considering the limited number of spines in the viewfield of high magnification lens under microscope, it is technically very difficult to catch those vesicles entering spines if there are any, let alone monitor the exchange of the motile puncta with synaptic Homer1 by FRAP. Moreover, previous studies using FRAP to measure mobility of PSD95 bound in the PSD do not provide strong support for the presumption that there is rapid exchange between synaptic and cytosolic PSD scaffold protein molecules (Blanpied et al., 2008). Again, we appreciate the wonderful question and suggestion from reviewers, and we would be more than happy to try our best to do the FRAP experiment. However, with the caveats and potential problems discussed above, FRAP results might not provide useful information to answer the question whether SNX6 is required for delivery of Homer1 from shaft to spine but rather means for investigation of Homer1 turnover in spine.

[Editors' note: further revisions were requested prior to acceptance, as described below.]

*In the revised version of the paper, the authors performed the experiments suggested by the reviewers. The results most relevant to the overall message of the paper are that the labeled Homer1c structures only very rarely enter spines and that photobleached Homer1c in spines shows SNX6-independent recovery by 50% within 10 minutes through exchange with the soluble pool. These are very significant results, because together, they provide strong support for the view that Homer1c is not delivered into spines on endosomes but simply exchanges with the soluble pool in a SNX6-independent manner, which is not unexpected. Since most of dendritic Homer1c is present in spines, but vesicular transport is not responsible for delivering Homer1c into spines, the role of SNX6-dependent vesicle transport in "dendritic delivery of Homer1b/c" becomes confusing. Is it needed to redistribute soluble Homer1c through dendrites, so that it can then diffuse into spines? Or does Homer1c on endosomes represent part of a Homer1c pathway distinct from Homer1c function in dendritic spines? Based on the data presented in the paper, these questions are impossible to address. However, the data as they stand now certainly bring the importance of the observed SNX6-dependent transport of Homer1c on endosomes into question.*

*Given the large amount of work the authors have invested in revising the paper, the good quality of much of the data and the complexity of the problem, the reviewers felt that the paper can still be published. However, it would be essential to very thoroughly revise the text of the paper in a way that would do justice to the data shown, and not just to the model that the authors were trying to prove. To achieve this, it would be essential to very strongly downplay the "SNX6-mediated transport and delivery" angle, particularly by removing it from the title, re-writing the Abstract and Introduction, adding the data on FRAP and the failure to detect vesicle-based delivery of Homer1c into spines into the main figures, and writing a very balanced discussion. The revised version of the paper should include all the data but should not attempt to create an impression that the authors have proven that the phenotypes of SNX6 knockout mice are due to altered transport and delivery of Homer1c on SNX6-positive endosomes into dendritic spines, thus explaining the spine phenotypes observed. The reviewers would like to emphasize that "cosmetic" changes to the writing will not be sufficient in this case.*

*It is possible that the authors might disagree with the opinion of the reviewers, and in this case they are advised to seek publication in another journal.*

We would like to thank the editors and the reviewers for their thoughtful comments and advice. We have revised the text accordingly.

1) As requested by editors/reviewers, we have changed the title from “Dendritic Delivery of Homer1b/c by SNX6-mediated Long-range Vesicular Transport” to “A Neuronal Role for SNX6 in Dendritic Trafficking of the Postsynaptic Scaffold Protein Homer1b/c”.

2) We revised the Abstract as the following:

“SNX6 is a ubiquitously expressed PX-BAR protein that plays important roles in dynein‒dynactin-driven, retromer-mediated retrograde vesicular transport from endosomes. Here we show that CNS-specific Snx6 knockout mice exhibit deficits in spatial learning and memory, accompanied with loss of spines from distal dendrites of hippocampal CA1 pyramidal cells. We find that SNX6 functions independently of retromer to mediate dynein‒dynactin-driven dendritic vesicular transport of Homer1b/c, a postsynaptic scaffold protein crucial for synaptic distribution of other postsynaptic proteins and structural integrity of dendritic spines. Ablation of SNX6 causes loss of Homer1b/c from distal dendrites as well as decreases in surface levels of AMPAR and AMPAR-mediated synaptic transmission. As trafficking of Homer1b/c from its site of biosynthesis to dendrites is vital for synapse formation and functioning, these findings reveal an important physiological role of SNX6-mediated long-range vesicular transport in CNS neurons.”

We also revised the Introduction as requested, adding a paragraph to distinguish trafficking of postsynaptic proteins from the cell body to dendrites from their trafficking from dendritic shaft to spines (please see highlighted text in the Introduction). Accordingly, in the Results section we present data to show that SNX6 is required for transport of Homer1b/c from the cell body to dendrites, but not translocation of Homer1b/c from shaft to spines. To do this, we added the data on FRAP, live imaging and quantitative results from confocal microscopy to the main figures as new Figure 8, under a new subtitle “Translocation of Homer1b/c from dendritic shaft to spines is SNX6-independent”, after the data showing that SNX6 mediates dynein-driven transport of Homer1b/c to distal dendrites.

3) We agree with editor/reviewers’ view that Homer1c is not delivered into spines on endosomes but rather in the form of free cytosolic proteins. We have modified the Discussion to make it more balanced. We modified the opening paragraph to tone down our conclusions, discussed possible mechanisms for translocation of Homer1b/c from shaft to spines, and discussed our new data indicating that entry of Homer1b/c into spines is SNX6-independent as the following:

“Once in dendrites, several mechanisms exist for transfer of postsynaptic components from shaft to synaptic sites in spines, including cytosolic diffusion, exocytosis of transmembrane proteins at the plasma membrane and lateral diffusion to synaptic sites, and active transport by molecular motors. The AMPARs enter dendritic spines via both lateral diffusion and actin-based, Myosin V-driven transport of recycling endosomes (Adesnik et al.et al., 2005; Correia et al.et al., 2008; Makino and Malinow, 2009; Wang et al.et al., 2008; Yudowski et al.et al., 2007). Since Homer1b/c is a scaffolding protein, its entry into spines might rely on diffusion of free molecules, possibly released from endosomal carriers, or transport of the vesicular cargo directly from the shaft by a different motor. Our results indicate that direct spine entry of Homer1c vesicles is an extremely rare event, and that neither spine:shaft ratio nor dynamic turnover of Homer1b/c in spines is affected by ablation of SNX6, suggesting that most likely Homer1b/c enters spines in the form of free cytosolic molecules.”

4) Following the editors’ and reviewers’ advice, to interpret the data precisely, we also changed one of the subtitles from “Dendritic vesicular transport and spine localization of Homer1b/c require SNX6” to “Dendritic vesicular transport of Homer1b/c requires SNX6”.

*Should you decide to resubmit, please address this additional point: the data of panel K of Figure 8 on the surface staining of mGlu5 do not look convincing. Please move them to the Supplement and move the supplementary biochemical data (Figure 8—figure supplement 1) to the main figure.*

Changes made as requested.

[Editors' note: further revisions were requested prior to acceptance, as described below.]

*Specifically, in the previous decision letters we have very clearly indicated that we cannot offer to publish your paper in the current form, because the reviewers found that your major claim that " SNX6 mediates long-range vesicular transport of Homer1b/c to distal dendrites" is not sufficiently supported by data. With other words, it is possible that SNX6 indeed mediates long-range transport of Homer1b/c, but it is also possible that Homer1b/c arrives to the dendritic spines by diffusion alone, and the interactions that you described serve another function (if any). Given the complexity of the subject and the large amount of work that you have put into revising this manuscript, we have offered you an opportunity to completely and thoroughly revise the whole paper in a way that would make it clear to the reader from the start to the end that your transport-related conclusions are not more than one of the possible explanations of the mouse phenotypes that you nicely describe. Such re-writing would entail not a minor but a major revision of your paper. Given your new suggestion for the title and the minor edits you have done, it is very clear to us that you are not willing to perform such a major revision. We of course fully respect your decision, because this is your paper, and you should write it in the way you find most appropriate. We would like to emphasize that your English writing is perfectly fine, and that our negative decision is only due to the fact the reviewers were not convinced that your major conclusion is sufficiently supported by data. We would also like to emphasize that since the reviewers are not co-authors of your paper, it is not their task to search for "sentences or words in the manuscript still misleading or overstated" – it is up to authors to make such changes.*

*We are willing to give you one more chance to re-write the paper. However, you would need to really thoroughly rethink and revise the whole manuscript to provide a more balanced view of different ways in which their data can be explained. You are very much in favor of pushing through the model that SNX6 mediates transport of Homer1b/c on vesicles, but the experimental support for this model is insufficient and the FRAP data directly contradict it. Therefore, the paper needs to be re-written in a way that would make it clear to the reader that transport of Homer1b/c on endosomes is not more than one of the possible explanations of the observed mouse phenotypes. Trying to solve the problem by adding a few disclaimers here and there is not enough, and also not what we have asked for. For example, in the latest version, you proposed to revert to the title with transport being the main message, while we asked for this change one revision before.*

We thank the editors and reviewers for their thoughtful comments and advice. After careful scrutiny of our data and conclusions, we realized that indeed, first, SNX6-mediated transport serves as just one of the possible mechanisms for Homer1 distribution in distal dendrites, and there are other possibilities that have not been explored in our current study. Second, defects in Homer1 distribution might also provide just one of the explanations for the mouse phenotypes we observed. Besides Homer1, SXN6 might interact with proteins that also play important roles in synaptic structure and function, a possibility we did not discuss in the previous versions of the manuscript. We also thank the reviewer for the great suggestion on FRAP, which shows SNX6-independent diffusion of free Homer1c molecules into the spine and dynamic exchange between the synaptic and the soluble pools. The FRAP data, in combination with the results from live imaging and quantitative analysis of the spine:shaft ratio of endogenous Homer1b/c, indicates that SNX6 and vesicular transport are not involved in its translocation from shaft to spines, and that Homer1 can enter spines by diffusion.

Therefore, we have revised the text throughout the manuscript to make the points above clear to the reader. Since it is very difficult to reorganize the huge amount of data in this study, we did not change the data organization of the manuscript significantly. Nevertheless, we have changed the title, rewritten the Abstract and revised the concluding paragraph of the Introduction; we have also revised the results, reorganized Figure 5 and Figure 6 to downplay the contribution of SNX6-mediated vesicular transport to the phenotypes, changed the conclusions for some of the subsections, and changed some of the subtitles for figures and results accordingly; we have also rewritten the Discussion section to provide a more critical and balanced review of our data. Specifically, we have made it clear that ablation of SNX6 only impairs the motility of a fraction of Homer1c molecules on vesicles in the dendritic shaft; we have also pointed out that mechanisms other than vesicular transport could contribute to the distribution of Homer1b/c in distal dendrites in both Results and Discussion; in addition to data presented in Results, discussion about the possible mechanisms for shaft to spine translocation of Homer1 and our conclusions in Discussion, we have added the conclusion on SNX6-independent Homer1 diffusion into spine in both Abstract and Introduction. The major changes in the revised manuscript are listed below.

1) To summarize our findings on the physiological role of SNX6 in CA1 neurons, we have changed the Title to “Ablation of SNX6 leads to defects in synaptic function of CA1 pyramidal neurons and spatial memory”. We think this descriptive title is the most objective.

2) We revised the Abstract as follows:

“SNX6 is a ubiquitously expressed PX-BAR protein that plays important roles in retromer-mediated retrograde vesicular transport from endosomes. Here we report that CNS-specific Snx6 knockout mice exhibit deficits in spatial learning and memory, accompanied with loss of spines from distal dendrites of hippocampal CA1 pyramidal cells. SNX6 interacts with Homer1b/c, a postsynaptic scaffold protein crucial for synaptic distribution of other postsynaptic density (PSD) proteins and structural integrity of dendritic spines. We show that SNX6 functions independently of retromer to regulate distribution of Homer1b/c in the dendritic shaft. We also find that Homer1b/c translocates from shaft to spines by protein diffusion, which does not require SNX6. Ablation of SNX6 causes reduced distribution of Homer1b/c in distal dendrites, decrease in surface levels of AMPAR and impaired AMPAR-mediated synaptic transmission. These findings reveal a physiological role of SNX6 in CNS excitatory neurons.”

3) To revise the Introduction, in the opening paragraph we described the known functions of SNX6 and the rationale for this study. In the second paragraph we introduced functions of the PSD scaffold proteins in dendritic spines and raised the question about mechanisms for their proper distribution in dendrites. In the last paragraph we described our main findings as follows:

“In this study, we investigated the physiological function(s) of SNX6 in mouse CNS neurons using multiple approaches including mouse genetics, behavior assays and electrophysiology, biochemistry and fluorescence imaging. Ablation of SNX6 in the CNS causes deficits in spatial learning and memory, decrease in spine density of the distal dendrites of hippocampal CA1 neurons and impairment of their AMPAR-mediated synaptic transmission, suggesting a role for SNX6 in synaptic structure and function. SNX6 interacts with Homer1b/c and loss of SNX6 leads to reduction in its distribution in distal dendrites. Intriguingly, although SNX6 is required for the motility of a subpopulation of Homer1c on vesicles in dendritic shaft, live imaging and FRAP analyses indicate that Homer1c enters dendritic spines via protein diffusion but not SNX6-dependent active transport. Overexpression of SNX6 or Homer1c restores the spine density and AMPAR surface levels of Snx6^-/-^ neurons. These findings uncover a physiological function for SNX6 in hippocampal CA1 excitatory neurons.”

4) In the Results section, to downplay the role of SNX6-mediated transport in dendritic distribution of Homer1b/c, we moved the data on live imaging of Homer1c from Figure 5 to Figure 6. We hope that this change would help to prevent creating the impression that reduced Homer1b/c distribution in distal dendrite is solely caused by defect in SNX6-mediated vesicular transport. In the new Figure 5, we present the data that ablation of SNX6 causes reduced distribution of Homer1 in distal dendrites. In Figure 6–Figure 8, we present a series of data obtained in an effort to understand mechanism(s) underlying the Homer1b/c distribution phenotype. First we present the data about the effect of SNX6 knockout on the motility of Homer1 vesicles in the dendritic shaft, and the involvement of dynein-dynactin in Homer1 distribution in dendrite (new Figure 6). Then we explore the possibility whether two known trafficking pathways are involved in dendritic trafficking of Homer1 (Figure 7). In Figure 8, we present data on SNX6-independent translocation of Homer1b/c from shaft to spines via protein diffusion, indicating that SNX6 only functions in distribution of Homer1b/c in the dendritic shaft but not its spine localization.

In addition, we have also modified the text, replacing “SNX6-mediated transport” with “SNX6-regulated dendritic distribution” wherever applicable to avoid confusion or overstatement about the role of vesicular transport in regulating dendritic distribution of Homer1b/c.

5) We have rewritten the Discussion to make it more balanced. To this end, we changed the focus of discussion from SNX6’s role in trafficking to mechanisms underlying dendritic distribution and synaptic localization of Homer1c. We not only discussed alternative mechanisms for dendritic distribution of Homer1b/c, but also removed the part discussing the potential role of SNX6-mediated vesicular transport of Homer1 in synaptic plasticity.

i) In the opening paragraph, we described our findings on the mouse phenotypes caused by SNX6 knockout:

“In this study, we demonstrate that ablation of SNX6 in the CNS causes deficits in spatial learning and memory, a hippocampal-dependent brain function. At the cellular level, loss of SNX6 causes decrease in spine density in the distal apical dendrites of CA1 hippocampal cells and impairment of their AMPAR-mediated synaptic transmission, indicating that SNX6 is required for synaptic structure and function of these excitatory neurons. We also show that SNX6 directly interacts with Homer1b/c, a PSD scaffolding protein crucial for the structural and functional integrity of dendritic spines, and that there is decrease in Homer1b/c distribution in distal dendrites in Snx6^-/-^ neurons. Moreover, the spine density and surface AMPAR level phenotypes of Snx6^-/-^ neurons could be rescued by overexpressing Homer1b/c or SNX6. These findings reveal an important physiological function of SNX6 in the CNS excitatory neurons.”

ii) In the second paragraph, in addition to discussion about the link between Homer1b/c expression in CA1 neurons and impairment in the CA3-CA1 pathway-dependent brain function of SNX6 knockout animals, we discussed the possibilities that 1) it could function via interaction with neuronal proteins other than Homer1 and 2) SNX6 could function in synaptic structure and function of neurons in the cortex:

“…Whether there are other SNX6-interacting proteins that are also required for CA1 neuron function awaits further investigation. As spatial learning and memory involve not only the hippocampus but also other cortical areas such as the entorhinal cortex and the medial prefrontal cortex (Jo et al., 2007; Nagahara et al., 1995; Nakazawa et al., 2004; Steffenach et al., 2005; Zhou et al., 1998), it also remains to be determined whether and how ablation of SNX6 affects the synaptic structure and function of neurons in other parts of the cortex.”

iii) In the third paragraph, we discussed the possibility that SNX6 functions via mechanism(s) distinct from dynein-driven transport to regulate dendritic distribution of Homer1, and proposed alternative mechanisms such as diffusion, cotransport with other proteins and active transport driven by different motor/adaptor.

“…We found that the motility of Homer1c-associated vesicles in dendritic shaft requires SNX6, and that ablation of SNX6 or inhibition of dynein-dynactin activity causes reduction in the amount of Homer1b/c in distal dendrites. Previously imaging assays and quantitative modeling have established that dynein-driven bidirectional transport contributes to polarized targeting of dendrite-specific cargo (Kapitein et al., 2010). Therefore, lack of dynein‒dynactin-driven transport in the dendritic shaft provides a possible mechanism for the Homer1b/c distribution phenotype in Snx6^-/-^ neurons. However, since the majority of Homer1c puncta are immobile in dendrites of steady-state neurons, and little is known about the cellular functions of SNX6 apart from its role as dynein cargo adaptor, it is also possible that SNX6 regulates the distribution of Homer1b/c in dendrites via mechanism(s) distinct from dynein‒dynactin-driven transport. Moreover, ablation of SNX6 does not cause complete loss of Homer1b/c from distal dendrites, indicating that mechanism(s) other than SNX6-mediated transport contributes to its localization to dendritic shaft far from the cell body. Since disruption of the secretory pathway does not affect Homer1b/c localization to dendritic shaft and spines, alternative mechanisms for its distribution in dendrites include diffusion of free protein molecules, cotransport with proteins other than the PSD95-GKAP-Shank complex or vesicular transport mediated by different motor(s) and/or adaptor(s).”

iv) In the fourth paragraph, first we described the observation that the spine:shaft ratio of Homer1b/c did not change in distal dendrites in SNX6 KO neurons, then we presented possible mechanisms for translocation of Homer1b/c from shaft to spines, finally we discussed our data indicating that entry of Homer1b/c into spines via SNX6-independent protein diffusion.

“Notably, in Snx6^-/-^ neurons, although there was a decrease in the amount of Homer1b/c in both shaft and spines of distal dendrites (Figure 5), the spine:shaft ratio of its signals remained constant throughout the dendrite (Figure 8), indicating that once in dendrite, Homer1b/c could enter the spines via SNX6-independent mechanism(s). In dendrites, several mechanisms exist for transfer of postsynaptic components from shaft to synaptic sites in spines, including cytosolic diffusion, exocytosis of transmembrane proteins at the plasma membrane and their lateral diffusion to synaptic sites, and active transport by molecular motors. The AMPARs enter dendritic spines via both lateral diffusion and actin-based, Myosin V-driven transport of recycling endosomes (Adesnik et al., 2005; Correia et al., 2008; Makino and Malinow, 2009; Wang et al., 2008; Yudowski et al., 2007). Since Homer1b/c is a scaffolding protein, its entry into spines might rely on diffusion of free molecules, possibly released from endosomal carriers or from the cytosolic pool in the shaft, or transport of the vesicular cargo directly from the shaft by a different motor. Our results from live imaging, FRAP and quantitative analyses show that direct spine entry of Homer1c puncta is an extremely rare event, and that the dynamic turnover of Homer1c in spines is not affected by ablation of SNX6. Collectively these data indicate that in steady-state neurons, Homer1b/c enters spines by cytosolic diffusion, and SNX6 is not required for its spine localization.”

v) In the last paragraph, we discussed the findings that SNX6 does not require retromer activity to regulate dendritic distribution of Homer1b/c as well as the potential roles of SNX family members in the CNS. We replaced “SNX6-mediated dendritic transport of Homer1b/c” with “SNX6-regulated distribution of Homer1b/c in dendritic shaft”.

6) Corresponding changes in subtitles in Results are listed below:

– “SNX6 associates with Homer1b/c on vesicles in dendritic shaft” is replaced with “SNX6 directly interacts with Homer1b/c”.

– “Dendritic vesicular transport of Homer1b/c requires SNX6” is replaced with “Ablation of SNX6 causes decrease in distribution of Homer1b/c in distal dendrites”.

– “SNX6-mediated dendritic transport of Homer1b/c is driven by dynein−dynactin” is replaced with “Active transport of a fraction of Homer1b/c molecules in the dendritic shaft requires SNX6”.

– “The SNX6-mediated Homer1b/c transport pathway is distinct from other trafficking pathways in dendrite” is replaced with “The Homer1b/c trafficking pathway is distinct from the PSD95 and secretory trafficking pathways in dendrite”.

– “Translocation of Homer1b/c from dendritic shaft to spines is SNX6-independent” is replaced with “Translocation of Homer1b/c from dendritic shaft to spines is SNX6- and vesicular transport-independent”.

– “Changes in synaptic transmission and receptor trafficking in Snx6^-/-^ neurons” is replaced with “Ablation of SNX6 causes impairment of AMPAR-mediated synaptic transmission and decrease in AMPAR surface expression”.

– “SNX6-mediated dendritic vesicular transport of Homer1b/c is retromer-independent” is replaced with “Activity of the retromer core complex is not required for SNX6-regulated dendritic distribution of Homer1b/c”.

7) Corresponding changes in subtitles for the Figures are listed below:

– Figure 5. “SNX6 mediates vesicular transport of Homer1b/c to synaptic sites in dendrites” is replaced with “Partial loss of Homer1b/c from distal dendrites of Snx6^-/-^ neurons”.

– Figure 6. “SNX6-mediated vesicular transport of Homer1b/c requires dynein−dynactin activity” is replaced with “SNX6 is required for motility of Homer1b/c vesicles in dendritic shaft and their association with dynein−dynactin”.

– Figure 8. “Translocation of Homer1b/c from dendritic shaft to spines does not require SNX6” is replaced with “Homer1b/c enters spines by SNX6-independent protein diffusion”.

– Figure 9. “Altered synaptic structure and function in SNX6-deficient neurons” is replaced with “Impaired synaptic transmission and decreased surface AMPAR levels of Snx6^-/-^ neurons”.

– Figure 10. “Long-range vesicular transport and synaptic localization of Homer1b/c is retromer-independent” is replaced with “The retromer core complex is not required for SNX6-regulated dendritic distribution of Homer1b/c”.